# Characterization of tandem aerosol classifiers for selecting particles: implication for eliminating multiple charging effect

Yao Song[1], Xiangyu Pei[1], Huichao Liu[1], Jiajia Zhou[1], Zhibin Wang[1,2,3*]

[1]College of Environmental and Resource Sciences, Zhejiang Provincial Key Laboratory of Organic Pollution Process and Control, Zhejiang University, Hangzhou 310058, China
[2]ZJU-Hangzhou Global Scientific and Technological Innovation Center, Hangzhou 311200, China
[3]Key Laboratory of Environment Remediation and Ecological Health, Ministry of Education, Zhejiang University, Hangzhou 310058, China

*Correspondence to*: Zhibin Wang (wangzhibin@zju.edu.cn)

**Abstract.** Accurate particle classification plays a vital role in aerosol studies. Differential mobility analyzer (DMA), centrifugal particle mass analyzer (CPMA) and aerodynamic aerosol classifier (AAC) are commonly used to select particles with a specific mobility diameter, aerodynamic diameter or mass, respectively. However, multiple charging effects cannot be entirely avoided when using either individual techniques or tandem systems such as DMA-CPMA, especially when selecting soot particles with fractal structures. In this study, we calculate the transfer functions of the DMA-CPMA and DMA-AAC in static configurations for flame generated soot particles. We propose an equation that constrains the resolutions of DMA and CPMA to eliminate the multiple charging effect when selecting particles with a certain mass-mobility relationship using the DMA-CPMA system. The equation for the DMA-AAC system is also derived. For DMA-CPMA in a static configuration, our results show that the ability to remove multiply charged particles mainly depends on the particle morphology and resolution settings of the DMA and CPMA. Using measurements from soot experiments and literature data, a general trend in the appearance of multiple charging effect with decreasing size when selecting aspherical particles is observed. As for DMA-AAC in a static configuration, the ability to eliminate particles with multiple charges is mainly related to the resolutions of the classifiers. In most cases, the DMA-AAC in a static configuration can eliminate multiple charging effect regardless of the particle morphology, but multiply charged particles will be selected when decreasing the resolution of the DMA or AAC. We propose that the potential influence of the multiple charging effect should be considered when using the DMA-CPMA or DMA-AAC systems in estimating size- and mass-resolved optical properties in field and lab experiments.

## 1 Introduction

Atmospheric aerosol particles span a wide size range from 1 nm to > 100 μm. A significant size dependence of aerosol physicochemical properties has been widely reported. Particle size can strongly alter the hygroscopic behavior (Biskos et al., 2006), phase state (Cheng et al., 2015) and cloud-nucleating ability

(Dusek et al., 2006) of aerosol nanoparticles, indicating the importance of particle size when assessing the climate effect. Hence, accurate particle classification is essential when investigating the size dependent behavior of aerosol particles.

At present, particles are generally classified by either size or mass in atmospheric aerosol studies. A differential mobility analyzer (DMA) is the most commonly used size classifier, which selects particles based on electrical mobility (Knutson and Whitby, 1975; Park et al., 2008; Stolzenburg and McMurry, 2008; Swietlicki et al., 2008; Wiedensohler et al., 2012). A particle mass analyzer (PMA) includes an aerosol particle mass analyzer (APM) and a centrifugal particle mass analyzer (CPMA), both of which classify particles based on their mass-to-charge ratio (Ehara et al., 1996; Olfert and Collings, 2005). The charge distribution of particles must be known by passing through a neutralizer or similar when classified by DMA or PMA. However, particles with higher-order charges and identical apparent mobility or mass-to-charge ratio can be selected simultaneously, which are referred to as the multiple charging effect. This may introduce uncertainty in the subsequent characterization. Radney et al. (2013) demonstrated that although single-charged particles account for the highest number fraction (46.3%) of DMA-classified particles (200 nm), their contributions to the total mass concentration and extinction are insignificant (10.8% and 7.96%, respectively). Thus, the reported extinction of particles with a certain diameter has been greatly overestimated due to the multiple charging effect.

Previous studies (Shiraiwa et al., 2010;Rissler et al., 2013; Johnson et al., 2014; Johnson et al., 2021) tried to utilize the combination of size and mass classifiers, such as DMA-APM or DMA-CPMA systems, to obtain singly charged particles. Theoretically, the ability of a DMA-APM to eliminate multiply charged particles is governed by the particle morphology and setups of the DMA and APM (Kuwata, 2015). This conclusion implies that multiply charged particles cannot be effectively excluded for aspherical particles, especially for soot particles. Radney and Zangmeister (2016) investigated the limitations of a DMA-APM with three types of particles (polystyrene latex (PSL) spheres, ammonium sulfate (AS) and soot particles). Their results demonstrated that a DMA-APM can resolve multiply charged particles for spherical particles (PSL and AS particles), but it failed for aspherical soot particles. Multiply charged soot particles led to over 110% errors in retrieving the mass specific extinction cross section.

In contrast to DMA and PMA, an aerodynamic aerosol classifier (AAC) is a novel instrument that selects the aerodynamic equivalent diameter of aerosol particles based on their relaxation time. The advantage of utilizing an AAC is that the charge state of the particles does not need to be known in particle classification compared with the aforementioned classifiers; hence, multiple charging effects can be avoided (Tavakoli and Olfert, 2013). However, the selected particles are not monodispersed in mobility diameter when an AAC is used to select aspherical particles (Kazemimanesh et al., 2022).

Morphology information, such as effective density ($\rho_{eff}$), mass–mobility exponent ($D_{fm}$) and dynamic shape factor ($\chi$), can be inferred using tandem DMA-PMA system (Park et al., 2003; Zhang et al., 2008; Rissler et al., 2013; Pei et al., 2018; Zangmeister et al., 2018), DMA-AAC (Tavakoli and Olfert, 2014) and AAC-CPMA systems (Kazemimanesh et al., 2022). The derived $\rho_{eff}$ and $\chi$ depend upon the combination of

instruments used, while the nonphysical values of $\chi$ and $\rho_{\text{eff}}$ for aspherical particles can be determined by the
AAC-APM (Yao et al., 2020) and AAC-CPMA (Kazemimanesh et al., 2022).
The theoretical transfer functions of individual classifiers (DMA, CPMA and AAC) and the DMA-APM
system have been previously discussed (Knutson and Whitby, 1975; Ehara et al., 1996; Olfert and Collings,
2005; Stolzenburg and McMurry, 2008; Tavakoli and Olfert, 2013). In this study, we focus on a DMA-
CPMA and DMA-AAC in static configurations to eliminate multiply charged particles. The DMA-CPMA
and DMA-AAC systems mentioned below refer to the tandems of a DMA and CPMA or a DMA and AAC
in a static configuration, respectively. We calculate the transfer functions of the DMA-AAC and DMA-
CPMA systematically. Combined with soot experiments, we demonstrate that multiple charging effects may
still exist after DMA-CPMA classification when selecting aspherical particles and evaluate the light
absorption of selected particles with different charging states using Mie theory. Furthermore, we propose
operating conditions for the DMA-CPMA and DMA-AAC to eliminate multiply charged particles in future
studies. Our results suggest that the size- and mass-resolved optical properties may be overestimated for
small soot particles when using the DMA-CPMA system, which will lower the prediction accuracy of the
fresh soot climate effect. In Sect. 3.1, we calculate the transfer functions of the DMA-CPMA and DMA-
AAC utilizing the literature data of soot particles from Pei et al. (2018). In Sect. 3.2, we measure the multiple
charging effect of the DMA-CPMA using laboratory-generated soot particles, and the bias of optical
measurement induced by multiply charged particles is evaluated in Sect. 3.3.
**2 Theory and experiment**
**2.1 Transfer function for individual aerosol classifiers**
**DMA**
The DMA, consisting of two coaxial electrodes, classifies particles based upon electrical mobility $Z_{\text{p}}$
(Knutson and Whitby, 1975), which can be calculated as follows:
$$Z_{\text{p}} = qB = \frac{neCc(d_{\text{m}})}{3\pi\mu d_{\text{m}}}, \tag{1}$$
where $q$ is the particle charge, $n$ is the number of elementary charges, $B$ is the mobility of the particle, $e$ is
the elementary charge, $\mu$ is the viscosity of air, and $Cc(d_{\text{p}})$ is the Cunningham slip correction factor. When
the aerosol inlet flow rate equals the aerosol sampling outlet flow rate, the centroid mobility, $Z_{\text{p}}^*$, selected by
the DMA is defined as
$$Z_{\text{p}}^* = \frac{Q_{\text{sh}}}{2\pi V_{\text{DMA}} L_{\text{DMA}}} \ln\left(\frac{r_{2\_\text{DMA}}}{r_{1\_\text{DMA}}}\right), \tag{2}$$
where $Q_{\text{sh}}$ is the sheath flow rate, $V_{\text{DMA}}$ is the voltage between the two electrodes, $L_{\text{DMA}}$ is the length of the
DMA, and $r_{1\_\text{DMA}}$ and $r_{2\_\text{DMA}}$ are the inner and outer radii of the DMA electrodes, respectively. Assuming
that the aerosol inlet and aerosol sampling flow rates are equal, the transfer function of the DMA can be
expressed as follows when particle diffusion is negligible (Knutson and Whitby, 1975; Stolzenburg and
McMurry, 2008):
$\quad\Omega(\tilde{Z}_\text{p}, \beta_\text{DMA}) = \frac{1}{2\beta_\text{DMA}}\left[\left|\tilde{Z}_\text{p} - (1 + \beta_\text{DMA})\right| + \left|\tilde{Z}_\text{p} - (1 - \beta_\text{DMA})\right| - 2\left|\tilde{Z}_\text{p} - 1\right|\right]$, $\hspace{2cm}$ (3)
$\quad$where, $\tilde{Z}_\text{p} = Z_\text{p}/Z_\text{p}^*$, $\beta_\text{DMA} = Q_\text{a}/Q_\text{sh}$, and $Q_\text{a}$ is the sample flow rate. The limiting electrical mobilities that DMA
$\quad$can select are $(1 \pm \beta_\text{DMA}) \cdot Z_\text{p}^*$. The maximum and minimum values of $d_\text{m}$ for particles with n charges can be
$\quad$derived combining $(1 \pm \beta_\text{DMA}) \cdot Z_\text{p}^*$ and Eq. (1), and denote as $d_\text{mn,max}$ and $d_\text{mn,min}$, respectively. The transfer
$\quad$function is an isosceles triangle with value of 1 at $Z_\text{p}^*$ and going to 0 at $(1 \pm \beta_\text{DMA}) \cdot Z_\text{p}^*$. It translates to
$\quad$asymmetry in $d_\text{m}$ since the relationship between $d_\text{m}$ and $Z_\text{p}$ is nonlinear.
$\quad$**CPMA**
$\quad$The APM consists of two coaxial electrodes which are rotating at an equal angular velocity and a voltage is
$\quad$applied between these electrodes to create an electrostatic field (Ehara et al., 1996). The construction of the
$\quad$CPMA is similar to the APM, but its inner cylinder rotates faster than the outer cylinder to create a stable
$\quad$system of forces (Olfert and Collings, 2005). In the CPMA, the equation of particle motion is expressed as
$\quad\dfrac{m}{\tau}\dfrac{dr}{dt} = \dfrac{mv_\theta(r)^2}{r} - \dfrac{qV_\text{CPMA}}{r\ln\left(\frac{r_\text{2\_CPMA}}{r_\text{1\_CPMA}}\right)}$, $\hspace{4cm}$ (4)
$\quad$and the trajectory equation is
$\quad\dfrac{dr}{dz} = \dfrac{dr}{dt}\left(\dfrac{dz}{dt}\right)^{-1} = \dfrac{c_r}{v_z}$, $\hspace{5cm}$ (5)
$\quad$where $\tau$ is the relaxation time, $m$ is the mass of the particle, $t$ is time, $V$ is the voltage difference between the
$\quad$two electrodes, and $r_\text{1\_CPMA}$ and $r_\text{2\_CPMA}$ are the radii of the inner and outer electrodes, respectively. $c_r$ is the
$\quad$particle migration velocity, $v_z$ is the axial flow distribution and $v_\theta$ is the velocity profile in the angular
$\quad$direction,
$\quad v_\theta = \omega_1 \dfrac{\hat{r}^2 - \hat{\omega}}{\hat{r}^2 - 1}r + \omega_1 r_\text{1\_CPMA}^2 \dfrac{\hat{\omega}-1}{\hat{r}^2-1}\dfrac{1}{r} = \alpha r + \dfrac{\beta}{r}$, $\hspace{3cm}$ (6)
$\quad$where $\hat{\omega} = \omega_2/\omega_1$ is the ratio of the rotational speed of the outer electrode to the inner electrode and $\omega_1$ and
$\quad\omega_2$ are the rotational speeds of the inner and outer electrodes, respectively. $\hat{r}$ is the ratio of the inner and outer
$\quad$radii. $\alpha$ and $\beta$ are the azimuthal flow velocity distribution parameters.
$\quad$Sipkens et al. (2019) presented methods to calculate the transfer function of the CPMA. They considered the
$\quad$Taylor series expansion about the center of the gap ($r_\text{c}=(r_\text{2\_CPMA}+r_\text{1\_CPMA})/2$) instead of the equilibrium radius
$\quad$to avoid problems with the scenario in which the equilibrium radius does not exist. This method is much
$\quad$simpler and more robust. In this case, the particle migration velocity in the radial direction is
$\quad c_\text{r} \approx C_3 + C_4(r - r_\text{c})$, $\hspace{6cm}$ (7)
$\quad$where
$\quad C_3 = \tau\left(\alpha^2 r_\text{c} + \dfrac{2\alpha\beta}{r_\text{c}} + \dfrac{\beta^2}{r_\text{c}^3} - \dfrac{C_0}{mr_\text{c}}\right)$, $\hspace{4cm}$ (8)
$\quad C_4 = \tau\left(\alpha^2 - \dfrac{2\alpha\beta}{r_\text{c}} - \dfrac{3\beta^2}{r_\text{c}^4} + \dfrac{C_0}{mr_\text{c}^2}\right)$, $\hspace{4cm}$ (9)
$\quad C_0 = \dfrac{qV_\text{CPMA}}{\ln(r_\text{2\_CPMA}/r_\text{1\_CPMA})}$, $\hspace{5cm}$ (10)
$\quad$Assuming plug flow, the transfer function would be
$\quad\Omega = \dfrac{r_b - r_a}{2\delta}$, $\hspace{6cm}$ (11)
where $\delta=(r_{2\_CPMA}-r_{1\_CPMA})/2$ is the half width of the gap between the two electrodes, and
$$r_a = \min\left\{r_{2\_CPMA}, \max\{r_{1\_CPMA}, G_0(r_{1\_CPMA})\}\right\}, \tag{12}$$
$$r_b = \min\left\{r_{2\_CPMA}, \max\{r_{1\_CPMA}, G_0(r_{2\_CPMA})\}\right\}, \tag{13}$$
$$G_0(r_L) = r_c + \left(r_L - r_c + \frac{C_3}{C_4}\right)\exp(-C_4 L\bar{v}) - \frac{C_3}{C_4}, \tag{14}$$
where $G_0(r)$ is the operator used to map the final radial position of the particle to its position at the inlet and
$\bar{v}$ is the average flow velocity. min{} and max{} are the minimum and maximum values of the quantities in
the brackets, respectively.
Reavell et al. (2011) calculated the resolution of the CPMA assuming that the gap between two electrodes is
narrow enough that the variation of force in the gap can be ignored. The mass resolution ($R_m$) of CPMA is
related to particles mobility. When selecting the particles with mass of $m_1$ and mobility of $B_1$, the $R_m$ can be
calculated by
$$R_m = \frac{2\pi B_1 L_{CPMA} r_c^2 \omega^2 m_1}{Q_{CPMA}}, \tag{15}$$
where $\omega$ is the equivalent rotational speed calculated by $\omega = \alpha + \frac{\beta}{r_c^2}$, $m_1$ is the nominal mass that the CPMA
can select, $Q_{CPMA}$ is the volumetric flow rate. The limiting mass can be calculated by
$$m_{n,min}^{n,max} = n \cdot m_1 \pm \frac{Q_{CPMA}}{2\pi B_{n,min}^{n,max} L_{CPMA} r_c^2 \omega^2} = n \cdot m_1 \pm \frac{m_1}{R_m} \cdot \frac{B_1}{B_{n,min}^{n,max}}, \tag{16}$$
where $m_{n,min}^{n,max}$ and $B_{n,min}^{n,max}$ are the maximum and minimum mass and corresponding mobility of particles
bearing number of elementary charges of n that the CPMA can select, respectively. Further details can be
found in Reavell et al. (2011) and Sipkens et al. (2019).
**AAC**
The AAC classifies particles based on relaxation time, which is defined by
$$\tau = Bm = \frac{Cc(d_{ae})\rho_0 d_{ae}^2}{18\mu}, \tag{17}$$
where $\mu$ is the viscosity of air. $Cc(d_{ae})$ is the slip correction factor. $\rho_0$ is the standard density with a value of 1
g/cm$^3$ (Johnson et al. 2018). When the aerosol inlet flow rate equals the aerosol sampling outlet flow rate,
the transfer function of the AAC can be expressed as (Tavakoli and Olfert, 2013)
$$\Omega = \frac{1}{2\beta_{AAC}}\left[|\tilde{\tau} - (1-\beta_{AAC})| + |\tilde{\tau} - (1+\beta_{AAC})| - 2|\tilde{\tau} - 1|\right], \tag{18}$$
$\tau^*$ is the nominal relaxation time, which is classified by the AAC,
$$\tau^* = \frac{2Q_{sh}}{\pi\omega^2(r_{1\_AAC}+r_{2\_AAC})^2 L}, \tag{19}$$
where $\beta_{AAC} = \frac{Q_a}{Q_{sh}}$, $\tilde{\tau} = \frac{\tau}{\tau^*}$, $r_{1\_AAC}$ and $r_{2\_AAC}$ are the inner and outer radii of the AAC, respectively. The
limiting $\tau$ that AAC can select are $(1 \pm \beta_{AAC}) \cdot \tau^*$. The maximum and minimum values of $d_{ae}$ can be derived
and denoted as $d_{ae,max}$ and $d_{ae,min}$, respectively.

## 2.2 Experimental setup

A schematic of the experimental setup is illustrated in Fig. 1. Soot particles were generated by a miniature inverted soot generator (Argonaut Scientific Ltd., Canada) with a propane flow of 74.8 SCPM (standard mL per minute, flow in mL min$^{-1}$ converted from ambient to T =298.15 K and P = 101.325 kPa) and an air flow rate of 12 SLPM (Standard L per minute, flow in L min$^{-1}$ converted from ambient to T =298.15 K and P = 101.325 kPa). Although this operation setting is not in the open-tip flame regime, the flame is open-tip consistent with Fig. 2d in Moallemi et al. (2019). Detailed aerosol generation methods can be found in Kazemimanesh et al. (2019b) and Moallemi et al. (2019). The polydispersed aerosols were dried to a relative humidity of <20% by a silica dryer and then passed through a soft X-ray neutralizer (Model 3088, TSI, Inc., USA). Five mobility diameters (80 nm, 100 nm, 150 nm, 200 nm and 250 nm) of soot particles were selected with the DMA (Model 3081, TSI Inc., USA, $\beta_{DMA}$ = 10). For the soot characterization, the mobility-selected aerosol flow was switched between two parallel lines and fed into the CPMA (Cambustion Ltd., UK) and AAC (Cambustion, Ltd., UK, $\beta_{AAC}$ = 10); meanwhile, the condensation particle counter (CPC, Model 3756, TSI, Inc., USA, 0.3 L min$^{-1}$) was switched between the CPMA and AAC. The distributions of particle number concentration as a function of particle mass ($m$) and aerodynamic diameter ($d_{ae}$) were measured by the scanning mode of the CPMA and AAC, respectively, while the CPC recorded their corresponding number concentrations at each setpoint. For each $d_m$, the $m$ and $d_{ae}$ distributions were measured three times. Between measurements of each $d_m$, the CPC was used behind the DMA, and the number size distribution of the generated soot particles was measured by a scanning mobility particle sizer (SMPS) to ensure the number size distribution of generated soot particles did not change during the whole experiment. The $m$ and $d_{ae}$ distributions were fitted to log-normal distributions; thus, the modal values denoted as $m_c$ and $d_{ae,c}$ for the mobility-selected particles were determined. The equation of log-normal distribution used in this study is expressed as

$$\begin{cases} N(m) = \frac{N_0}{\sqrt{2\pi}\ln\sigma_m}\exp(\frac{-(\log(m)-\log(m_c))^2}{2(\ln\sigma_m)^2}) \\ N(d_{ae}) = \frac{N_0}{\sqrt{2\pi}\ln\sigma_{ae}}\exp(\frac{-(\log(d_{ae})-\log(d_{ae,c}))^2}{2(\ln\sigma_{ae})^2}) \end{cases}, \tag{20}$$

where $\sigma_m$ and $\sigma_{ae}$ are the geometric standard deviations of $m$ and $d_{ae}$ distributions, respectively. $m_c$ and $d_{ae,c}$ are the geometric mean of $m$ and $d_{ae}$, respectively.

The CPMA and AAC were calibrated with certified PSL spheres (Thermo, USA) with sizes of 70 nm, 150 nm and 303 nm before the measurement. The measured $m$ and $d_{ae}$ were compared to $m_{PSL}$ and $d_{ae,PSL}$, which were calculated with the nominal diameter and density of PSL (1050 kg m$^{-3}$). The deviations between measured $m$ and $m_{PSL}$ or measured $d_{ae}$ and $d_{ae,PSL}$ were 2.75% and 5.14%, respectively. To quantify the multiple charging effect of particles selected by the DMA-CPMA system, the soot particles were initially selected by the DMA-CPMA at different $d_m$ and the corresponding $m$. Then, the $d_{ae}$ distribution of mobility and mass selected particles was obtained by stepping the AAC rotation speed of the cylinder with simultaneous measurement of the particle concentration at the AAC outlet using a CPC (Fig. 1b).

## 3 Results and discussion

### 3.1 Transfer function of the tandem system

The DMA, PMA and AAC select particles based on the electrical mobility diameter, mass and aerodynamic diameter, respectively. These properties can be connected as follows (Decarlo et al. 2004):

$$\frac{Cc(d_{ae})\rho_0 d_{ae}^2}{6} = \frac{Cc(d_m)\rho_{eff}d_m^2}{6} = m\frac{Cc(d_m)}{\pi d_m} , \tag{21}$$

where $\rho_{eff} = \frac{6m}{\pi d_m^3}$. The transfer function of the DMA-APM has been well documented and can be found in Kuwata (2015). The convolution of the transfer functions of the DMA-CPMA and DMA-AAC were calculated by the following equations.

$$\Phi_{DMA-CPMA} = \Omega_{CPMA}\Omega_{DMA} , \tag{22}$$

$$\Phi_{DMA-AAC} = \Omega_{DMA}\Omega_{AAC} , \tag{23}$$

where $\Phi$ and $\Omega$ are the transfer functions of the combined and individual classification systems expressed by subscripts, respectively. In the following discussion, we explain the transfer functions of the DMA-CPMA and DMA-AAC utilizing the literature data of soot particles (Pei et al., 2018). The $d_m$ and $m$ of the representative particles are 100 nm and 0.33 fg, respectively, and the corresponding $d_{ae}$ is 68.3 nm according to Eq. (21). In the calculation, the following parameter set was employed: $d_m = 100$ nm, $Q_{DMA} = 0.3$ L min$^{-1}$, $\beta_{DMA} = 0.1$, $m = 0.33$ fg, $Q_{CPMA}=0.3$ L min$^{-1}$, $R_m = 8$, $d_{ae} = 68.3$ nm, $Q_{AAC} = 0.3$ L min$^{-1}$, $\beta_{AAC} = 0.1$. The transfer functions of DMA-CPMA and DMA-AAC were solved iteratively using logarithmically spaced $d_m$, $m$ and $d_{ae}$, which included 600 points each. The ranges of $d_m$, $m$ and $d_{ae}$ used in the calculations were from 0.8 times $d_{m1,min}$ to 1.2 times $d_{m2,max}$, and from 0.8 times $m_{1,min}$ to 1.2 times $m_{2,max}$, from 0.8 times $d_{ae,min}$ to 1.2 times $d_{ae,max}$, respectively. The dimensions of the individual classifiers are summarized in Table 1.

**DMA-CPMA**

The DMA-CPMA transfer function ($\Phi_{DMA-CPMA}$) for particles mentioned above, i.e., particles with $d_m$ of 100 nm and $m$ of 0.33 fg, is calculated in $\log(d_m)$-$\log(m)$ space, as shown in Fig. 2. The particles are shown in Fig. 2 in actual $d_m$ and $m$, but when we calculate the resolution of DMA and CPMA, the mobility and effective mass are used. The resolution of CPMA can be calculated by Eq. (15), where $m_1$ is the mass of singly charged particles which can be selected by the CPMA, i.e., effective mass. In $\log(d_m)$-$\log(m)$ space, the mass–mobility relationship is

$$(m/fg) = k_f(d_m/nm)^{D_{fm}} , \tag{24}$$

$$\log(m/fg) = D_{fm}\log(d_m/nm) + \log(k_f) , \tag{25}$$

In general, $D_{fm}$ equals 3 for spherical particles and smaller than 3 for aspherical particles, although $D_{fm}$ can be larger than 3 for particles that are non-spherical at small $d_m$ and approach spherical as $d_m$ increases. In the $\log(d_m)$-$\log(m)$ space, the relationship of $m$ and $d_m$ is linear, with the slope expressed as the mass–mobility exponent ($D_{fm}$) and the intercept representing the pre-exponential factor ($k_f$). Under this specific operation condition, no overlap was observed between the spherical particle population (black line) and the classification region (the colored blocks) for doubly charged particles, implying that only the singly charged

particles were selected. For aspherical particles with $D_{fm} < 3$, such as soot particles with aggregate structures,
the particle population may overlap the doubly charged region when the slope ($D_{fm}$) is small enough; however,
the combination of DMA and CPMA is generally used to avoid the multiple charge effect in soot studies.
The reported $D_{fm}$ values are typically in the range of 2.2–2.4 for fresh soot particles (Rissler et al., 2013) and
diesel soot particles (Park et al., 2003). In the exemplary case (Pei et al., 2018), the derived $D_{fm}$ of premixed
flame-generated soot particles was 2.28, resulting in the particles population always going through the
transfer area of doubly charged particles. This implies that the performance of the DMA-CPMA to eliminate
multiply charged particles to a certain extent depends on the particle morphology.
The DMA-CPMA system can eliminate the multiply charged particles only if the $D_{fm}$ of the particles is larger
than the slope of a line connecting $(d_m, m) = (d_{m2,min}, m_{2,max})(d_{m1}, m_1)$ (as PP$_0$ shown in Fig. 2). Since the
CPMA is used downstream of the DMA, $m_{2,max}$ at the $d_m$ of $d_{m2,min}$ can be calculated using Eq. (16) with the
known mobility. Accordingly, the ideal condition under static operation to completely eliminate the multiply
charged particles is
$$D_{fm} > PP_0 = \frac{\log(m_{2,max}/m_1)}{\log(d_{m2,min}/d_{m1})} = \frac{\log\left(2+\frac{2}{R_m(1+\beta_{DMA})}\right)}{\log\left(\frac{2}{(1+\beta_{DMA})}\frac{Cc(d_{m2,min})}{Cc(d_{m1})}\right)} \tag{26}$$

The ability of the DMA-CPMA to eliminate multiply charged particles depends on the selected $d_m$, $m$ and
resolutions of both the DMA and CPMA. Combining Eq. (15), equation (26) gives instructions in actual
operation to eliminate multiply charged particles. When selecting particles of certain $d_m$ and $m$, by decreasing
$Q_{CPMA}$, or increasing $\omega$ and $\beta_{DMA}$, i.e., by increasing the resolution of the measurement, the potential of
multiply charged particles is reduced. Thus, the key to evaluating whether there is a multiple charging effect
lies in the particle morphology ($D_{fm}$) and the slope of PP$_0$ calculated from Eq. (26) theoretically.
In addition to the instrument setup, the particle morphology is also crucial for the DMA-CPMA. Here, we
simulate the critical slope of PP$_0$ when selecting different $d_m$ and $m$ under the common selecting conditions
($\beta_{DMA} = 0.1$, $Q_{CPMA}=0.3$ L min$^{-1}$, $R_m = 8$) using Eq. (26), which is represented as contour lines in Fig. 3 (A
black and white version is shown as Fig. S4). Under these selection conditions, the DMA-CPMA can select
monodispersed particles when the $D_{fm}$ of the particles is larger than the critical slope of PP$_0$. When selecting
small aspherical particles or particles with extremely low density, the critical slope of PP$_0$ is relatively higher,
and the DMA-CPMA classification is sensitive to multiple charging effect. As shown in Fig. 3, $d_m$, $m$ and
the corresponding $D_{fm}$ were taken from the literature (Park et al., 2003; Rissler et al., 2013; Tavakoli et al.,
2014; Ait Ali Yahia et al., 2017; Dastanpour et al., 2017; Forestieri et al., 2018; Pei et al., 2018;
Kazemimanesh et al., 2019a). Generally, for soot particles with $D_{fm}$ of 2.2-2.4, the multiple charging effect
can be avoided for the DMA-CPMA when selecting soot particles with mobility diameters larger than 200
nm, while it fails to eliminate multiply charged particles when selecting small soot particles, as shown by the
circles and squares in Fig. 3. These potential uncertainties are discussed in detail with flame-generated soot
particles in Sect. 3.2.
**DMA-AAC**
The advantage of the AAC versus the CPMA is that there is no need for a neutralizer to charge aerosol
particles to a known charge state. Measuring solely with an AAC will avoid multiple charging. However,
aspherical particles with different mass can be selected by the AAC as having identical aerodynamic diameter
(Kazemimanesh et al., 2022). According to Eq. (21), the population selected by AAC has one physical size
($d_{ae}$) but the $d_m$ range of this population is wide since soot particles have different densities. Multiple charging
becomes a problem when the tandem measurement is made with a DMA or PMA. According to Eq. (21) and
Eq. (24), the relationship of $d_{ae}$ and $d_m$ of aspherical particles can be expressed as follows:
$$\log(d_{ae}/\text{nm}) = \frac{1}{2}(D_{fm} - 1)\log(d_m/\text{nm}) + \frac{1}{2}\log\left(\frac{6}{\pi}\frac{Cc(d_m)k_f}{Cc(d_{ae})\rho_0} \cdot 10^9\right),$$    (27)
which indicates that the relationship between $d_{ae}$ and $d_m$ is nonlinear since $Cc(d_m)$ and $Cc(d_{ae})$ vary with $d_m$
and $d_{ae}$, respectively. Particle morphology can be derived from the relationship between $d_m$ and $d_{ae}$ measured
by a DMA and AAC, respectively. To simulate the transfer function of the DMA-AAC, the same particles
($d_m = 100$ nm, $m = 0.33$ fg, $D_{fm} = 2.28$) as those used in the calculations of the DMA-CPMA were selected.
The corresponding $d_{ae}$ was numerically solved using the known mass-mobility relationship. The transfer
function of the DMA-AAC is shown in $\log(d_{ae})$-$\log(d_m)$ (Fig. 4a). In the transfer function of DMA-CPMA,
the classification regions of singly charged particles and doubly charged particles are on the diagonal. The
oblique line of particles population is more likely to go through the region of doubly charged particles in the
transfer function of DMA-CPMA. The transfer functions of singly charged and doubly charged particles are
in parallel for the DMA-AAC, suggesting that the particles population is less likely to overlap with the region
of multiply charged particles. Using the example setups ($d_m = 100$ nm, $Q_{DMA} = 0.3$ L min$^{-1}$, $\beta_{DMA} = 0.1$, $d_{ae} =$
68.3 nm, $Q_{AAC} = 0.3$ L min$^{-1}$, $\beta_{AAC} = 0.1$) of the DMA-AAC, truly monodispersed particles are selected for
spherical particles and typical soot particles.
Similar to the DMA-CPMA system, eliminating multiply charged particles requires that the $d_{ae,max}$ of the
AAC at $d_{m2,min}$ must be smaller than the $d_{ae}$ of particles of interest, which can be derived from $d_{m2,min}$ and $D_{fm}$
(Eq. (27)),
$$d_{ae}(d_{m2,min}, D_{fm}) > d_{ae,max}(d_{m2,min}),$$
$$\Rightarrow D_{fm} > \frac{\log(2 \cdot \frac{1+\beta_{AAC}}{1+\beta_{DMA}})}{\log[\frac{2}{1+\beta_{DMA}} \cdot \frac{Cc(d_{m2,min})}{Cc(d_{m1})}]},$$    (28)
This equation describes the minimum value of $D_{fm}$ to eliminate the multiple charging effect. It is clearly
shown that the mobility resolution of the DMA and the relaxation time resolution of the AAC determine the
limiting condition, and the resolution of the AAC is more important compared with the resolution of the
DMA. The limiting condition is also related to the selected $d_m$ of the DMA but independent of the selected
$d_{ae}$ of the AAC (Fig. S1). Setting the same resolutions for the DMA and AAC, particle selection is more
susceptible to multiple charging effects when selecting small sizes. In Fig. 4a, the values of $\beta_{DMA}$ and $\beta_{AAC}$
are 0.1, resulting in a minimum $D_{fm}$ of 1.41. This $D_{fm}$ is smaller than that for most aerosols. Hence, the
selected particles of the DMA-AAC are truly monodisperse regardless of the particle morphology. However,
in actual operations, a larger sample flow rate may be required to satisfy the apparatus downstream, while
the maximum sheath flow rate of the classifier is restricted by the instrument design (e.g., 30 L min$^{-1}$ for the
DMA and 15 L min$^{-1}$ for the AAC). In addition, the maximum size ranges are also restricted by the sheath
flow, so in some cases, a lower sheath flow rate is required to select larger particles. When increasing $\beta_{AAC}$
to 0.3 (decreasing the resolution of AAC) and leaving $\beta_{DMA}$ unchanged, the transfer function becomes broader
(Fig. 4b). The minimum $D_{fm}$ is 2.44, which indicates that the multiple charging effect exists for typical soot
particles with $D_{fm}$ of 2.2-2.4. The line representing soot particles overlaps with the region of doubly charged
particles. Thus, reducing the resolutions of the DMA or AAC is not suggested in actual operations.
We think the transfer functions of DMA-AAC or AAC-DMA are identical regardless of the order of DMA
and AAC. For example, we use AAC-DMA to select particles with $d_{ae}$ of 68 nm and $d_m$ of 100 nm. In Fig.
4a, the transfer function of AAC is the region between the horizontal lines of $d_{ae,max}$ (75 nm) and $d_{ae,min}$ (63
nm). The soot particles population (red line) goes through this region will be selected by AAC. The mobility
diameter distribution of these relaxation time selected particles is around 80 nm to 120 nm. Then the DMA
is fixed to select particles with $d_m$ of 100 nm, the particles with double charges and the same mobility ($d_m$ of
150 nm) have been excluded by AAC. As a result, AAC-DMA select monodispersed particles with $d_{ae}$ of
68.3 nm and $d_m$ of 100 nm. In Fig. 4b, the resolution of AAC is lower and transfer function of AAC is broader
than that in Fig. 4a. The soot particles population (red line) goes through the transfer function region between
the horizontal lines at $d_{ae}$ of $d_{ae,max}$ (50 nm) and $d_{ae,min}$ (86 nm). The mobility diameter distribution of these
relaxation time selected particles is very wide from less than 80 nm to about 158 nm. Then these relaxation
time selected particles were charged and selected by DMA at $d_m$ of 100 nm, singly charged particles with $d_m$
of 95 nm~106 nm and doubly charged particles with $d_m$ of 142 nm~158 nm will be selected.
If we use the DMA-AAC, the particles are selected by DMA first. For example, in Fig. 4b, the transfer
function of DMA is shown as two vertical regions which particles with single and double charges can
penetrate. The soot particles (red line) goes through it and two populations of soot particles with mode $d_m$ of
100 nm and 150 nm will be selected. The corresponding $d_{ae}$ distributions of these singly and doubly charged
particles are 66 nm~70 nm and 81 nm~87 nm. These mobility-selected particles are selected at $d_{ae}$ of 68.3
nm by AAC and the transfer function of AAC shows that particles with $d_{ae}$ of 50 nm~86 nm can penetrate.
As a result, singly charged particles with $d_{ae}$ of 66 nm~70 nm and doubly charged particles with $d_{ae}$ of 81 nm
~86 nm can be selected.
As a summary, the transfer functions of DMA-AAC and AAC-DMA in a static configuration are the same
no matter the ordering of DMA and AAC.

## 3.2 Evaluation of the multiple charging effect

To quantify the possible biases of the multiple charging effect in the DMA-CPMA system, we conducted a
soot experiment, as demonstrated in Fig. 1. For each mobility-selected particles, the distributions of number
density as a function of $d_{ae}$ and $m$ were determined by the scans. These distributions were then fit to a log-
normal to determine the modal values ($d_{ae,c}$, $m_c$) and from these values the $\rho_{eff}$ were determined.. The
uncertainties of $d_{ae,c}$ and $m_c$ were standard deviation of multiple measurements. Representative plots for the
measured distributions of $m$ and $d_{ae}$ of particles with $d_m$ of 150 nm and 250 nm are shown in Fig. S2. The
results are summarized in Table 2. The fitted values of $D_{fm}$ and $k_f$ were 2.28 and $7.49 \times 10^{-6}$ fg, respectively,
indicating a fractal structure, which is the same as in previous studies (Pei et al., 2018). The effective densities
of generated soot particles vary from >500 kg m$^{-3}$ at $d_m = 80$ nm to <300 kg m$^{-3}$ at $d_m$ of 250 nm determined
by DMA-CPMA and DMA-AAC. In general, the deviation of values of $\rho_{eff}$ measured by DMA-CPMA and
DMA-AAC monotonically decreases with increasing particle size. The deviation is 7.65% for particles of 80
nm, whereas it decreased to <1% for particles larger than 200 nm. The results reveal a strict agreement
between the two methods for retrieving the particle effective density.
According to Fig. 3, the critical slopes of PP$_0$ for soot particles with $d_m$ of 80 nm, 100 nm, 150 nm, 200 nm
and 250 nm are 2.46, 2.41, 2.29, 2.17 and 2.08, respectively. The measured $D_{fm}$ of 2.28 is smaller than the
calculated PP$_0$ for particles with $d_m$ smaller than 200 nm, which suggests that the contributions from the
multiply charged particles cannot be eliminated.
When selecting particles with $d_m$ of 80 nm and $m$ of 0.16 fg, the corresponding DMA-CPMA transfer function
is shown in Fig. 5a. DMA-CPMA is set to select singly charged particles with $d_m$ of 80 nm and $m$ of 0.16 fg,
while the doubly charged particles with $d_m$ of 119.3 nm and $m$ of 0.32 fg will also be selected and the transfer
function is presented as upper right region. Soot particles curve (red line) goes through the upper- right region
which doubly charged particle can penetrate ($d_m$ of 113 nm~118 nm, $m$ of 0.35 fg~0.39 fg). As a result, we
conclude that multiple charging effect still exists when DMA-CPMA select soot particles with $d_m$ of 80 nm
and $m$ of 0.16 fg. Since the classification of the AAC is different from the DMA and CPMA, the aerodynamic
size distributions of mobility- and mass- selected particles were characterized. Fig. 5b shows the particles
number density aerodynamic size distribution (PNSD$_{ae}$) scanned by the AAC. For each measurement,
PNSD$_{ae}$ was fitted using log-normal distributions, and three peaks corresponding to singly, doubly and triply
charged particles were identified. The fractional number concentration of particles with different charging
state is expressed as follows,
$$f_{N,n} = \frac{\int_{d_{ae,low}}^{d_{ae,high}} \frac{dN_n}{d\log(d_{ae})} d\log(d_{ae})}{\sum_{n=1}^{3} \int_{d_{ae,low}}^{d_{ae,high}} \frac{dN_n}{d\log(d_{ae})} d\log(d_{ae})} ,$$    (29)
where $f_{N,n}$ and $N_n$ are the fractional number concentration and number concentration of particles bearing n
charges. $d_{ae,low}$ and $d_{ae,high}$ denote the minimum and maximum values of $d_{ae}$ scanned by AAC, respectively.
The uncertainties are standard deviations of multiple measurements. Some small particles remaining in the
AAC induced the peak at $d_{ae}$ <40 nm. These residual particles were measured even if the sample flow was
filtered. For particles with $d_m = 80$ nm, the modal $d_{ae}$ values were 53.9 nm, 60.6 nm and 70.9 nm, and the
corresponding $d_{ae}$ values were calculated as 51.5 nm, 62.0 nm and 70.7 nm using Eq. (1) and Eq. (17). The
experimental results are consistent with the theoretical results with deviations within 5.3%.
When selecting particles with $d_m$ of 200 nm and $m$ of 1.28 fg, the transfer function is shown in Fig. 6a. The
PP$_0$ slope of 2.17 is smaller than that $D_{fm}$ of 2.28, and the generated particles population does not overlap
with the block of doubly charged particles; thus, the DMA-CPMA classified particles were truly
monodispersed. PNSD$_{ae}$ measured by the AAC is unimodal, implying that the classified particles were singly
charged (Fig. 6b).
The results of other experiments are shown in Fig. S3. Although the critical slope of PP$_0$ when selecting 150
nm particles is close to $D_{fm}$ and the transfer function of DMA-CPMA also showed that negligible multiply
charged particles would be selected (Fig. S3d), doubly charged particles were measured in PNSD$_{ae}$ (Fig. S3e).
These doubly charged particles were selected, probably owing to particle diffusion. The nondiffusion models
were used to calculate the transfer function, but the transfer function can be broader because of diffusion. In
summary, for a type of particle with the same mass–mobility relationship, the possibility of multiple charging
increases for small particles when selected by the DMA-CPMA system, which is consistent with the
theoretical calculation in Sect. 3.1.
**3.3 Atmospheric implication**
The DMA-APM and DMA-CPMA systems are usually adopted to eliminate multiply charged particles in
soot aerosol studies. Although they might fail to select monodispersed particles, downstream measurements
by instruments such as a single-particle soot photometer (SP2) will not be interfered with, which characterizes
the distinct information of a single particle. Nevertheless, for techniques measuring the properties of an entire
aerosol population, e.g., scattering coefficient by a nephelometer or absorption coefficient by a photoacoustic
spectrometer, multiply charged particles can induce significant bias. A previous study (Radney and
Zangmeister, 2016) noted that the DMA-APM failed to resolve multiply charged particles for soot particles
when selecting 150 nm flame-generated particles, which caused a 110% error in extinction measurement. To
investigate the multiple charging effect for DMA-CPMA classification, the optical absorption coefficient of
particles with different charging states after DMA-CPMA classification was calculated from PNSD$_{ae}$. Mie
theory was used to calculate the theoretical absorption coefficient at a wavelength of 550 nm. Mie theory is
probably not the "best" method to use here since soot particles are aspherical agglomerates. Realistically,
however, the Mie comparison is only being used to prove a point about the impact of multiple charging.
Therefore, in this instance, any errors in the calculated optical properties are somewhat inconsequential. The
refractive index used in the Mie code was 1.95+0.79i (Bond and Bergstrom, 2006). The PNSD$_{ae}$ for different
charging state particles was converted to volume-equivalent diameter size distributions (PNSD$_{ve}$), which was
used in Mie theory to determine the absorption coefficient. The method to calculate PNSD$_{ve}$ is described in
Sect. S1. Subsequently, the absorption coefficient, $\alpha_{abs}$, was derived using Mie theory and the PNSD$_{ve}$ of
particles with different charging states. The fractional absorption coefficient for particles with different
charging state is calculated as follows,
$$f_{abs,n} = \frac{\int_{d_{ve,low,n}}^{d_{ve,high,n}} \frac{d\alpha_{abs,n}}{d\log(d_{ve})} d\log(d_{ve})}{\sum_{i=1}^{3} \int_{d_{ve,low,n}}^{d_{ve,high,n}} \frac{dN_n}{d\log(d_{ve})} d\log(d_{ve})} , \tag{30}$$

where $f_{abs,n}$ and $\alpha_{abs,n}$ are the fractional absorption coefficient and absorption coefficient of particles bearing
n charges, respectively. $d_{ve,low,n}$ and $d_{ve,high,n}$ denote the minimum and maximum value of $d_{ve}$ of particles with
n charges, which are converted from $d_{ae,low}$ and $d_{ae,high}$ scanned by AAC, respectively.
The overestimation of mass absorption cross-section (MAC) is calculated by
$$\frac{\Delta \text{MAC}}{\text{MAC}} = \frac{\frac{\alpha_{\text{abs,tot}}}{m_p N_{\text{tot}}} - \frac{f_{\text{abs,1}} \cdot \alpha_{\text{abs,tot}}}{m_p \cdot f_{\text{N,1}} \cdot N_{\text{tot}}}}{\frac{f_{\text{abs,1}} \cdot \alpha_{\text{abs,tot}}}{m_p \cdot f_{\text{N,1}} \cdot N_{\text{tot}}}} = \frac{f_{\text{N,1}}}{f_{\text{abs,1}}} - 1 \; , \tag{31}$$
where $\alpha_{\text{abs,tot}}$ and $N_{\text{tot}}$ is the total absorption coefficient and number concentration of particles selected by
DMA-CPMA, respectively. $m_p$ is the actual mass of singly charged particles selected by DMA-CPMA. The
uncertainties were calculated from propagation of errors. For soot particles with diameters <200 nm, the
optical absorption contributions of particles with different charging states and the MAC overestimation are
summarized in Table 3. For soot particles with a diameter of 80 nm, the contributions of particles with
different charging states are shown in Fig. 5c. Doubly charged particles only account for 26.7% ±3.0% of the
total number concentration but provide a large fractional contribution to the total absorption (45.7% ±4.2%).
Additionally, a small fraction (1.1% ±0.4%) of triply charged particles accounted for 3.7% ±1.5% of the
absorption. As a result, the MAC was overestimated by 42.7% ±9.1%, and the directive radiative force (DRF)
was overestimated by 42.7% ±9.1%. The DRF was calculated using previous global climate models (Bond et
al., 2016). For particles selected by the DMA-CPMA at a $d_m$ of 200 nm and an m of 1.28 fg, the selected
particles were truly dispersed, and the measured optical properties were valid (Fig. 6c).
A large amount of 70 nm -90 nm soot particles was emitted from diesel engine (Wierzbicka et al., 2014), and
neglecting the multiple charging effect in the measurement of mass-specific MAC on this size range will
result in significant bias in the estimation of radiative forcing of automobile-emitted soot particles, which
may lead to large errors in climate model.
According to Table 3, the number fraction of doubly charged particles declines with the size of the nominated
particles, i.e., 26.7% ±3.0% and 17.6% ±0.5% for 80 nm and 100 nm particles, respectively, but only 4.2% ±1.1%
for 150 nm particles. Accordingly, the MAC was largely overestimated for 80 nm and 100 nm particles
(42.7% ±9.1% and 28.0% ±1.8%, respectively) but moderately overestimated for 150 nm particles
(9.2% ±4.1%). To summarize, our results indicated that the combination of tandem classifiers is not sufficient
to completely eliminate multiply charged particles when selecting small flame-generated soot particles,
which introduced noticeable bias for absorption measurements and led to overestimation of the MAC. As a
result, the DRF of soot particles was also overestimated.
**4 Conclusion**
In this study, we demonstrate the transfer functions of DMA-CPMA and DMA-AAC and discuss their
limitations to eliminate multiply charged particles. For aspherical particles, there is no guarantee that the
multiple charging effect can be avoided in DMA-CPMA or DMA-AAC systems. Usually, a DMA-AAC can
select truly monodisperse particles, but the method can suffer from multiple charging when decreasing the
resolutions of the DMA and AAC. The ability of the DMA-CPMA to eliminate multiple charging effect
mainly depends on the particle morphology and the instrument resolutions. This tandem system is more
sensitive to multiple charging effect with decreasing $D_{\text{fm}}$ and decreasing nominal size of particles. The DMA-
CPMA failed to eliminate multiply charged particles when selecting soot particles with diameters < 150 nm.
Although doubly charged particles accounted for a small fraction of the number concentration, they
contributed most significantly to light absorption, which indicated that multiply charged particles can induce
an obvious contribution to light absorption and lead to an overestimation of DRF for flame-generated soot
particles.

*Code/Data availability*. Code/Data are available upon request.
*Author contributions*. ZW determined the main goal of this study. YS and XP designed the methods. YS
carried them out and prepared the paper with contributions from all coauthors. YS, HL and JZ analyzed the
optical data.
*Competing interests*. The authors declare that they have no conflicts of interest.
*Acknowledgments.* The study was supported by the National Natural Science Foundation of China (91844301
and 41805100). We especially acknowledge useful comments and suggestions on the MATLAB script of the
CPMA transfer function from Timothy A. Sipkens.
**Appendix A**
**A1. Nomenclature**

| Parameter | Definition |
|---|---|
| $B$ | Mechanical mobility |
| $C_c(d_p)$ | Cunningham slip correction factor |
| $c_r$ | Particle migration velocity |
| $d_{ae}$ | Aerodynamic equivalent diameter |
| $d_{ae,c}$ | the geometric mean of $d_{ae}$ distribution measured by AAC-CPC |
| $d_{ae,high}$ | The maximum value of $d_{ae}$ scanned by AAC |
| $d_{ae,low}$ | The minimum value of $d_{ae}$ scanned by AAC |
| $d_{ae,max}$ | The maximum $d_{ae}$ of particles that can be selected in AAC classification |
| $d_{ae,min}$ | The minimum $d_{ae}$ of particles that can be selected in AAC classification |
| $d_m$ | Mobility equivalent diameter |
| $d_{mn,max}$ | The maximum $d_m$ of particles with n charges that can be selected in DMA classification |
| $d_{mn,min}$ | The minimum $d_m$ of particles with n charges that can be selected in DMA classification |
| $d_{ve}$ | Volume-equivalence size |
| $D_{fm}$ | Mass-mobility exponent |
| $e$ | Elementary charge |

| | |
|---|---|
| $f_{N,n}$ | The fractional number concentration of particles with n charges |
| $f_{abs,n}$ | The fractional absorption coefficient of particles with n charges |
| $k_f$ | Mass-mobility pre-exponential factor |
| $L$ | Length of DMA, CPMA or AAC |
| $m$ | Particle mass |
| $m_c$ | the geometric mean of $m$ distribution measured by CPMA-CPC |
| $m_{n,max}$ | The maximum $m$ of particles with n charges that can be selected in CPMA classification |
| $m_{n,min}$ | The minimum $m$ of particles with n charges that can be selected in CPMA classification |
| $n$ | Number of elementary charges on the particle |
| $N_{tot}$ | The total number concentration of particles selected by DMA-CPMA |
| PNSD | Particle number size distribution |
| $PNSD_{ae}$ | Particle number aerodynamic size distribution |
| $PNSD_{ve}$ | Particle number volume-equivalent size distribution |
| $q$ | Electrical charge on the particle |
| $Q_a$ | Sample flow rate |
| $Q_{sh}$ | Sheath flow rate |
| $Q_{CPMA}$ | The volumetric flow rate in CPMA |
| $r_a$ | Lower initial radial position that passes through the classifier |
| $r_b$ | Upper initial radial position that passes through the classifier |
| $r_1$ | Inner radium |
| $r_2$ | Outer radium |
| $\hat{r}$ | $r_1 / r_2$ |
| $R_m$ | Mass resolution of CPMA |
| $t$ | Time |
| $\bar{v}$ | Average flow velocity |
| $v_z$ | Axial flow distribution |
| $v_\theta$ | Velocity profile in the angular direction |
| $V$ | Voltage between the two electrodes of DMA or CPMA |
| $Z_p$ | Electrical mobility |
| $Z_p^*$ | $Z_p$ at the maximum transfer function of DMA |
| $\tilde{Z}_p$ | $Z_p/Z_p^*$ |
| $\alpha \beta$ | Azimuthal flow velocity distribution parameter |
| $\alpha_{abs}$ | Absorption coefficient |
| $\alpha_{abs,tot}$ | The total absorption coefficient of particles selected by DMA-CPMA |
| $\beta_{AAC}$ | The ratio of flow rates of aerosol flow and sheath flow of AAC |

| | |
|---|---|
| $\beta_{DMA}$ | The ratio of flow rates of aerosol flow and sheath flow of DMA |
| $\delta$ | Half width of the gap between the two electrodes |
| $\mu$ | Air viscosity |
| $\rho_0$ | Standard density, which equals 1 kg m$^{-3}$ |
| $\rho_{eff}$ | Effective density |
| $\sigma_m$ | The geometric standard deviation of $m$ distribution |
| $\sigma_{ae}$ | The geometric standard deviation of $d_{ae}$ distribution |
| $\tau$ | Relaxation time |
| $\tau^*$ | $\tau$ at the maximum of the transfer function |
| $\tilde{\tau}$ | Dimensionless particle relaxation time, $\tilde{\tau} = \tau/\tau^*$ |
| $\chi$ | The dynamic shape factor |
| $\omega_1$ | Rotational speed of the inner electrode |
| $\omega_2$ | Rotational speed of the outer electrode |
| $\hat{\omega}$ | $\omega_1/\omega_2$ |
| $\Omega$ | Transfer function |

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

**Table 1 Dimensions of the three classifiers used for transfer function calculation**

| Parameter | DMA | CPMA | AAC |
|---|---|---|---|
| $r_1$ (mm) | 9.37 | 100 | 43 |
| $r_2$ (mm) | 19.61 | 103 | 45 |
| L (mm) | 44.369 | 200 | 210 |
| $\omega_2/\omega_1$ | — | 0.945 | — |


**Table 2. Mobility diameter, mass, aerodynamic diameter, effective densities calculated by DMA-AAC and DMA-**
**CPMA, and the deviation between them for fresh soot particles in the size range of 80–250 nm**

| $d_m$ (nm) | $m_c$ (fg) | $d_{ae,c}$ (nm) | $\rho_{DMA-AAC}$ (kg m$^{-3}$) | $\rho_{DMA-CPMA}$ (kg m$^{-3}$) | Deviation |
|---|---|---|---|---|---|
| 80 | 0.16±0.01 | 48.2±0.3 | 551.2±6.9 | 596.8±37.30 | 7.65% |
| 100 | 0.27±0.01 | 54.8±0.3 | 488.0±5.32 | 515.7±19.10 | 5.38% |
| 150 | 0.66±0.07 | 67.8±0.3 | 359. 1±3.22 | 373.5±39.61 | 3.86% |
| 200 | 1.28±0.10 | 82.1±0.6 | 303.2±4.44 | 305.6±23.87 | 0.77% |
| 250 | 2.17±0.16 | 95.9±0.9 | 262.8±4.92 | 265.2±19.56 | 0.90% |


**Table 3. Number concentration fractions and absorption contributions for different size fresh soot particles with**
**single, double or triple charges and the overestimation of MAC accordingly**

| $d_m$ (nm) | $f_{N,1}$(%) | $f_{abs,1}$(%) | $f_{N,2}$(%) | $f_{abs,2}$(%) | $f_{N,3}$(%) | $f_{abs,3}$(%) | MAC overestimation(%) |
|---|---|---|---|---|---|---|---|
| 80 | 72.2±2.5 | 50.6±2.7 | 26.7±3.0 | 45.7±4.2 | 1.1±0.4 | 3.7±1.5 | 42.7±9.1 |
| 100 | 82.4±0.5 | 64.4±0.8 | 17.6±0.5 | 35.6±0.8 | - | - | 28.0±1.8 |
| 150 | 95.8±1.2 | 87.7±3.1 | 4.2±1.1 | 12.3±3.1 | - | - | 9.2±4.1 |


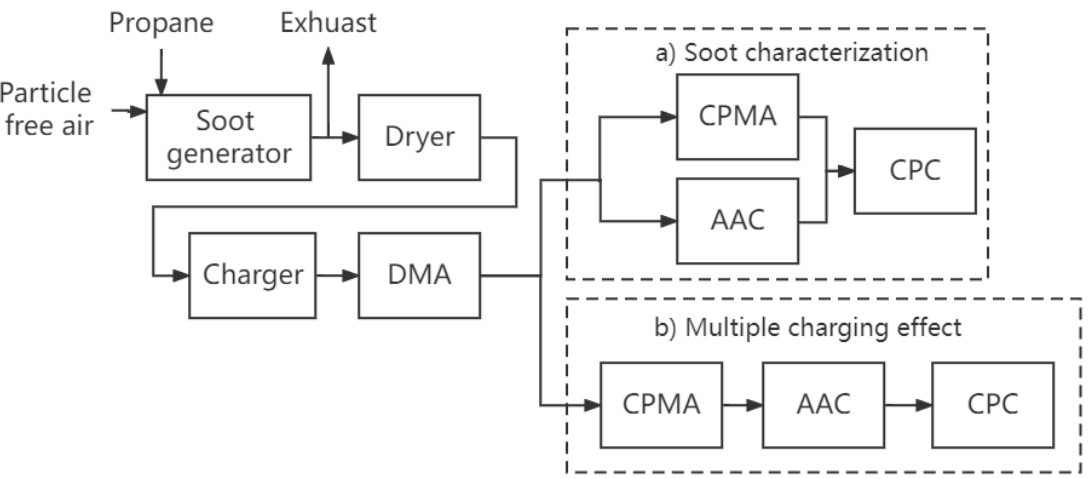


**Figure 1: Schematic of the experimental setup: (a) soot characterization and (b) evaluation of multiple charging**
**effects.**

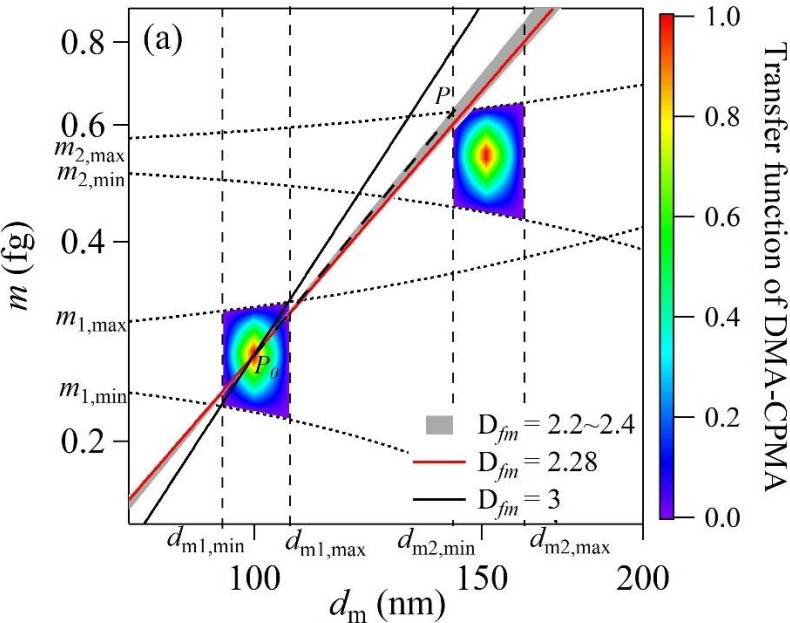


Figure 2: Example of the DMA-CPMA transfer function of flame-generated soot particles (Pei et al., 2018) in $\log(m)$-$\log(d_m)$. The following parameter set was employed for the calculations: $d_m = 100$ nm, $\beta_{DMA} = 0.1$, $m = 0.33$ fg, $Q_{CPMA}=0.3$ L min$^{-1}$, $R_m = 8$. The color blocks are the transfer function of DMA-CPMA, with the rainbow color representing the transfer function for singly charged (lower left block) and doubly charged (upper right block) particles. The black and red solid lines are particles populations with $D_{fm}$ values of 3 and 2.28, respectively. The gray region is the particle population with $D_{fm}$ of 2.2-2.4, which is typical for soot aerosols. The dotted lines are the limits of $d_m$ and m of DMA and CPMA, respectively. The dashed line is the critical slope of $PP_0$. The DMA–CPMA transfer function for +2 particles does not overlap with the line for spherical particles with a single charge ($D_{fm}=3$).

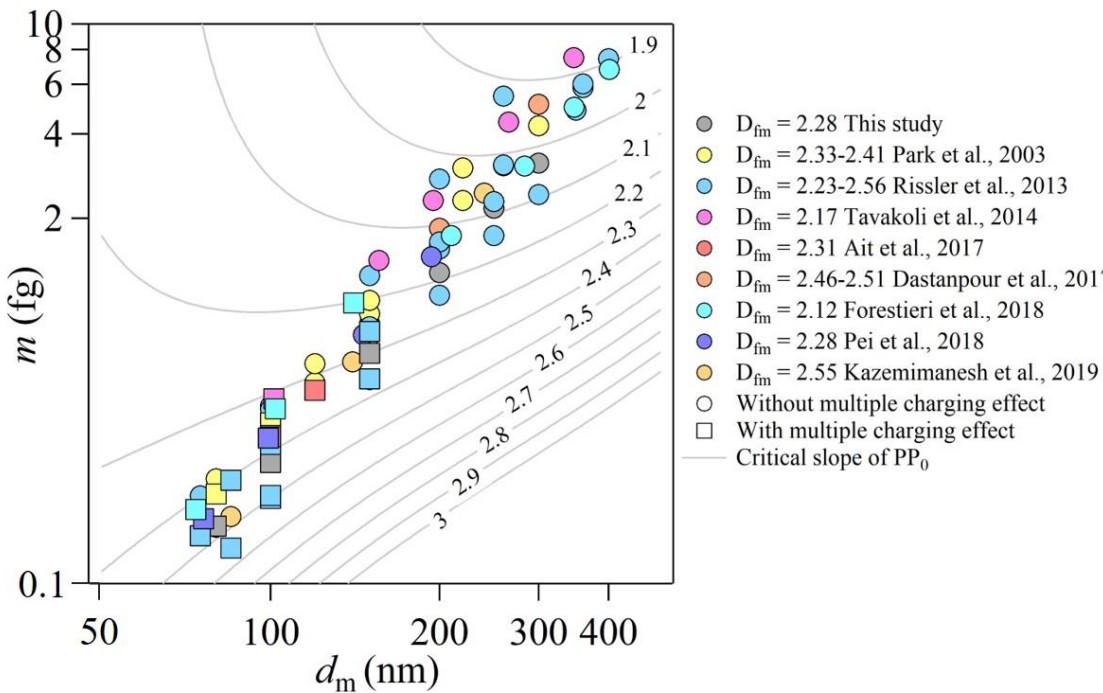


**Figure 3: Variations of the slope of PP$_0$ as a function of classified $d_m$ and $m$. The following parameter set was**
**employed for the calculations: $\beta_{DMA} = 0.1$, $Q_{CPMA}=0.3$ L min$^{-1}$, $R_m = 8$. The contour lines denote the critical slope**
**of PP$_0$, with values labeled on them. The data points are soot particles measured in the literature (Park et al., 2003;**
**Rissler et al., 2013; Tavakoli et al., 2014; Ait Ali Yahia et al., 2017; Dastanpour et al., 2017; Forestieri et al., 2018;**
**Pei et al., 2018; Kazemimanesh et al., 2019) and generated in this study (see details in Sect 3.2). The $D_{fm}$ values of**
**these data points are listed in the legend. The data points become square when $D_{fm}$ is smaller than the critical**
**slope of PP$_0$ in the background, i.e., the potential multiple charging effect may exist.**

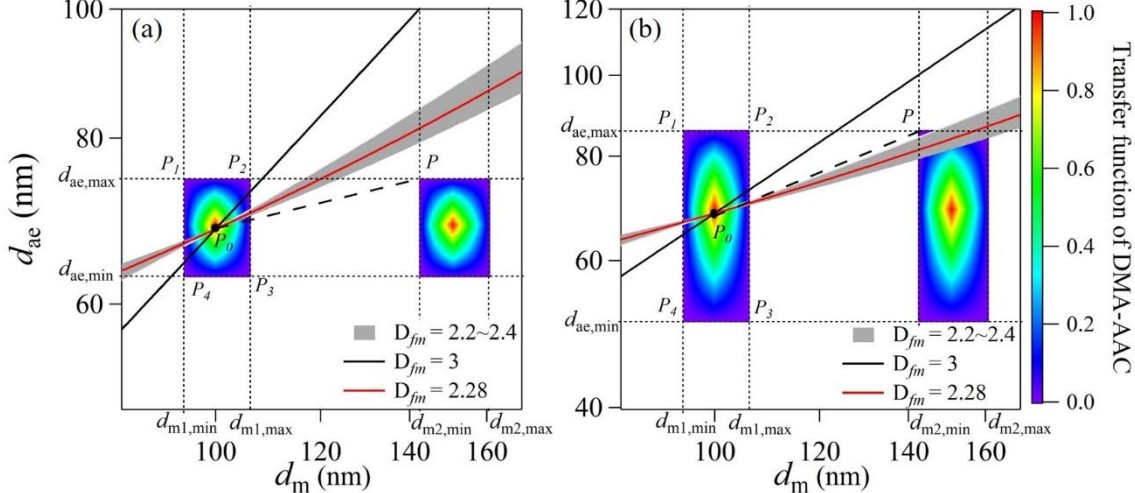


**Figure 4: Examples of transfer function calculation of DMA-AAC of flame-generated soot particles (Pei et al.,**
**2018). The following parameter set was employed for the calculations: $Q_a=0.3$ L min$^{-1}$, $d_{m1} = 100$ nm, $d_{ae} = 68.3$**
**nm, (a) $\beta_{DMA} = 0.1$, $\beta_{AAC} = 0.1$, (b) $\beta_{DMA} = 0.1$, $\beta_{AAC} = 0.3$. The color blocks are the transfer functions of DMA-AAC.**
**The black and red solid lines are particle populations with $D_{fm}$ values of 3 and 2.28, respectively. The gray region**
**is the particles population with $D_{fm}$ of 2.2-2.4, which is typical for soot aerosol. The dashed line is the critical slope**
**of PP$_0$. The dotted lines are the limiting $d_m$ and $d_{ae}$ of DMA and AAC, respectively.**

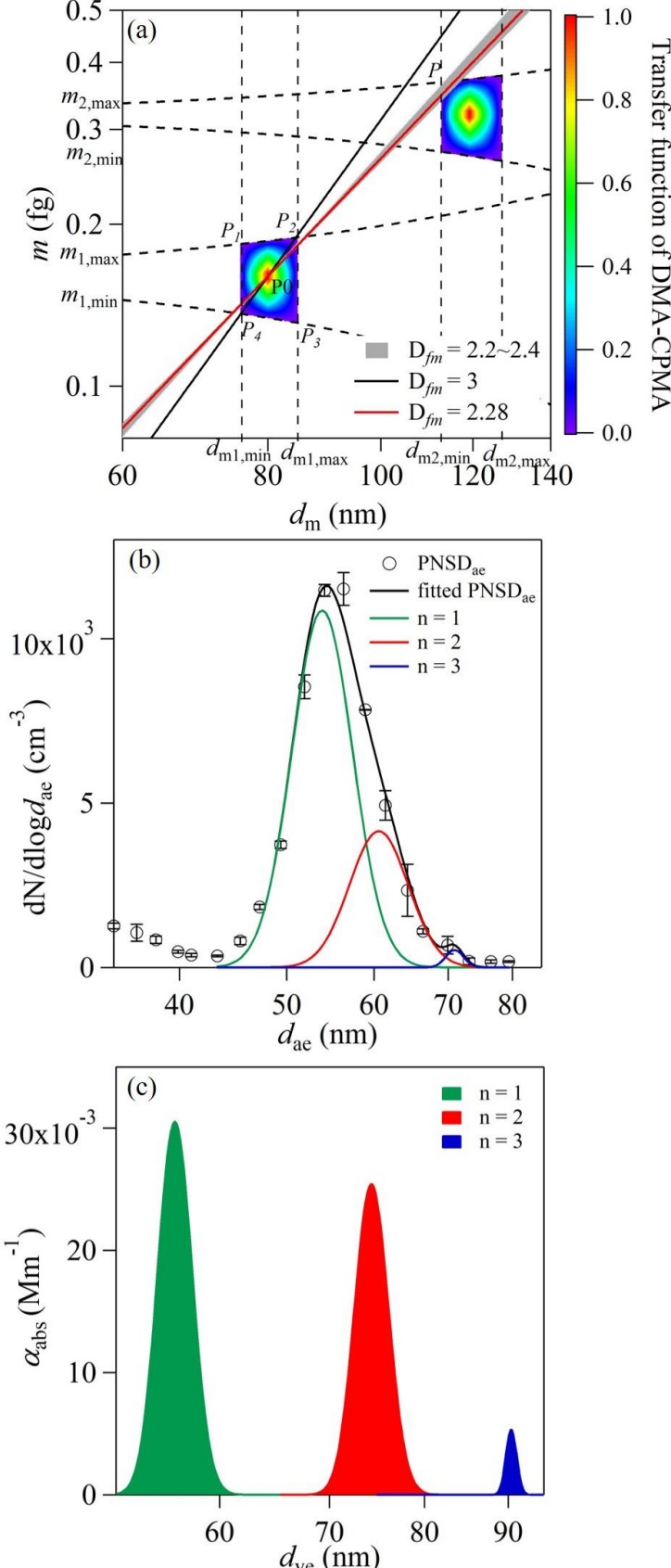


**Figure 5: (a) Transfer functions of DMA-CPMA when selecting 80 nm and 0.16 fg particles. The following**
**parameter set was employed for the calculations: $d_{m1}$ = 80 nm, $\beta_{DMA}$ = 0.1, $m_1$ = 0.16 fg, $Q_{CPMA}$=0.3 L min$^{-1}$, $R_m$ =**
**8. The red solid line is the generated soot particle population. (b) The aerodynamic size distribution of particles**
**classified by DMA-CPMA. The circles are data measured by AAC-CPC, and the black, green, red and blue lines**
**are log-normal fitted distributions of bulk, singly charged, doubly charged and triply charged particles**
**populations. (c) The contributions to light absorption of particles with single, double and triple charges calculated**
**with Mie theory.**

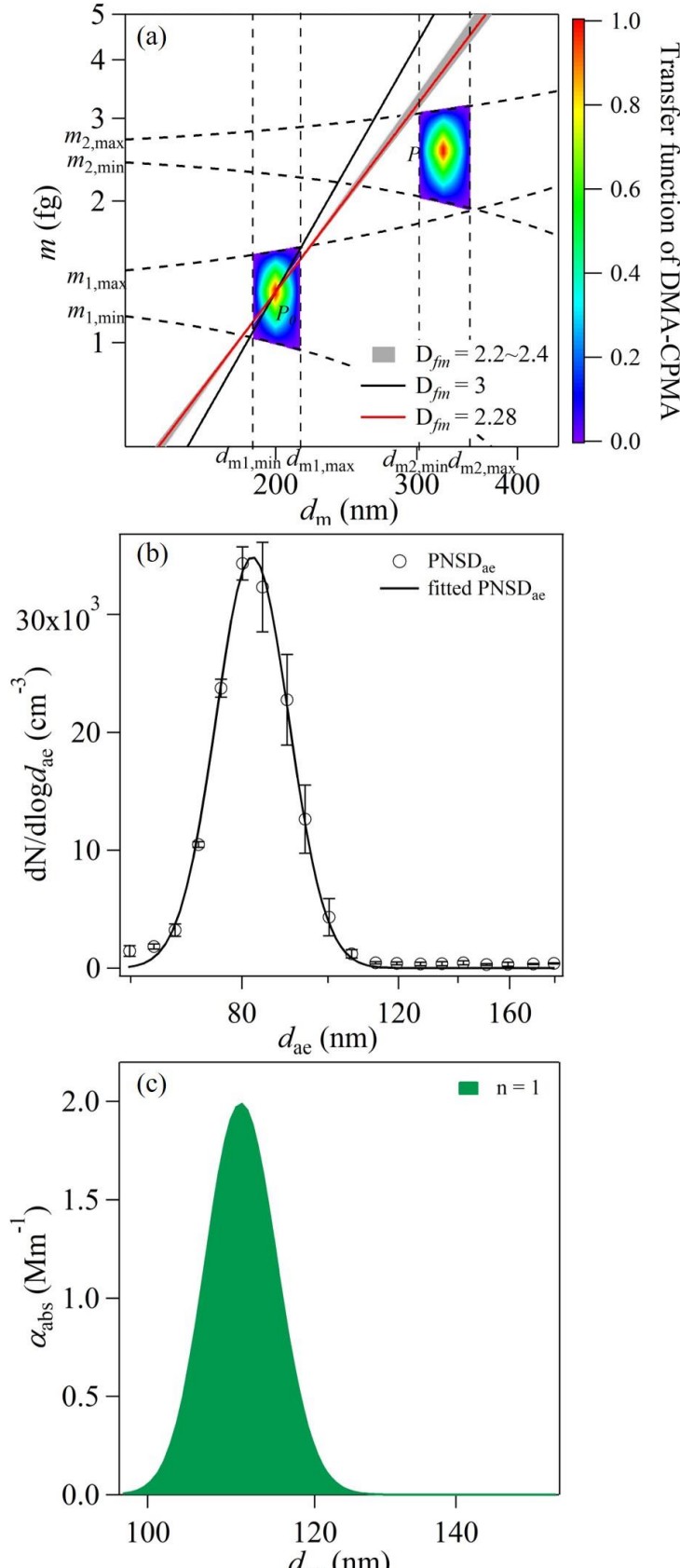

**Figure 6: (a) The transfer functions of DMA-CPMA when selecting 200 nm and 1.28 fg particles. The following**
**parameter set was employed for the calculations: $d_{m1}$ = 200 nm, $\beta_{DMA}$ = 0.1, $m_1$ = 1.28 fg, $Q_{CPMA}$=0.3 L min$^{-1}$, $R_m$ =**
**8. The red solid line is the generated soot particle population. (b) The aerodynamic size distribution of particles**
**classified by DMA-CPMA. The circles are data measured by AAC-CPC, and the solid line is the log-normal fitted**
**distribution. (c) Contributions to light absorption of particles with a single charge calculated with Mie theory.**