# Peer review of "Characterization of tandem aerosol classifiers for selecting"

_Atmospheric Measurement Techniques, 2021_

## Referee Comment (RC1)

**Review of:**

**Characterization of tandem aerosol classifiers for selecting particles: implication for eliminating multiple charging effect**

Yao Song, Xiangyu Pei, Huichao Liu, Jiajia Zhou, Zhibin Wang

**General comments:**

The manuscript by Song et al. examines the problem of multiple charging when using tandem aerosol measurement methods. Specifically, the authors investigated the combinations of a differential mobility analyzer (DMA) with either a centrifugal particle mass analyzer (CPMA) or aerodynamic aerosol classifier (AAC); DMA-CPMA and DMA-AAC, respectively. The authors modeled the transfer function of all instruments to derive a limiting case below which it would not be possible to resolve the contributions of particles bearing multiple charges and refer to this as PP0. Using flame generated soot, the authors then demonstrate the ranges of mobility diameters ($D_m$) where the particle contributions as a function of charge can be isolated.

The technical quality of this manuscript is not very good; the authors appear to have performed their transfer function calculations incorrectly.

**Specific AMT review criteria:**

1. The manuscript could represent a substantial contribution to scientific progress within the scope of Atmospheric Measurement Techniques and this is specifically highlighted by the derivation of Equations 25 and 27 for the limiting cases for the complete separation of multiply charged particles. In my opinion, the authors need to refer to these cases as something better than PP0 and make mention them in the abstract since this seems to be the most important takeaway. However, the clarity of the presented results is lacking, and it appears the presented transfer functions are not correct, which severely detracts from the manuscript. Rating: poor

2. The scientific approach and applied methods seem valid, but not necessarily the calculations, with limited discussion of the results but in an appropriate and balanced way. Rating: fair

3. The presentation of the scientific results and conclusions needs significant improvement. The use of English language is not fluent or precise in places and this distracts from the information the authors are trying to convey. Additionally, there are sections of the discussion that should be significantly expanded, and this expansion should aid in clarity. Rating: fair

This manuscript is likely publishable after significant revision since the topic is of interest to the community.

**Specific comments:**

1. I'm really struggling to understand the figures because of the rainbow color scale used and I strongly recommend using a different color scheme. Additionally, this color scheme is not visually accessible to all readers; *e.g.* (Nuñez, Anderton and Renslow 2018).

2. Uncertainties are missing on reported values throughout.

3. The authors investigated the pairwise combinations of DMA-CPMA and DMA-AAC but some mention of their expectations when utilizing other orderings (AAC-CPMA, AAC-DMA, DMA-CPMA) is needed. This is especially true for an AAC-DMA since the ratio of the $\beta$'s may depend upon this ordering. (i.e. does it matter whether the transfer function of the first instrument is narrower/similar/wider than the second?)

4. It appears that the calculated transfer functions do not include the effect of the mass to charge ratio on the resolution of the CPMA in Figures 2, 5 and S2; effective masses, not absolute masses are the key quantity being measured.

For example: in Figure 5a, the authors state that $D_m$ = 80 nm, $m$ = 0.16 fg and $D_{fm}$ = 2.28. Solving the $\underline{D}_{fm}$ relationship yields $\rho_f$ = 7.3 × 10$^{-6}$ fg nm$^{-2.28}$. At $D_m$ = 120 nm, m ≈ 0.40 fg which agrees with the data shown in the figure. Unfortunately, because $q$ = 2, the effective mass would be a factor of 2 lower and should be ≈ 0.2 fg. It appears that a similar error is present in Figures 2 and S2.

5. Figure 4 appears to correspond to utilizing an AAC-DMA combination rather than a DMA-AAC as is discussed in the text. In the DMA-AAC, you'd get two populations of particles in $D_{ae}$-space since the distributions of particles have equivalent $Z_p$ exiting the DMA but are physically different sizes with different $q$. In the AAC-DMA, you'd only have one population in $D_{ae}$-space, the AAC selects one physical size, and then that distribution would have multiple charge states exiting the DMA.

5. The authors fit the distributions of aerodynamic diameter ($D_{ae}$) utilizing multiple log-normal distributions, but it doesn't seem to me that they have the resolution to constrain these values even though we know that multiple charges are present *a priori*. More discussion of the fitting routine is necessary. For example, in Fig. 5b, a) How does the fit compare to using just a single log-normal distribution? Or a single Gaussian or summation of multiple Gaussian distributions? b) Were the central values of $D_{ae}$ constrained prior to the fit? Or were they allowed to float? c) Were the widths of the distributions constrained in any way prior to the fit? d) What are the magnitude of the uncertainties in each of the fit coefficients? e) Were the uncertainties in particle number densities included in the fits? f) The peak of the distribution is significantly underfit. Is it possible that more than q = 1 and 2 are contained within the primary peak and what was identified as q = 3 is 4 or higher?

**Technical corrections:**
**Line 15:** "effect"

effects

**Line 16:** "technique"

techniques

**Line 18:** "the potential multiple charging effect"

Elaborate.

**Line 19:** "remove"

Resolve?

**Line 20:** "instruments setups of DMA-CPMA system"

What is meant by instrument setups? Elaborate.

**Line 23:** "DMA-AAC can eliminate multiple charging effect"

This is not strictly correct as written since you can only "remove" the multiple charging artifact when used in a static configuration. In a scanning mode, the contributions would be resolvable.

**Line 24:** "while particles with multiple charges can be selected when decreasing resolutions of DMA and AAC"

Confusing as written.

**Line 25:** "We propose that the multiple charging effect should be reconsidered when using DMA-CPMA or DMA-AAC system in estimating size and mass resolved optical properties in the field and lab experiments."

This statement is not clear as written. How should the effects be "reconsidered"?

**Line 35:** "is the most commonly used size classifier"

If the DMA is the most commonly used classifier, why is only the original Knutson and Whitby reference provided?

**Line 38:** "particles are required to be pre-charged"

In what sense do they need to be pre-charged?

**Line 39:** "resulting in that particles"

Delete "that".

**Line 41:** "subsequence"

Subsequent

**Line 49:** "This conclusion implies that it can hardly to achieved that all the multiply charged particles are effectively excluded for aspherical particles, especially for soot particles."

Grammar makes this sentence confusing.

**Line 51:** "conducted"

Investigated?

**Line 52:** "ammonia"

Ammonium

**Line 60:** "dynamic shape factor ($\chi$), can be inferred…"

The measured $\chi$ may depend upon the combination of instruments used. See (Yao et al. 2020) and potentially Table 2 here.

**Line 80:** "particle"

Elementary

**Line 89:** "The transfer function is an isosceles triangle with value of 1 at $Z$p * and going to 0 at (1 ± $\beta$DMA)·$Z$p*."

I think it is important to mention that this translates to asymmetric distributions in Dm and mp since their relationship with $Z_p$ is nonlinear.

**Line 106:** "is much simpler and more robust"

Elaborate.

**Line 107:** "radical"

Radial

**Line 113:** "should"

would

**Line 134 and Line 328:** "nominated"

Nominal?

**Line 142:** "80, 100, 150, 200"

Missing units.

**Line 146:** "while the condensation particle counter"

Was only a single CPC used during the soot characterization experiments? In the previous sentences, the author make it seem like the measurements were made simultaneously and in parallel. Also, please include flow rates.

**Line 147:** "concentration"

Number density of particles. Concentration is assumed to have units of mole per unit volume.

**Line 148:** "fitted to log-normal distribution"

Please include the equation that you used to fit your distribution since there are many ways to define the same relationship. Also, considering the shapes of the distributions shown in Fig. 5 and S2, why was a log-normal distribution utilized? The distributions appear symmetric, and they're plotted on a linear axis, so some justification is warranted.

**Line 149:** "70, 150"

Missing units.

**Line 151:** "density of PSL"

Please enumerate.

**Line 153:** "effect of particles selected by DMA-CPMA system, the $d$ae distribution of twice classified particles".

Please provide additional information about this portion of the procedure.

**Line 167:** "we explain the transfer functions of DMA-CPMA and DMA-AAC utilizing the literature data of soot particles"

When the transfer functions were calculated, what range of parameters were used? And how exactly were the transfer functions solved? Iteratively or something else? If iteratively, what was the $\Delta t$ for each step and the number of individual trajectories considered? These details need to be provided somewhere in the manuscript.

**Line 178:** "representing pre-exponential factor ($\rho f$)"

Having the pre-exponential factor share a variable with effective density ($\rho_{eff}$) is confusing since I would expect them to share units when they do not; the units are (g nm$^{-Dfm}$) and (g nm$^{-3}$) for $\rho_f$ and $\rho_{eff}$, respectively. Additionally, I recommend including a normalization factor in the Dm term to avoid having fractional units, *e.g.*, $D_{fm}$ = 2.28 will have $\rho_f$ with units of g nm$^{-2.28}$.

**Line 182:** "however,"

However is used twice in the same sentence.

**Line 184:** "In the exemplary case, the derived $D$fm of premixed flame generated soot particles was 2.28,"

What study does this refer to? Reference?

**Line 189:** "The DMA-CPMA system can eliminate the multiply charged particles only if the $D$fm of particles is larger than the slope of a line connecting line connecting ($d$m, $m$) = ($d$m2,min, $m$2,max)($d$m1,,$m$1) (as PP0 shown in Fig. 2)."

The point (dm2, m2) appears to correspond to the actual mass and dm of a particle bearing 2 charges instead of the effective mass (m/2). This is unclear and has significant implications for the calculated transfer functions and resultant discussion since the effective mass is ultimately what affects instrument resolution.

**Line 190:** "line connecting ($d$m, $m$) = ($d$m2,min, $m$2,max)($d$m1,,$m$1) (as PP0 shown in Fig. 2)."

I'm assuming that the location of (dm1, m1) is point P0 and is at the center of the q = 1 transfer function? This is not clear in the figure. There's an extra "," after dm1 in the text.

**Line 201:** "are necessary to reduce the potential of multiply charged particles"

By increasing the resolution of the measurement?

**Line 205:** "PP0 of 3.55 was derived when DMA-APM selects the same example soot particles"

Compared to what?

**Line 208:** "critical slope of PP0"

Z-axis in Figure is labelled as "The slope of PP0".

**Line 210:** "when the $D$fm of particles is larger than the slope of PP0 which is represented as background color."

So, the color of the data point will be red shifted relative to the background when multiple charging is not affecting the output distribution? This is unclear and the rainbow color scheme isn't helping.

**Line 213:** "Fig. 3, the $d$m, $m$ and corresponding $D$fm were taken from literature"

In the caption, the authors mention that the shapes correspond to the individual studies. Please elaborate which is which here?

**Line 214:** "Generally, multiple charging effect can be avoided for DMA-CPMA to select soot particles with diameter larger than 200 nm.,"

I don't think the authors can conclude this without providing more data to support the claim. For fresh soot, my experience has been that multiple charging can be a problem at almost all Dm; e.g. see Figure S1 of (Radney et al. 2014). Also, there's a "," after the "." at the end of the sentence.

**Line 215:** "diameter"

To which diameter metric are you referring?

**Line 216:** "eliminate"

Resolve?

**Line 220:** "Therefore, the multiple charge effect could be avoided theoretically."

Measurements by just an AAC will avoid multiple charging. Multiple charging only becomes a problem again when the tandem measurement is a DMA or PMA.

**Line 226 to end of paragraph:** "In order to simulate the transfer function of DMA-AAC selecting the same particles as that used in calculations of DMA-CPMA. The corresponding $d$ae…"

See Specific Comment 5.

**Line 242:** "resulting in the minimum $D_{fm}$ of 1.41, which is the case for most atmospheric aerosol particles."

Is this the black dashed line drawn as PP0? If so, please label in figure. Also, please differentiate these dashes from the vertical and horizontal ones.

**Line 245:** "sheath flow rate of classifier is restricted by the instrument design"

It's also important to note that sheath flow restricts the maximum size ranges.

**Line 252:** "uncertainties"

Uncertainties does not seem to be the correct word here. Biases?

**Line 255:** "$D$fm was 2.28"

What was the value of $\rho_0$?

**Line 258:** "along"

Delete word.

**Line 261:** "80, 100, 150, 200"

Units?

**Line 263:** "which suggested that multiply charged particles are still classified in this circumstance."

This suggests that the contributions from the multiply charged particles can't be resolved.

**Line 268:** "the multiply charged particles can be resolved in aerodynamic size distribution"

I disagree that the $q$ = 1 and 2 can be resolved. Please provide more evidence to support this claim.

**Line 270:** "PNSDae was fitted using log-normal distributions and three peaks which correspond to singly, doubly and triply charged particles were identified."

What does the peak at Dae < 40 nm correspond to? Please mention.

**Line 296:** "Mie theory was used to calculate the theoretical absorption coefficient at the wavelength of 550 nm."

Mie theory probably isn't the "best" method to use here since soot particles are aspherical agglomerates. Realistically though, the Mie comparison is only being used to prove a point about the impact of multiple charging. So, in this instance any errors in the calculated optical properties are somewhat inconsequential. Some mention of this nuance should be mentioned.

**Line 301:** "integral concentration for particles"

Integrated number density of particles. Concentration is assumed to have units of mole per unit volume.

**Line 308:** "70"

Units?

**Table 2:** It'd be interesting (but not necessary) to include a comparison of the derived shape factors ($\chi$) for each method.

**Figure 2:** "Example of DMA-CPMA transfer function"

Transfer function for what? Soot?

**Figure 3:** The coloration of this figure is very hard to understand.

**Figure 4:** Z-axis is labelled "Transfer function of DMA-CPMA" and the caption says DMA-AAC.

**Figure S1:** Having the Z-axis go all the way to zero is confusing.

**References:**

Nuñez, J. R., C. R. Anderton, R. S. Renslow. 2018. Optimizing colormaps with consideration for color vision deficiency to enable accurate interpretation of scientific data. *PLOS ONE* 13:e0199239. doi: 10.1371/journal.pone.0199239.

Radney, J. G., R. You, X. Ma, J. M. Conny, M. R. Zachariah, J. T. Hodges, C. D. Zangmeister. 2014. Dependence of soot optical properties on particle morphology: Measurements and model comparisons *Environ. Sci. Technol.* 48:3169-3176. doi: 10.1021/es4041804.

Yao, Q., A. Asa-Awuku, C. D. Zangmeister, J. G. Radney. 2020. Comparison of three essential sub-micrometer aerosol measurements: Mass, size and shape. *Aerosol Sci. Technol.*:1-13. doi: 10.1080/02786826.2020.1763248.

---

## Author Comment (AC1)

**Response to James Radney**

We thank the reviewer for the constructive suggestions and comments concerning our manuscript entitled "Characterization of tandem aerosol classifiers for selecting particles: implication for eliminating multiple charging effect" (ID: amt-2021-436). Those comments are valuable and very helpful for improving our paper, as well as the important guiding significance to our studies. Below, we provide a point-by-point response to individual comment (Reviewer comments in italics, responses in plain font; page numbers refer to the AMTD version; Tables used in the response are labeled as Table S1, Table S2,…, figures used in the response are labeled as Fig. R1, Fig. R2,…)

**General Comments and suggestions:**

*The manuscript could represent a substantial contribution to scientific progress within the scope of Atmospheric Measurement Techniques and this is specifically highlighted by the derivation of Equations 25 and 27 for the limiting cases for the complete separation of multiply charged particles. In my opinion, the authors need to refer to these cases as something better than PP0 and make mention them in the abstract since this seems to be the most important takeaway. However, the clarity of the presented results is lacking, and it appears the presented transfer functions are not correct, which severely detracts from the manuscript. Rating: poor*

*The scientific approach and applied methods seem valid, but not necessarily the calculations, with limited discussion of the results but in an appropriate and balanced way. Rating: fair*

*The presentation of the scientific results and conclusions needs significant improvement. The use of English language is not fluent or precise in places and this distracts from the information the authors are trying to convey. Additionally, there are sections of the discussion that should be significantly expanded, and this expansion should aid in clarity. Rating: fair*

**Responses and Revisions:**

Thank you for the advice. In summary, we are very sorry for the misunderstanding due to the poor expression. The revisions could be found in individual comment in details.

For our main contribution, Equations 26 and 28 demonstrate the relationship between $D_{\text{fm}}$ and the instrument resolutions when using DMA-CPMA to eliminate multiple charging effect. We have included a brief description of these equations in the abstract: "We propose an equation that constrains the resolutions of DMAs and CPMAs to eliminate the multiple charging effect when selecting particles with a certain mass–mobility relationship using the DMA-CPMA system. The equation for the DMA-AAC system is also derived"

For transfer functions of tandem system of DMA-CPMA and DAM-AAC, we think our calculation is credible, while the expression leads to the misunderstanding. We calculated the transfer functions of DMA-CPMA and DMA-AAC both in a static configuration. The transfer function of DMA-CPMA was derived by multiplying the transfer functions of DMA and CPMA. The transfer function of DMA-AAC was derived by multiplying the transfer functions of DMA and AAC. We have reorganized our language in the revised manuscript. The details could be found in Comments 4.

Our English writing has been promoted by native speaker.

**1. *Comments and suggestions:**

*I'm really struggling to understand the figures because of the rainbow color scale used and I strongly recommend using a different color scheme. Additionally, this color scheme is not visually accessible to all*

*readers; e.g. (Nuñez, Anderton and Renslow 2018).*

**Responses and Revisions:**

We have revised Fig.3 to contour plot. It is more straightforward to compare the reported $D_{fm}$ and critical slope PP$_0$ values. The selected particles with multiple charging effect are resented as squares. The colors are used to distinguish literature data. Moreover, we have included the black and white version in the supplement (Fig. S4). According to the Color BLIndness Simulator, this figure should be readable for all readers.

[Figure]

**Figure 3: Variations of the slope of PP0 as a function of classified $d_m$ and $m$. The following parameter set was employed for the calculations: $\beta_{DMA} = 0.1$, $Q_{CPMA}=0.3$ L min$^{-1}$, $R_m = 8$. The contour lines denotes the slope of PP0 with values labeled on them. The data points are soot particles measured in literatures (Park et al., 2003; Rissler et al., 2013; Tavakoli et al., 2014; Ait Ali Yahia et al., 2017; Dastanpour et al., 2017; Forestieri et al., 2018; Pei et al., 2018; Kazemimanesh et al., 2019) and generated in this study (See details in section 3.2).The $D_{fm}$ of these data points are listed in the legend. The data points become square when the $D_{fm}$ is smaller than the critical slope of PP0 in the background, i.e. the potential multiple charging effect may exist.**

In the revised manuscript, the figures (Fig. 2, Fig. 4, Fig. 5a and Fig 6a) with rainbow color scale are kept to compare with the previous study representing the DMA-APM transfer function (Kuwata et al., 2015), we hence kept them likewise.

**2. *Comments and suggestions:**
*Uncertainties are missing on reported values throughout.*

**Responses and Revisions:**

We have added the uncertainties in Table 2 and Table 3.

**3. *Comments and suggestions:**
*The authors investigated the pairwise combinations of DMA-CPMA and DMA-AAC but some mention of their expectations when utilizing other orderings (AAC-CPMA, AAC-DMA, DMA-CPMA) is needed. This is especially true for an AAC-DMA since the ratio of the ß's may depend upon this ordering. (i.e. does it matter whether the transfer function of the first instrument is narrower/similar/wider than the second?)*

**Responses and Revisions:**

This is a very good suggestion. The transfer function of AAC-DMA is calculated with the transfer functions of two classifiers (Tavakoli and Olfert, 2014):

$$\Phi_{AAC-DMA} = \Omega_{DMA}\Omega_{AAC} , \qquad\qquad R1$$

we think the transfer function of DMA-AAC in a static configuration is independent on their ordering. An example can be found in comment 5.

As for DMA-CPMA, the resolution of CPMA is calculated assuming that all the particles have exactly the same mobility:

$$R_m = \frac{m_1}{m_{1,max}-m_1} = \frac{2\pi B_{1,max} L_{CPMA} r_c^2 \omega^2 m_1}{Q_{CPMA}} \qquad\qquad R2$$

where $m_1$ and $m_{1,max}$ are the nominal mass and the maximum mass that can be selected by CPMA, respectively. $B_{1,max}$ is the mobility of particles with mass of $m_{1,max}$. We assume that $B_{1,max}$ is equal to the mobility of nominal particles. This assumption is valid when the particles are mobility selected before fed into the CPMA, so the ordering of DMA-CPMA can't be changed.

We didn't include the transfer function of AAC-CPMA or CPMA-AAC in our study. We attempted to calculate the transfer function of AAC-CPMA, but we found it was difficult since particle mobility should be provided first to connect the calculations of AAC and CPMA. The relationship between $\tau$ selected by AAC and $m$ selected by CPMA is shown as follows (Yao et al., 2020):

$$\tau = m_{sp}B = \chi m B, \qquad\qquad R3$$

in which $m_{sp}$ is mass of spherical particles. CPMA determines real mass of particles without any assumption of $\chi$. Mobility cannot be derived from Eq. R3 since $\chi$ is unknown.

**4. *Comments and suggestions:**

*It appears that the calculated transfer functions do not include the effect of the mass to charge ratio on the resolution of the CPMA in Figures 2, 5 and S2; effective masses, not absolute masses are the key quantity being measured.*

*For example: in Figure 5a, the authors state that $D_m = 80$ nm, $m = 0.16$ fg and $D_{fm} = 2.28$. Solving the $\underline{D_{fm}}$ relationship yields $\rho_f = 7.3 \times 10^{-6}$ fg nm$^{-2.28}$. At $D_m = 120$ nm, $m \approx 0.40$ fg which agrees with the data shown in the figure. Unfortunately, because $q = 2$, the effective mass would be a factor of 2 lower and should be $\approx 0.2$ fg. It appears that a similar error is present in Figures 2 and S2.*

**Responses and Revisions:**

Thank you for the comment. We thought that we did not clearly clarify our instruments configuration. Usually, the tandem setup of DMA-CPMA is that DMA firstly selects a fixed mobility, while CPMA is used in scanning mode to derive the corresponding mass . For example, in Figure 2 in Radney et al. (2013), they fixed the DMA to select particles at $d_m$ of 200 nm and particles bearing higher-order charges were also selected because of multiple charging effect. The mobility-selected particles included particles bearing 1, 2 and 3 charges with $d_m$ of 200 nm, 321 nm and 434 nm, respectively. The corresponding $m$ of 7.4 fg, 30.7 fg and 75.8 fg can be calculated using $d_m$ and effective density, respectively. The scanning mode of APM was used downstream to resolve multiple charging effect of these mobility-selected particles. The APM selects particles according to their mass-to-charge ratio. As a result, the effective masses measured by APM were 7.4 fg, 15.3 fg, 25.3 fg, respectively, which is shown as grey circles in Figure 2 (Radney et al., 2013)

In our study, we also use scanning mode of CPMA after DMA selection to determine the mode mass of the selected particles, then we use the tandem setup of DMA and CPMA both at fixed mode to select particle at fixed mobility and mode mass i.e. DMA and CPMA are used in a static configuration, no scanning for either

instrument is used. In Figure 5a, DMA-CPMA is set to select singly charged particles with $d_m$ of 80 nm and $m$ of 0.16 fg, while the doubly charged particles with $d_m$ of 119.3 nm and $m$ of 0.32 fg will also be selected and the transfer function is presented as upper right region. Soot particles curve (red line) goes through the upper-right region which doubly charged particle can penetrate ($d_m$ of 113 nm~118 nm, $m$ of 0.35 fg~0.39 fg). As a result, we conclude that multiple charging effect still exists when DMA-CPMA select soot particles with $d_m$ of 80 nm and $m$ of 0.16 fg.

**5. *Comments and suggestions:**

*Figure 4 appears to correspond to utilizing an AAC-DMA combination rather than a DMA-AAC as is discussed in the text. In the DMA-AAC, you'd get two populations of particles in $D_{ae}$-space since the distributions of particles have equivalent $Z_p$ exiting the DMA but are physically different sizes with different q. In the AAC-DMA, you'd only have one population in $D_{ae}$-space, the AAC selects one physical size, and then that distribution would have multiple charge states exiting the DMA.*
*AND*
***Line 226 to end of paragraph:*** *"In order to simulate the transfer function of DMA-AAC selecting the same particles as that used in calculations of DMA-CPMA. The corresponding $d_{ae}$..."*
*See Specific Comment 5.*

**Responses and Revisions:**
Yes, we agree that AAC selects only one population. This population has one physical size ($d_{ae}$) but the $d_m$ range of this population is wide since soot particles have different densities. Kazemimanesh et al. (2022) demonstrated that AAC does not constrain the properties of nonspherical particles as monodisperse as DMA or CPMA classification. The AAC selects relaxation time, not directly aerodynamic diameter. The relationship between the relaxation time and mobility diameter can be expressed as follows,

$$\tau = \frac{\rho_{eff} d_m^2 Cc(d_m)}{18\mu}, \tag{R4}$$

which indicates the AAC selects monodispersed particles when particles have the same effective density. In our study, the effective density of soot particles decreases with increasing $d_m$. Particles with different $d_m$ but the same relaxation time will be selected.

We think the transfer functions of DMA-AAC or AAC-DMA are identical regardless of the order of DMA and AAC. For example, we use AAC-DMA to select particles with $d_{ae}$ of 68 nm and $d_m$ of 100 nm. In figure 4a in this study, the transfer function of AAC is the region between the horizontal lines of $d_{ae,max}$ (75 nm) and $d_{ae,min}$ (63 nm). The soot particles population (red line) goes through this region will be selected by AAC. The mobility diameter distribution of these relaxation time selected particles is around 80 nm to 120 nm. Then the DMA is fixed to select particles with $d_m$ of 100 nm, the particles with double charges and the same mobility ($d_m$ of 150 nm) have been excluded by AAC. As a result, AAC-DMA select monodispersed particles with $d_{ae}$ of 68.3 nm and $d_m$ of 100 nm. In Fig. 4b, the resolution of AAC is lower and transfer function of AAC is broader than that in Fig. 4a. The soot particles population (red line) goes through the transfer function region between the horizontal lines at $d_{ae}$ of $d_{ae,max}$ (50 nm) and $d_{ae,min}$ (86 nm). The mobility diameter distribution of these relaxation time selected particles is very wide from less than 80 nm to about 158 nm. Then these relaxation time selected particles were charged and selected by DMA at $d_m$ of 100 nm, singly charged particles with $d_m$ of 95 nm~106 nm and doubly charged particles with $d_m$ of 142 nm~158 nm will be selected.

If we use the DMA-AAC, the particles are selected by DMA first. For example, in Figure 4b, the transfer function of DMA is shown as two vertical regions which particles with single and double charges can penetrate. The soot particles (red line) goes through it and two populations of soot particles with mode $d_m$ of 100 nm and 150 nm will be selected. The corresponding $d_{ae}$ distributions of these singly and doubly charged particles are

66 nm~70 nm and 81 nm~87 nm. These mobility-selected particles are selected at $d_{ae}$ of 68.3 nm by AAC and the transfer function of AAC shows that particles with $d_{ae}$ of 50 nm~86 nm can penetrate. As a result, singly charged particles with $d_{ae}$ of 66 nm ~70 nm and doubly charged particles with $d_{ae}$ of 81 nm ~86 nm can be selected.

As a summary, the transfer functions of DMA-AAC and AAC-DMA in a static configuration are the same no matter the ordering of DMA and AAC.

**6. *Comments and suggestions:**

*The authors fit the distributions of aerodynamic diameter ($D_{ae}$) utilizing multiple log-normal distributions, but it doesn't seem to me that they have the resolution to constrain these values even though we know that multiple charges are present a priori. More discussion of the fitting routine is necessary. For example, in Fig. 5b, a) How does the fit compare to using just a single log-normal distribution? Or a single Gaussian or summation of multiple Gaussian distributions? b) Were the central values of $D_{ae}$ constrained prior to the fit? Or were they allowed to float? c) Were the widths of the distributions constrained in any way prior to the fit? d) What are the magnitude of the uncertainties in each of the fit coefficients? e) Were the uncertainties in particle number densities included in the fits? f) The peak of the distribution is significantly underfit. Is it possible that more than q = 1 and 2 are contained within the primary peak and what was identified as q = 3 is 4 or higher?*

**Responses and Revisions:**

Thank you for raising this question. First of all, the size distribution of aerosols is often found to be a log-normal distribution. Then, the $d_{ae}$ distribution is asymmetrical on a linear axis and we found that $d_{ae}$ distribution was fitted well with log-normal distribution.

a)  We used DMA-CPMA in a static configuration to select soot particles with specific $d_m$ and $m$. Particles with higher order charges can also be selected and the values of $d_m$ and $m$ can be calculated. According to Eq. (1) and Eq. (16), the $d_{ae}$ of particles with different charges can be calculated by:

$$\frac{\pi}{6}\rho_0 Cc(d_{ae})d_{ae}^2 = \frac{Cc(d_m)}{d_m}k_f[d_m(nm)]^{D_{fm}} \cdot 10^{-18} , \qquad\qquad R5$$

The corresponding $d_{ae}$ of the selected particles with different charges can be calculated with selected $d_m$ and $m$ according to Eq. R5. The $d_{ae}$ for different sizes soot particles with single, double and triple charges are shown in Table R1. The $d_{ae}$ distribution for different sizes particles was fit well with single log-normal distribution and the values of mode $d_{ae}$ were determined, which denoted as $d_{ae\_sd}$. The deviations between the calculated $d_{ae}$ and fitted $d_{ae}$ for particles with $d_m$ of 150 nm, 200 nm and 250 nm are within 0.31% while the $d_{ae\_sd}$ for particles with 80 nm and 100 nm are much larger than the calculated $d_{ae}$. The deviations for 80 nm and 100 nm particles are 7.38% and 6.85%, respectively. We think that the peaks shift right because of multiple charged particles, so we use the summation of multiple log-normal distributions. Although the deviation for 150 nm particles is 0.29%, the summation of multiple log-normal distributions is used because the points with $d_{ae}$ larger than 90 nm can't be fitted well with single log-normal distribution.

**Table R1. The calculated $d_{ae}$ for mobility and mass selected particles with single, double or triple charges and the single log-normal fitted $d_{ae\_sd}$. The deviations between the calculated $d_{ae\_1q}$ of particles with single charge and the fitted $d_{ae\_sd}$ for different size soot particles.**

| $d_m$(nm) | $m$ (fg) | Calculated $d_{ae}$ (nm) | $d_{ae\_sd}$ (nm) | Deviation |
|---|---|---|---|---|

|      |       | $d_{ae\_1q}$ | $d_{ae\_2q}$ | $d_{ae\_3q}$ | $d_{ae\_4q}$ |       |        |
|------|-------|--------------|--------------|--------------|--------------|-------|--------|
| 80   | 0.16  | 51.5         | 62           | 70.7         | 78.5         | 55.3  | 7.38%  |
| 100  | 0.27  | 56.9         | 70.4         | 81.8         | 92.3         | 60.8  | 6.85%  |
| 150  | 0.66  | 70.1         | 91.8         | 111.2        | 129.4        | 70.3  | 0.29%  |
| 200  | 1.28  | 82.9         | 113.9        | 141.8        | 167.7        | 83.1  | 0.24%  |
| 250  | 2.17  | 95.6         | 157.8        | 172.2        | 205.5        | 95.9  | 0.31%  |

b) The DMA-CPMA selected particles with specific $d_m$ and m, and the corresponding $d_{ae}$ of selected particles with different charges can be calculated with Eq. R5. The calculated values are set as the start points of central values of $d_{ae}$. For example, the central values of $d_{ae}$ used in 80 nm particles fitting are 51.5 nm, 62 nm and 70.7 nm.

c) The constraint of peak width of particles with single, double and triple charges were the same. The start point of the coefficient $\sigma$ (two times of peak width) was e^0.5, and the lower and upper bounds were e^0 and e^1, respectively.

d) The fitting coefficients with 95% confidence bounds are summarized in Table R2.

**Table R2. The fitting coefficients with 95% confidence bounds for different mobility and mass selected particles with single, double and triple charges.**

|      | N (95% confidence bounds)    | $\mu$ (95% confidence bounds) | $\sigma$ (95% confidence bounds) |
|------|------------------------------|-------------------------------|----------------------------------|
| 80   | 1718.8 (1476.9, 1961.9)      | 53.9 (fixed at bound)         | 1.07 (1.05, 1.08)                |
|      | 637.1 (337.7, 936.5)         | 60.65 (fixed at bound)        | 1.06 (1.02, 1.10)                |
|      | 25.1 (906.7, 957.0)          | 70.9 (fixed at bound)         | 1.02 (0.07, 15.30)               |
| 100  | 5985.0 (5412.2, 6557.9)      | 59.5 (fixed at bound)         | 1.09 (1.08, 1.10)                |
|      | 1279.7 (746.7, 1812.6)       | 68.6 (fixed at bound)         | 1.08 (1.04, 1.12)                |
| 150  | 8361.9 (7997.1, 8726.7)      | 69.9 (69.5, 70.4)             | 1.11 (fixed at bound)            |
|      | 329.3 (-31.4, 690.0)         | 93.5 (78.6, 111.3)            | 1.11 (fixed at bound)            |
| 200  | 8086.6 (7864.7, 8308.4)      | 83.1 (82.9, 83.3)             | 1.099 (1.096, 1.102)             |
| 250  | 9835.5 (9603.9, 10067.1)     | 95.9 (95.7, 96.1)             | 1.095 (1.093, 1.098)             |

e) We have included the uncertainties in particles number concentration. We scanned the $d_{ae}$ distributions at least three times for each size particles. The average of the number densities was used and error bar is shown in the figures.

f) The $d_{ae}$ of particles with higher order of charges (q>3) can also be calculated using Eq. R5. The values

are very large (e.g. 78.5 nm for 80 nm particle with 4 charges) and close to upper limit of scanning range (35 nm~80 nm for 80 nm particles). Nonetheless, we think that particles with higher order of charges do not exist since the number density already tends to zero at the end of the scanning.

**Technical corrections:**

**7. *Comments and suggestions:***
***Line 18:*** *"the potential multiple charging effect"Elaborate.*

**Responses and Revisions:**
Thank you for the comment. The potential multiple charging effects of Tandem System, such as DMA-CPMA and DMA-AAC et al., have been discussed in the following text from Line 21 to Line 28: "Our results show that the ability to remove multiply charged particles mainly depends on particles morphology and instruments resolutions of DMA-CPMA system. Using measurements from soot experiments and literature data, a general trend in the appearance of multiple charging effect with decreasing size when selecting aspherical particles was observed. Otherwise, our results indicated that the ability of DMA-AAC in a static configuration to eliminate particles with multiple charges is mainly related to the resolutions of classifiers. In most cases, DMA-AAC can eliminate multiple charging effect regardless of the particle morphology in a static configuration, but multiply charged particles will be selected when decreasing resolution of DMA or AAC".

**8. *Comments and suggestions:***
***Line 19:*** *"remove"*
*Resolve?*

**Responses and Revisions:**
Thank you for your advice. We think "remove" should be used since we discuss DMA-CPMA in a static configuration.

**9. *Comments and suggestions:***
***Line 20:*** *"instruments setups of DMA-CPMA system"What is meant by instrument setups? Elaborate.*

**Responses and Revisions:**
We have changed this in the revised manuscript:
"Our results show that the ability to remove multiply charged particles mainly depends on particles morphology and instruments resolutions of DMA and CPMA".

**10. *Comments and suggestions:***
***Line 23:*** *"DMA-AAC can eliminate multiple charging effect"*
*This is not strictly correct as written since you can only "remove" the multiple charging artifact when usedin a static configuration. In a scanning mode, the contributions would be resolvable.*

**Responses and Revisions:**
Sorry we didn't make it clear that we used DMA-AAC in a static configuration. We have changed it in the revised manuscript:
"In most cases, DMA-AAC in a static configuration can eliminate multiple charging effect regardless of the particle morphology".

**11. *Comments and suggestions:***

*Line 24:* *"while particles with multiple charges can be selected when decreasing resolutions of DMA and AAC"*
*Confusing as written.*

**Responses and Revisions:**
We have changed this in the revised manuscript:
"In most cases, DMA-AAC in a static configuration can eliminate multiple charging effect regardless of the particle morphology, but multiply charged particles will be selected when decreasing resolutions of DMA or AAC".

**12.** *Comments and suggestions:*
*Line 25:* *"We propose that the multiple charging effect should be reconsidered when using DMA-CPMA or DMA-AAC system in estimating size and mass resolved optical properties in the field and lab experiments."*
*This statement is not clear as written. How should the effects be "reconsidered"?*

**Responses and Revisions:**
We have changed this in the revised manuscript:
"We propose that the potential influence of the multiple charging effect should be considered when using DMA-CPMA or DMA-AAC systems in estimating size- and mass-resolved optical properties in field and lab experiments".

**13.** *Comments and suggestions:*

*Line 35:* *"is the most commonly used size classifier"*
*If the DMA is the most commonly used classifier, why is only the original Knutson and Whitby reference provided?*

**Responses and Revisions:**
Park et al. (2008) reviewed the tandem techniques of DMA and other measurements. We have included it and other references of application of DMA in laboratory and field studies therein.

**14.** *Comments and suggestions:*

*Line 38:* *"particles are required to be pre-charged"In what sense do they need to be pre-charged?*

**Responses and Revisions:**
We have changed this in the revised manuscript:
"However, particles must be precharged when classified by a DMA or PMA because DMA and PMA classify particles based on electrical mobility and mass-to-charge ratio".

**15.** *Comments and suggestions:*

*Line 49:* *"This conclusion implies that it can hardly to achieved that all the multiply charged particles are effectively excluded for aspherical particles, especially for soot particles."*
*Grammar makes this sentence confusing.*

**Responses and Revisions:**
We have changed this in the revised manuscript:

"This conclusion implies that multiply charged particles cannot be effectively excluded for aspherical particles, especially for soot particles"

**16. *Comments and suggestions:***

***Line 60:*** *"dynamic shape factor (χ), can be inferred…"*
*The measured χ may depend upon the combination of instruments used. See (Yao et al., 2020) and potentially Table 2 here.*

**Responses and Revisions:**
We have added the sentence: "The derived $\rho_{eff}$ and χ depend upon the combination of instruments used, while the nonphysical values of χ and $\rho_{eff}$ for aspherical particles can be determined by the AAC-APM(Yao et al., 2020) and AAC-CPMA (Kazemimanesh et al., 2022)".

**17. *Comments and suggestions:***

***Line 89:*** *"The transfer function is an isosceles triangle with value of 1 at Zp * and going to 0 at (1 ±*

*βDMA)·Zp*."*

*I think it is important to mention that this translates to asymmetric distributions in Dm and mp since their relationship with Zp is nonlinear.*

**Responses and Revisions:**
We have added the sentence: "It translates to asymmetric in $d_m$ since the relationship between $d_m$ and $Z_p$ is nonlinear".

**18. *Comments and suggestions:***

***Line 106:*** *"is much simpler and more robust"*
*Elaborate.*

**Responses and Revisions:**
We have changed this in the revised manuscript:
"They considered the Taylor series expansion about the center of the gap ($r_c=(r_{2\_CPMA}+r_{1\_CPMA})/2$) instead of the equilibrium radius to avoid problems with the scenario in which the equilibrium radius does not exist. This method is much simpler and more robust"

**19. *Comments and suggestions:***

***Line 146:*** *"while the condensation particle counter"*
*Was only a single CPC used during the soot characterization experiments? In the previous sentences, the author make it seem like the measurements were made simultaneously and in parallel. Also, please include flow rates.*

**Responses and Revisions:**
Sorry for the misunderstanding. In this study, we only use one CPC. We have revised the manuscript and Figure 1. In the previous sentence, we have revised it to "For the soot characterization, the monodisperse

aerosol flow was switched between two parallel lines and fed into the CPMA (Cambustion Ltd., UK) and AAC (Cambustion, Ltd., UK, $Q_{sh}/Q_a = 10$); meanwhile, the condensation particle counter (CPC, Model 3756, TSI, Inc., USA, 0.3 L·min⁻¹) was switched between the CPMA and AAC. The particle mass ($m$) and aerodynamic diameter ($d_{ae}$) were determined by the scanning mode of the CPMA and AAC, while the CPC recorded their corresponding number concentrations at each setpoint. For each $d_m$, the $m$ and $d_{ae}$ distributions were measured three times. Between measurements of each $d_m$, the CPC was used behind the DMA, and the number size distribution of the generated soot particles was measured by SMPS to ensure that the generated soot particles did not change during the whole experiment".

**20. Comments and suggestions:**

**Line 147:** *"concentration"*
*Number density of particles. Concentration is assumed to have units of mole per unit volume.*

**Responses and Revisions:**

We have revised it to "number concentration".

**21. Comments and suggestions:**

**Line 148:** *"fitted to log-normal distribution"*
*Please include the equation that you used to fit your distribution since there are many ways to define the same relationship. Also, considering the shapes of the distributions shown in Fig. 5 and S2, why was a log-normal distribution utilized? The distributions appear symmetric, and they're plotted on a linear axis, so some justification is warranted.*

**Responses and Revisions:**

The equation $N\left(d_p\right) = \frac{N_0}{\sqrt{2\pi}\ln\sigma} \exp\left(\frac{-(\ln(d_p) - \ln(\mu))^2}{2(\ln\sigma)^2}\right)$ has been added. In Fig. 5 and S2, the distributions are plotted on a log axis, not a linear axis. This distribution is asymmetric on linear axis, hence we use log-normal distribution.

**22. Comments and suggestions:**

**Line 151:** *"density of PSL"Please enumerate.*

**Responses and Revisions:**

We have added the density value of 1050 kg m⁻³ for PSL.

**23. Comments and suggestions:**

**Line 153:** *"effect of particles selected by DMA-CPMA system, the $d_{ae}$ distribution of twice classified particles".*
*Please provide additional information about this portion of the procedure.*

**Responses and Revisions:**

We have changed this in the revised manuscript:
"To quantify the multiple charging effect of particles selected by the DMA-CPMA system, the soot particles were initially selected by the DMA-CPMA at different $d_m$ and the corresponding $m$. Then, the $d_{ae}$ distribution

of mobility and mass selected particles was obtained by stepping the AAC rotation speed of the cylinder with simultaneous measurement of the particle concentration at the AAC outlet using a CPC".

**24. *Comments and suggestions:***

***Line 167:*** *"we explain the transfer functions of DMA-CPMA and DMA-AAC utilizing the literature data of soot particles"*
*When the transfer functions were calculated, what range of parameters were used? And how exactly were the transfer functions solved? Iteratively or something else? If iteratively, what was the Δt for each step and the number of individual trajectories considered? These details need to be provided somewhere in the manuscript.*

**Responses and Revisions:**
This is a very useful suggestion. We have included the details in the revised manuscript: . "We explain the transfer functions of DMA-CPMA and DMA-AAC utilizing the literature data of soot particles (Pei et al., 2018). The $d_m$ and $m$ of the representative particles are 100 nm and 0.33 fg, respectively, and the corresponding $d_{ae}$ is 68.3 nm according to Eq. (19). In the calculation, the following parameter set was employed for the calculations: $d_m = 80$ nm, $Q_{CDMA} = 0.3$ L min$^{-1}$, $\beta_{DMA} = 0.1$, $m = 0.16$ fg, $Q_{CPMA} = 0.3$ L min$^{-1}$, $R_m = 8$, $d_{ae} = 68.3$ nm, $Q_{AAC} = 0.3$ L min$^{-1}$, $\beta_{AAC} = 0.1$".

The maximum and minimum values of $d_m$ for particles with $n$ charges can be derived and denote as $d_{m\,n,max}$ and $d_{m\,n,min}$, respectively. The maximum and minimum m of particles bearing single and double charges which are derived from Eq. 15 denote as $m_{1,\,max}$, $m_{1,min}$, $m_{2,max}$ and $m_{2,min}$. The maximum and minimum values of $d_{ae}$ that can be selected by AAC denote as $d_{ae,max}$ and $d_{ae,min}$. The transfer functions of DMA-CPMA and DMA-AAC were solved iteratively using logarithmically spaced $d_m$, $m$ and $d_{ae}$, which included 600 points, respectively. the ranges of $d_m$, $m$ and $d_{ae}$ used in the calculations were from $< d_{m1,min}$ to $> d_{m2,max}$, from $< m_{1,min}$ to $> m_{2,max}$, from $< d_{ae,min}$ to $> d_{ae,max}$, respectively.

**25. *Comments and suggestions:***

***Line 178:*** *"representing pre-exponential factor (ρf)"*
*Having the pre-exponential factor share a variable with effective density (ρeff) is confusing since I would expect them to share units when they do not; the units are (g nm$^{-Dfm}$) and (g nm$^{-3}$) for ρf and ρeff, respectively. Additionally, I recommend including a normalization factor in the Dm term to avoid having fractional units, e.g., $D_{fm} = 2.28$ will have ρf with units of g nm$^{-2.28}$.*

**Responses and Revisions:**
Equation 22 has been changed to $m = k_f \frac{(d)_m^{D_{fm}}}{nm}$, then $k_f$ will have the unit of "g". Equation 23 has been changed to $\log(m) = D_{fm} \log\left(\frac{d_m}{nm}\right) + \log(k_f)$.

**26. *Comments and suggestions:***

***Line 184:*** *"In the exemplary case, the derived Dfm of premixed flame generated soot particles was 2.28,"*
*What study does this refer to? Reference?*

**Responses and Revisions:**
The reference (Pei et al., 2018) has been added.

**27. *Comments and suggestions:***

*Line 189: "The DMA-CPMA system can eliminate the multiply charged particles only if the Dfm of particlesis larger than the slope of a line connecting line connecting (dm, m) = (dm2,min, m2,max)(dm1,,m1) (as PP0 shown in Fig. 2)."*
*The point (dm2, m2) appears to correspond to the actual mass and $d_m$ of a particle bearing 2 charges instead of the effective mass (m/2). This is unclear and has significant implications for the calculated transfer functions and resultant discussion since the effective mass is ultimately what affects instrument resolution.*

**Responses and Revisions:**

Thank you for your comment. We use the DMA-CPMA in a static configuration. When particles are selected by the DMA-CPMA with certain $d_m$ and m, the corresponding doubly charged particles of $d_{m2}$ and $m_2$ would also be selected.

The particles are shown in figure 2 in actual $d_m$ and $m$, but when we calculate the resolution of DMA and CPMA, the mobility and effective mass are used. The resolution of CPMA can be calculated by Eq. R2, where $m_1$ is the mass of singly charged particles which can be selected by the CPMA, i.e. effective mass.

**28. *Comments and suggestions:***

*Line 190: "line connecting (dm, m) = (dm2,min, m2,max)(dm1,,m1) (as PP0 shown in Fig. 2)."*
*I'm assuming that the location of (dm1, m1) is point P0 and is at the center of the q = 1 transfer function? This is not clear in the figure. There's an extra "," after dm1 in the text.*

**Responses and Revisions:**

We have removed the extra "," and the $PP_0$ has been marked out in Fig. 2.

**29. *Comments and suggestions:***

*Line 201: "are necessary to reduce the potential of multiply charged particles" By increasing the resolution of the measurement?*

**Responses and Revisions:**

We have changed this in the revised manuscript:
"When selecting particles of certain $d_m$ and $m$, by decreasing $Q_{CPMA}$, or increasing $\omega$ and $\beta_{DMA}$ i.e. by increasing the resolution of the measurement, the potential of multiply charged particles is reduced".

**30. *Comments and suggestions:***

*Line 205: "PP0 of 3.55 was derived when DMA-APM selects the same example soot particles" Compared to what?*

**Responses and Revisions:**

We have changed this in the revised manuscript:
"the slope of $PP_0$ of 3.55 was derived when the DMA-APM selects the same example soot particles from Pei et al. (2018) ($d_m$ of 100 nm and $m$ of 0.33 fg) with a $D_{fm}$ of 2.28".

**31. *Comments and suggestions:***

***Line 208:*** *"critical slope of PP0"*
*Z-axis in Figure is labelled as "The slope of PP0".*

**Responses and Revisions:**
We have revised it to "the critical slope of $PP_0$" in Fig. 3, consistent with the text.

**32. *Comments and suggestions:***

***Line 210:*** *"when the Dfm of particles is larger than the slope of PP0 which is represented as background color."*
*So, the color of the data point will be red shifted relative to the background when multiple charging is not affecting the output distribution? This is unclear and the rainbow color scheme isn't helping.*

**Responses and Revisions:**
The color scheme has been changed. The data points become square when $D_{fm}$ is smaller than the critical slope of PP0 in the background, i.e., the potential multiple charging effect may exist.

**33. *Comments and suggestions:***

***Line 213:*** *"Fig. 3, the dm, m and corresponding Dfm were taken from literature"*
*In the caption, the authors mention that the shapes correspond to the individual studies. Please elaborate which is which here?*

**Responses and Revisions:**
We have added more data to Fig. 3 and the references have been labeled in the figure.

**34. *Comments and suggestions:***

***Line 214:*** *"Generally, multiple charging effect can be avoided for DMA-CPMA to select soot particles with diameter larger than 200 nm.,"*
*I don't think the authors can conclude this without providing more data to support the claim. For fresh soot, my experience has been that multiple charging can be a problem at almost all Dm; e.g. see Figure S1 of (Radney et al. 2014). Also, there's a "," after the "." at the end of the sentence.*

**Responses and Revisions:**
Thank you for your comment. More data has been added to Fig. 3. Generally, we think that the $D_{fm}$ of soot particles is around 2.2~2.4. For these soot particles, we conclude that multiple charging effect can be avoided for DMA-CPMA to select particles with mobility diameter larger than 200 nm. We have revised the conclusion to "Generally, for soot particles with $d_m$ of 2.2~2.4, multiple charging effect can be avoided for DMA-CPMA to select soot particles with mobility diameter larger than 200 nm".
We think the conclusion is consistent with the study of Radney et al (2014). The main difference between these two studies is the instruments configuration. In Figure S1 in Radney et al. (2014), DMA was used in static configuration and particles with $d_m$ of 350 nm were selected. The APM was used in scanning mode and the mass distribution of mobility selected particles was determined. However, we used the static configuration of both DMA and CPMA in this study. Particles were sequentially selected by their mobility and mass. Usually we select a certain $d_m$ and the corresponding mode mass. In the bottom of Figure S1 in Radney et al. (2014), three regions are determined and singly charged are selected at mass < 4 fg. In the top of Figure S1 in Radney

et al. (2014), the mode mass of particles with $d_m$ of 350 nm is around 3.5 fg, as a result, no multiple charging exist when selecting these particles. Nonetheless, we demonstrate different result. The $d_m$ of lacy soot particles in Radney et al. is 1.79, which is smaller than common soot particles mentioned above. Applying particles with $d_m$ of 350 nm and $m$ of 3.5 fg to Figure 3 in this study, the $d_m$ of 1.79 is smaller than the critical PP0 of 2.02, i.e. multiple charging effect still exists after mobility and mass selection. We also apply other particles with smaller mass and the result shows that singly charged particles are selected at mass < 3 fg. In the bottom of Figure S1 (Radney et al., 2014), the slope of red line doesn't change obviously at mass of 3 fg maybe because the quantity of particles with higher charges is too small compared with particles with single charge since the peak of number concentration is at 3.5 fg. Moreover, we apply particles with $d_m$ of 350 nm and mass of 6 fg to Figure 3 and the result shows that doubly charged particles are selected by DMA-CPMA, which is consistent with the green line in the bottom of Figure S1 (Radney et al., 2014).

[Figure]

**RC-Figure 1:** The transfer functions of soot particles with $d_m$ of 350 nm and $m$ of (a) 3 fg, (b) 3.5 fg, (c) 4 fg and (d) 6 fg from Figure S1 in Radney et al. (2014).

**35.** *Comments and suggestions:*

*Line 215:* *"diameter"*
*To which diameter metric are you referring?*

**Responses and Revisions:**

It has been changed to "mobility diameter".

**36. Comments and suggestions:**

*Line 216:* *"eliminate"*
*Resolve?*

**Responses and Revisions:**

Thank you for your comment. We think it should be 'eliminate'. 'Resolve' is used when APM or CPMA is used in scanning mode and multiply charged particles can be identified from the mass distribution. We use the static configuration of DMA-CPMA and we are discussing if multiple charging can be avoided totally after mobility and mass selection. Hence, we think 'eliminate' is more proper.

**37. Comments and suggestions:**

*Line 220:* *"Therefore, the multiple charge effect could be avoided theoretically."*
*Measurements by just an AAC will avoid multiple charging. Multiple charging only becomes a problem again when the tandem measurement is a DMA or PMA.*

**Responses and Revisions:**

We have changed this in the revised manuscript:
"Measuring solely with an AAC will avoid multiple charging. However, AAC cannot constrain the properties of aspherical particles as monodisperse as DMA or CPMA classification (Kazemimanesh et al., 2022). Multiple charging becomes a problem when the tandem measurement is made with a DMA or PMA".

**38. Comments and suggestions:**

*Line 242:* *"resulting in the minimum $D_{fm}$ of 1.41, which is the case for most atmospheric aerosol particles."*
*Is this the black dashed line drawn as PP0? If so, please label in figure. Also, please differentiate these dashes from the vertical and horizontal ones.*

**Responses and Revisions:**

The label of the black dashed line drawn as $PP_0$ has been added. The figure has been replotted.

**39. Comments and suggestions:**

*Line 245:* *"sheath flow rate of classifier is restricted by the instrument design"It's also important to note that sheath flow restricts the maximum size ranges.*

**Responses and Revisions:**

Agree. We have added the sentence: "In addition, the maximum size ranges are also restricted by the sheath flow, so in some cases, a lower sheath flow rate is required to select larger particles.".

**40. Comments and suggestions:**

*Line 255:* *"Dfm was 2.28" What was the value of $\rho_0$?*

**Responses and Revisions:**

We have added the value of $\rho_0$ and this sentence has been revised to "The fitted value of $D_{\text{fm}}$ and $k_f$ were 2.28 and $7.49 \times 10^{-6}$, respectively".

**41. *Comments and suggestions:***

***Line 263:*** *"which suggested that multiply charged particles are still classified in this circumstance."*
*This suggests that the contributions from the multiply charged particles can't be resolved.*

**Responses and Revisions:**
Thank you for your advice. We have changed this in the revised manuscript:
"which suggests that the contributions from the multiply charged particles can't be eliminated". We use "eliminate" here instead of "resolve" because we use the DMA-CPMA in a static configuration.

**42. *Comments and suggestions:***

***Line 268:*** *"the multiply charged particles can be resolved in aerodynamic size distribution"*
*I disagree that the q = 1 and 2 can be resolved. Please provide more evidence to support this claim.*

**Responses and Revisions:**
We have changed this in the revised manuscript:
"Since the classification of AAC is different from DMA and CPMA, the aerodynamic size distributions of mobility and mass selected particles were characterized."

**43. *Comments and suggestions:***

***Line 270:*** *"PNSDae was fitted using log-normal distributions and three peaks which correspond to singly, doubly and triply charged particles were identified."*
*What does the peak at Dae < 40 nm correspond to? Please mention.*

**Responses and Revisions:**
Some small particles remaining in the AAC induced the peak at $d_{\text{ae}}$ <40 nm. These residual particles were measured even if the sample flow is filtered. This reason has been added in the text.

**44. *Comments and suggestions:***

***Line 296:*** *"Mie theory was used to calculate the theoretical absorption coefficient at the wavelength of 550 nm."*
*Mie theory probably isn't the "best" method to use here since soot particles are aspherical agglomerates. Realistically though, the Mie comparison is only being used to prove a point about the impact of multiple charging. So, in this instance any errors in the calculated optical properties are somewhat inconsequential. Some mention of this nuance should be mentioned.*

**Responses and Revisions:**
Agree. The following text has been added in the text:
"Mie theory is probably not the "best" method to use here since soot particles are aspherical agglomerates. Realistically, however, the Mie comparison is only being used to prove a point about the impact of multiple charging. Therefore, in this instance, any errors in the calculated optical properties are somewhat inconsequential."

**45. Comments and suggestions:**

*Line 301: "integral concentration for particles"*
*Integrated number density of particles. Concentration is assumed to have units of mole per unit volume.*

**Responses and Revisions:**

It has been changed to "number concentration".

**46. Comments and suggestions:**

*Table 2: It'd be interesting (but not necessary) to include a comparison of the derived shape factors (χ) for each method.*

**Responses and Revisions:**

The derived shape factors have been added. The shape factors calculated from two methods are consistent. We think the values of $\chi$ are not necessary in this study, so we didn't include them in the text.

| $d_m$(nm) | $m$(fg) | $d_{ae}$(nm) | $\rho_{DMA-AAC}$ (kg m$^{-3}$) | $\rho_{DMA-CPMA}$ (kg m$^{-3}$) | Deviation | $\chi_{DMA-AAC}$ | $\chi_{DMA-CPMA}$ | Deviation |
|---|---|---|---|---|---|---|---|---|
| 80 | 0.16 | 48 | 551.2 | 596.8 | 7.65% | 2.05 | 1.96 | 5.05% |
| 100 | 0.27 | 55 | 488 | 515.7 | 5.38% | 2.17 | 2.10 | 3.42% |
| 150 | 0.66 | 67 | 359. 1 | 373.5 | 3.86% | 2.49 | 2.43 | 2.34% |
| 200 | 1.28 | 82 | 303.2 | 305.6 | 0.77% | 2.62 | 2.60 | 0.44% |
| 250 | 2.17 | 96 | 262.8 | 265.2 | 0.90% | 2.71 | 2.70 | 0.50% |

**47. Comments and suggestions:**

*Figure 2: "Example of DMA-CPMA transfer function"*
*Transfer function for what? Soot?*

**Responses and Revisions:**

The reference has been added in the sentence. It has been changed to "Example of DMA-CPMA transfer function of flame-generated soot particles (Pei et al., 2018)".

**48. Comments and suggestions:**

*Figure 3: The coloration of this figure is very hard to understand.*

**Responses and Revisions:**

Agree. The figure has been changed.

**49. Comments and suggestions:**

*Figure 4: Z-axis is labelled "Transfer function of DMA-CPMA" and the caption says DMA-AAC.*

**Responses and Revisions:**

The Z-axis label has been changed to "Transfer function of DMA-AAC".

**50. Comments and suggestions:**

***Figure S1:*** *Having the Z-axis go all the way to zero is confusing.*

**Responses and Revisions:**
Agree. The figure has been changed.

**51.** ***Comments and suggestions:***

*Grammar mistakes.*

**Responses and Revisions:**
Grammar mistakes have been corrected.

**References:**

Kazemimanesh, M., Rahman, M.M., Duca, D., Johnson, T.J., Addad, A., Giannopoulos, G., Focsa, C. and Boies, A.M.: A comparative study on effective density, shape factor, and volatile mixing of non-spherical particles using tandem aerodynamic diameter, mobility diameter, and mass measurements. J. Aerosol Sci., 161, 105930, https://doi.org/10.1016/j.jaerosci.2021.105930, 2022.

Tavakoli, F., and Olfert, J. S.: Determination of particle mass, effective density, mass–mobility exponent, and dynamic shape factor using an aerodynamic aerosol classifier and a differential mobility analyzer in tandem, J. Aerosol Sci., 75, 35-42, https://doi.org/10.1016/j.jaerosci.2014.04.010, 2014.

Radney, J. G., R. You, X. Ma, J. M. Conny, M. R. Zachariah, J. T. Hodges, C. D. Zangmeister.: Dependence of soot optical properties on particle morphology: Measurements and modelcomparisons *Environ. Sci. Technol.* 48:3169-3176. https://doi: 10.1021/es4041804, 2014

Radney, J. G., Ma, X., Gillis, K. A., Zachariah, M. R., Hodges, J. T., and Zangmeister, C. D.: Direct Measurements of Mass-Specific Optical Cross Sections of Single-Component Aerosol Mixtures, Anal. Chem., 85, 8319-8325, https://doi.org/10.1021/ac401645y, 2013.

Yao, Q., Asa-Awuku, A., Zangmeister, C. D., and Radney, J. G.: Comparison of three essential sub-micrometer aerosol measurements: Mass, size and shape, Aerosol Sci. Technol., 1-18, https://doi.org/10.1080/02786826.2020.1763248, 2020.

---

## Author Comment (AC2)

**Response to Anonymous Referee #2**

We thank the reviewer for the constructive suggestions and comments concerning our manuscript entitled "Characterization of tandem aerosol classifiers for selecting particles: implication for eliminating multiple charging effect" (ID: amt-2021-436). Those comments are valuable and very helpful for improving our paper, as well as the important guiding significance to our studies. Below, we provide a point-by-point response to individual comment (Reviewer comments in italics, responses in plain font; page numbers refer to the AMTD version)

**1. *Comments and suggestions:***

***Line 88, Eq. (3)****: The equation for the transfer function of the DMA in non-diffusing caseis not given in the general form. Eq. (3) is only correct when aerosol inlet and aerosol sampling flow rates are equal (i.e. the DMA is operated in balanced flow mode), leadingto δ = 0 in the general form of DMA transfer function [Eq. (7) in Stolzenburg and McMurry (2008)]. This information should be noted in the paper.*

**Responses and Revisions:**
Thank you for your comment. We have changed it in the revised manuscript:
"Assuming that aerosol inlet and aerosol sampling flow rates are equal, the transfer function of DMA can be expressed as follows when particle diffusion is negligible".

**2. *Comments and suggestions:***

***Line 139:*** *The primary reference for the miniature inverted soot generator is as follows,which should be cited in the text: Kazemimanesh, M., Moallemi, A., Thomson, K., Smallwood, G., Lobo, P. and Olfert, J.S., 2019. A novel miniature inverted-flame burnerfor the generation of soot nanoparticles. Aerosol Science and Technology, 53(2), pp.184-195.*

**Responses and Revisions:**
Thank you for the reminder. We have added it in the text.
"Soot particles were generated by a miniature inverted soot generator (Argonaut Scientific Ltd., Canada) with a propane flow of 74.8 SCPM and an air flow rate of 12 SLPM. Although this operation setting is not in the open-tip flame regime, the flame is open-tip consistent with Fig. 2d in Moallemi et al. (2019). Detailed aerosol generation methods can be found in Kazemimanesh et al. (2019b) and Moallemi et al. (2019)"

**3. *Comments and suggestions:***

***Line 138-140:*** *To reproduce the experiments in this study, it is necessary to givedetails of propane and air flow rates used in the soot generator.*

**Responses and Revisions:**
Thank you for your reminder. We have changed it in the revised manuscript:
"Soot particles were generated by a miniature inverted soot generator (Argonaut Scientific Ltd., Canada) with the propane flow of 74.8 mL min$^{-1}$ and the air flow rate of 12 L min$^{-1}$. Although this operation setting is not in the open-tip flame regime, the flame is open-tip consistent with the Fig.2d in Moallemi et al. (2019)".

**4. *Comments and suggestions:***

***Lines 145-155:*** *For one or two mobility-selected particles, please add representativeplots for the measured spectral density of mass (dN/dlog $m_p$) and aerodynamic diameter (dN/dlog $d_{ae}$), either in the paper or in the supplementary material.*

**Responses and Revisions:**

The measured spectral density of mass (d$N$/dlog $m_{\mathrm{p}}$) and aerodynamic diameter (d$N$/dlog $d_{\mathrm{ae}}$) for particles with $d_{\mathrm{m}}$ of 150 nm and 250 nm have been added in the supplementary material.

**5. *Comments and suggestions:**

***Lines 198-199:*** *It states that the ability of DMA-CPMA to eliminate multiply charged particles depends on the resolutions of both DMA and CPMA; however, dependence onthe resolution of CPMA is not obvious from Eq. (25). Can the authors clarify this?*

**Responses and Revisions:**

Thank you for your comment. The resolution can be calculated by

$$R_m = \frac{m_1}{m_{1,max}-m_1} = \frac{2\pi B_{1,max} L_{CPMA} r_c^2 \omega^2 m_1}{Q_{CPMA}},$$

We have changed the Eq. (25) to

$$D_{fm} > PP_0 = \frac{\log(m_{2,max}/m_1)}{\log(d_{m2,min}/d_{m1})} = \frac{\log\left(2+\frac{1}{R_m(1+\beta_{DMA})}\right)}{\log\left(\frac{2}{(1+\beta_{DMA})}\frac{Cc(d_{m2,min})}{Cc(d_{m1})}\right)}$$

in the text and the relationship between the slope of PP$_0$ and the resolution is more obvious.

**6. *Comments and suggestions:**

***Line 201:*** *The sentence states that a larger $\beta$ is necessary to reduce the potential ofmultiply-charged particles, but it seems that Eq. (25) shows the opposite. Can the authors double-check this?*

**Responses and Revisions:**

Thank you for your comment. $(1 - \beta_{DMA})$ has been replaced by $(1 + \beta_{DMA})$. We have revised the Eq. (25) to

$$D_{fm} > PP_0 = \frac{\log(m_{2,max}/m_1)}{\log(d_{m2,min}/d_{m1})} = \frac{\log\left(2+\frac{1}{R_m(1+\beta_{DMA})}\right)}{\log\left(\frac{2}{(1+\beta_{DMA})}\frac{Cc(d_{m2,min})}{Cc(d_{m1})}\right)}$$

**7. *Comments and suggestions:**

***Line 303:*** *In Fig. 5c and all related figures, the unit used incorrectly for absorption cross section is Mm$^{-1}$, which is the unit for absorption coefficient (length$^{-1}$). The correctunit for absorption cross section should be m$^2$ particle$^{-1}$ (derived from absorption coefficient/particle number concentration).*

**Responses and Revisions:**

Thank you for the comment. The Y axis is the absorption coefficient $\alpha_{\mathrm{abs}}$ (Mm$^{-1}$). We have revised it in the Fig. 5c and 6c. The text has also been changed accordingly.

**8. *Comments and suggestions:**

***Lines 306-310:*** *It is not very clear to me why multiple charging effects due to the use of DMA-CPMA would affect the MAC or DRF of soot particles. As far as I understand, global climate models consider a specific (mostly constant) MAC value for black carbon particles to estimate their DRF, without regard for multiple charging. Unless the authorsare claiming that the MAC values used in current climate models*

*are grossly incorrect.*

**Responses and Revisions:**

Sorry we didn't make it clear. Previous studies used DMA-APM or DMA-CPMA to investigate the mass-specific MAC (Radney et al., 2013; Zangmeister et al., 2018). Our study tried to illustrate that multiple charging effect can affect classification of DMA-CPMA and the measured value of MAC. If using the measured MAC to evaluate the DRF of fresh soot particles, it can cause uncertainties.

**9. *Comments and suggestions:***

*Introduction and discussion section: There are a few recent studies that have looked at tandem measurements of mobility diameter, mass, and aerodynamic diameter to studythe effective density and shape factor of spherical and non-spherical particles. These studies have used a combination of DMA, AAC, and APM or CPMA and, in my view, are relevant to this paper and should be mentioned in the introduction and their results discussed where necessary:*

*Yao, Q., Asa-Awuku, A., Zangmeister, C.D. and Radney, J.G., 2020. Comparison of three essential sub-micrometer aerosol measurements: Mass, size and shape. AerosolScience and Technology, 54(10), pp.1197-1209.*

*Kazemimanesh, M., Rahman, M.M., Duca, D., Johnson, T.J., Addad, A., Giannopoulos, G., Focsa, C. and Boies, A.M., 2022. A comparative study on effective density, shape factor, and volatile mixing of non-spherical particles using tandem aerodynamic diameter, mobility diameter, and mass measurements. Journal of Aerosol Science, 161,p.105930.*

**Responses and Revisions:**

Thank you for your suggestion. We have added these literatures in the introduction (Line 72 and Line 73) and discussion (Line 265)

**Editorial and technical corrections**

**10. *Comments and suggestions:***

**Line 272:** *I cannot find Eq. (30) in the paper.*

**Responses and Revisions:**

Sorry for our mistake. We have revised it to Eq. (16).

**11. *Comments and suggestions:***

**Line 301:** *Change the sentence to "Subsequently, absorption cross section, σabs, was derived using the absorption coefficient and total number concentration of particles with different charging states."*

**Responses and Revisions:**

We have changed it in the revised manuscript:

"Subsequently, absorption coefficient, $\alpha_{abs}$, was derived using the Mie theory and PNSD$_{ve}$ of particles with different charging states.".

**12. *Comments and suggestions:***

**Lines 308-310:** *This sentence is written very poorly (huge amount? huge error?). Please rephrase this sentence and avoid ambiguous adjectives.*

**Responses and Revisions:**

We have changed it in the revised manuscript:

"A large amount of 70 nm -90 nm soot particles was emitted from a diesel engine (Wierzbicka et al., 2014), and neglecting the multiple charging effect in the measurement of mass-specific MAC on this size range will result in significant bias in the estimation of radiative forcing of automobile-emitted soot particles".

**13. *Comments and suggestions:***

*Line 312: Reference to Table 3 should be given in the earlier paragraph (perhaps in line 302).*

**Responses and Revisions:**

Thank you for the comment. Table 3 now is given in line 374.

**14. *Comments and suggestions:***

*Grammar mistakes.*

**Responses and Revisions:**

Grammar mistakes have been corrected.

---

## Referee Report (RR1)

**Review of:**

**Characterization of tandem aerosol classifiers for selecting particles: implication for eliminating multiple charging effect**

Yao Song, Xiangyu Pei, Huichao Liu, Jiajia Zhou, Zhibin Wang

**General comments:**

The manuscript by Song et al. examines the problem of multiple charging when using two tandem systems: a differential mobility analyzer (DMA) and centrifugal particle mass analyzer (CPMA) and a DMA and an aerodynamic aerosol classifier (AAC). These systems are collectively referred to as a DMA-CPMA and DMA-AAC, respectively. The manuscript analyzed the transfer function of these systems theoretically, under static conditions, using data from the literature and then analyzed the presence or absence of multiply charged particles when working with flame generated soot. They find that the ability of the DMA-CPMA and DMA-AAC to remove multiply charged particles depends upon the resolution of the classifiers, the morphology of the particles and the mobility diameter ($D_m$) under investigation; smaller Dm are more likely to be contaminated with multiply charged particles.

The manuscript has been greatly improved since the first review and can be accepted pending a minor revision.

**Specific AMT review criteria:**

**1.** Scientific significance: The manuscript represents a substantial contribution to understanding the limits of nanoparticle separation when using a tandem DMA-CPMA or DMA-AAC. Rating: good

**2.** The scientific approaches and applied methods are valid. While the discussion of the results has been improved since the initial submission, there are places where further discussion would improve manuscript clarity. Rating: good

**3.** The presentation of the results and conclusions has improved greatly since the initial submission, but there are still many places where improvements to clarity would be beneficial to the reader. Rating: good

This manuscript is publishable after minor revision since the topic is of interest to the community.

**Specific comments:**

**1.** The authors put quite a few equations inline in the text. It may improve readability to have them offset like the remainder, especially when the values are used multiple times throughout.

**2.** The authors need to double check that they are defining each variable to represent a single quantity. For example, m is used as mass in general or as the mass axis in a distribution and as the mode mass of the particle distribution.

**3.** It is still not clear to me how the charge (when > 1) is being included in the calculated transfer functions (*e.g.*, paragraph starting on line 230 and Figure 2) since the authors are presenting everything as absolute

mass instead of effective mass. Is the charge term somehow "baked into" the calculated transfer functions? Some clarification is warranted.

In the author response document, their response to technical comment 27 was "The particles are shown in figure 2 in actual dm and m, but when we calculate the resolution of DMA and CPMA, the mobility and effective mass are used. The resolution of CPMA can be calculated by Eq. R2, where m1 is the mass of singly charged particles which can be selected by the CPMA, i.e. effective mass."

Including this would be sufficient.

**4.** In the reviewer response, the authors include the following paragraph: "In our study, we also use scanning mode of CPMA after DMA selection to determine the mode mass of the selected particles, then we use the tandem setup of DMA and CPMA both at fixed mode to select particle at fixed mobility and mode mass i.e. DMA and CPMA are used in a static configuration, no scanning for either instrument is used. In Figure 5a, DMA-CPMA is set to select singly charged particles with dm of 80 nm and m of 0.16 fg, while the doubly charged particles with dm of 119.3 nm and m of 0.32 fg will also be selected and the transfer function is presented as upper right region. Soot particles curve (red line) goes through the upper-right region which doubly charged particle can penetrate (dm of 113 nm~118 nm, m of 0.35 fg~0.39 fg). As a result, we conclude that multiple charging effect still exists when DMA-CPMA select soot particles with dm of 80 nm and m of 0.16 fg."

This is one of the best and most concise paragraphs describing the DMA-CPMA data in the manuscript. I highly recommend integrating it into the body of the manuscript.

**5.** In the reviewer response, the authors provide a discussion on the ordering of the DMA-AAC and the impact on static measurements in response to my comment 5. (Starts at "This population has one physical size (dae) but the dm"). This is an excellent discussion and I strongly recommend integrating it into the manuscript.

**Technical corrections:**

**Line 15:** "specific size or mass."

Size is a nebulous term with respect to mobility ($D_m$) or aerodynamic ($D_{ae}$) diameter. Maybe rewrite to "specific mobility diameter, aerodynamic diameter or mass, respectively."?

**Line 17:** "demonstrate"

Calculate?

**Line 18:** "in static configurations."

For flame generated soot particles.

**Line 22:** "resolutions"

How is resolution defined in this sense?

**Line 24:** "Otherwise, our results indicate…"

Repeated sentence from line 21.

**Line 36:** "size dependence"

size-dependent

**Line 44:** "particles must be precharged"

Unclear what this means since, depending upon particle size, most aerosols possess a net charge. Instead, I think the authors are referring to bringing the particles to a known charge distribution by passing through a charge neutralizer or similar?

**Line 45:** "mass-to-charge ratio,"

mass-to-charge ratio, respectively,

**Line 55:** "of DMA-APM"

of the DMA and APM

**Line 64:** "is that no charging process is needed"

Should this be "the charge state of the particles does not need to be known"?

**Line 71:** "(Johnson et al., 2018)."

This is not a peer-reviewed manuscript but rather a conference presentation.

**Line 73:** "APM(Yao"

Missing a space between APM and (Yao

**Line 97:** "elemental"

elementary

**Line 98:** "Zp*"

$Z_p$* needs to be defined. Also, the symbol used for Zp* on line 98 and in the table of symbols is different than that used in Eq. 2.

**Line 102:** "DMA, respectively."

DMA electrodes, respectively.

**Line 109:** "$d_{m\,n,max}$ and $d_{m\,n,min}$"

Is there supposed to be a space or a , between m and n?

**Line 111:** "Zp"

Z should be italic.

**Line 113:** "The construction of the CPMA is similar to the APM,"

How so? No discussion on the construction of the APM has been provided.

**Line 123:** α and β, in this usage, should be explicitly defined to avoid confusion with other quantities (e.g., $\alpha_{abs}$, $\beta_{DMA}$, etc.)

**Line 136:** "Assuming a plug flow,"

Delete "a".

**Line 139 and 140:** Formatting on functions should be consistent throughout. Min, max, exp should all be formatted similarly. Also, min (and max, preferably) should be explicitly defined after use as the minimum (and maximum) of the quantities in the brackets since min is also the abbreviation for minutes.

**Line 161:** "τ"

τ should be italic.

**Line 162:** "denote"

denoted

**Line 165 and 166:** "SCPM" and "SLPM" are non-standard units and should be explicitly defined; *e.g.* SLPM (standard L per min, flow in L $min^{-1}$ converted from ambient to T = and P = ).

**Line 171 and 173:** "Qsh/Qa = 10"

the $\beta_{DMA}$ and $\beta_{AAC}$ formulation should be used for consistency throughout.

**Line 171:** "monodisperse"

This is not technically correct as written and the authors should remind the reader that the stream is monodisperse in $Z_p$.

**Line 174:** "The particle mass (m) and aerodynamic diameter ($d_{ae}$) were determined by the scanning mode"

This is not exactly true as written. The distributions of particle number density as a function of particle mass and aerodynamic were measured and mode particle mass was then determined from a fit of that distribution.

**Line 175:** "CPMA and AAC, while"

CPMA and AAC, respectively, while

**Line 178:** "SMPS"

Needs to be defined.

**Line 179:** "soot particles did not change during the whole experiment."

It is unclear to what change the authors are referring.

**Line 180:** "The m and $d_{ae}$ distributions were fitted to log- normal distributions; thus, the modes m and $d_{ae}$ for the mobility-selected particles were determined"

According to this sentence, m and $d_{ae}$ represent multiple quantities, both the axis and the modal value. Separate variables should be used for each quantity to avoid confusion.

**Line 181:** "equation of log-normal distribution used in this study is expressed as"

This equation only applies to $N(d_p)$ and not to the m or $d_{ae}$ fits that the authors are referring.

**Line 183:** "where σ is the geometric standard deviation and μ is the geometric mean"

Did the distributions of m and Dae have the same σ and μ? That is what is implied.

**Line 194:** "electrical diameter"

electrical mobility diameter

**Line 196:** "ρeff"

The mathematical relationship for $\rho_{eff}$ has not been defined anywhere.

**Line 202:** "where Φ and Ω are the transfer functions of each classification system expressed by subscripts."

Should be "where Φ and Ω are the transfer functions of the combined and individual classification systems expressed by subscripts, respectively."

**Line 206:** "dm = 80 nm"

Why is the dm, m and dae different in this sentence than in the previous?

**Line 208:** "included 600 points, respectively"

600 points each?

**Line 210:** ">m2,max, from" How far < or > the respective values were investigated?

missing "and"

**Line 213:** "The DMA-CPMA transfer function is calculated in log(dm)-log(m) space, as shown in Fig. 2."

"DMA-CPMA transfer function ($\Phi_{DMA-CPMA}$)". What is the transfer function calculated for?

**Line 215 and 216:** "*nm*"

nm should not be italicized when used as a unit.

**Line 217:** "and smaller than 3 for aspherical particles"

Dfm can be larger than 3 for particles that are non-spherical at small $D_m$ and approach spherical as $D_m$ increases.

**Line 219:** "Under this specific operation condition"

What specific operating condition? Please explain.

**Line 220:** "spherical particle population (black line)"

Is this a "theoretical" spherical particle population? What would be the $\rho_{eff}$ of these particles?

**Line 220:** "classification region"

What is the classification region? Please elaborate.

**Line 231:** "than the slope of a line connecting (dm, m) = (dm2,min, m2,max)(dm1,m1) (as PP0 shown in Fig. 2)"

PP0 is not clearly shown in Fig. 2 and this was **the** source of my confusion. From the figure, it appears that PP0 is drawn as the Dfm = 2.28 line. So, while the Dfm = 3 discussion seems reasonable, the Dfm = 2.28 does not. In contrast, for Figure 4 the $PP_0$ line is clearly visible making the discussion much easier to understand. My recommendation is to either switch the ordering of the DMA-CPMA and DMA-AAC sections or to add an additional plot to Figure 2 at a larger dm1 where PP0 is clearly visible.

**Line 236:** ". Accordingly, the ideal condition…"

Under static operation at this set point

**Line 240:** "Eq. (26) gives instructions in actual operation"

It is unclear how Eq. 26 gives instructions.

**Line 244:** "and the slope of PP0 derived from the actual condition"

This is unclear as written. How is the slope of $PP_0$ derived from actual conditions? Weren't the transfer functions from which $PP_0$ is determined theoretically calculated?

**Line 246:** "According to the theoretical calculation described in Kuwata (2015), the slope of PP0 of 3.55 was derived when the DMA-APM selects the same…"

Kuwata did not calculate a $PP_0$, so it is unclear where this value is coming from. What was the slope of $PP_0$ for the DMA-CPMA for reference? Should the value of 3.55 be 2.55? If not, are the authors claiming that the DMA-APM would be unable to separate spherical particles (Dfm = 3) under these theoretical conditions? I completely agree that the DMA-APM is more susceptible to multiple charging, but this comparison to the APM needs to be clarified or removed.

**Line 251:** "Rm = 8"

What is Rm? This is the first instance of it in the manuscript.

**Line 254:** "the slope"

the critical slope

**Line 251:** "contour lines in Fig. 3"

How were these contours calculated? From Eq. 26? If so, how was dm2 determined?

**Line 259:** "mobility diameters larger than 200 nm, while it fails to eliminate …"

As shown by the circles and squares in Figure 3.

**Line 263:** "to charge aerosol particles"

To Boltzmann distribution or a known charge state?

**Line 264:** "AAC cannot constrain the properties of aspherical particles as monodisperse as DMA or CPMA classification"

Unclear as written.

**Line 267:** "selecting the same representative particles"

The same as what? Please give values.

**Line 268:** "aspherical particles can be expressed as follows"

The log(Dae) is expressed on the next line. Not aspherical particles.

**Line 273:** "the same particles"

Please give values as a reminder.

**Line 276:** "are in parallel for the DMA-AAC"

Unclear as written. I think the authors are referring to the fact that transfer function will have the same $D_{ae}$ and different $D_m$?

**Line 277:** "the example setups"

What example setups?

**Line 290:** "which is the case for most atmospheric aerosol particles."

What is the case? This $D_{fm}$ is smaller than for most aerosols.

**Line 292:** "is required"

"may be required"

**Line 295:** "When increasing $\beta_{AAC}$ to 0.3"

Increasing is a misnomer here since an increase in $\beta_{AAC}$ is a decrease in resolution. Please remind the reader of this distinction.

**Line 302:** "the corresponding dae and m were determined using the AAC and CPMA scan modes"

This isn't exactly true as written. The distributions of number density as a function of $D_{ae}$ and m were determined by the scans. These distributions were then fit to a log-normal to determine the modal values and from these values the $\rho_{eff}$ were determined.

**Line 304:** "measured spectral density"

Measured distributions?

**Line 305:** "The results are summarized in Table 2."

How were the uncertainties in Table 2 determined? 1σ standard deviation of multiple measurements? Or something else? Please describe.

**Line 306:** kf has units of mass.

**Line 308:** "two methods"

Which two methods?

**Line 308:** "the deviation"

What deviation?

**Line 316:** "the corresponding transfer function"

DMA-CPMA transfer function?

**Line 317:** "The particle population"

Shown by the red $D_{fm}$ = 2.28 line?

**Line 320:** "particles number aerodynamic size distribution"

Should be "particles number density aerodynamic size distribution"

**Line 324:** "The mean dae values"

For particles with dm = 80 nm?

**Line 327:** "In contrast, …"

In contrast to what?

**Line 332:** "PP0"

Subscript 0.

**Line 363:** "26.7±3.0%"

What is the unit on 26.7? Is it %? If so, should be written as (26.7±3.0) % or 26.7 % ± 3.0 % to avoid the confusion of 3.0 % being a relative value and the absolute being 26.7 ± 0.8. Other values in this paragraph need to be similarly corrected.

**Line 387:** "Under the same setups"

Same as what?

**Table A1:** Thank you for including this table. But, can you please sort values alphabetically to assist the reader in locating values and include the corresponding units where appropriate.

**Table 2:** "*M* (fg)"

Is this the modal mass from the log-normal fit? This is the wrong symbol.

**Table 3:** fN and fabs are not mathematically defined in the body of the manuscript. Can the authors provide the calculated MAC for each size and the overall? And should "MAC overestimation" have a units associated with it?

**Figure 2 caption:** Please note that this plot is in log-log space as a reminder for the reader. It is not clear from just looking at the axes.

**Line 533:** "DMA and CPMA."

DMA and CPMA, respectively.

**Figure 3:** What are the minimum values on the dm and m axes?

**Line 537:** "The contour lines denote the slope"

Critical slope?

---

## Author Response (AR3)

**Response to James Radney**

We thank the reviewer for the constructive suggestions and comments concerning our manuscript entitled "Characterization of tandem aerosol classifiers for selecting particles: implication for eliminating multiple charging effect" (ID: amt-2021-436). Those comments are valuable and very helpful for improving our paper, as well as the important guiding significance to our studies. Below, we provide a point-by-point response to individual comment (Reviewer comments in italics, responses in plain font; page numbers refer to the AMTD version; Tables used in the response are labeled as Table S1, Table S2,…, figures used in the response are labeled as Fig. R1, Fig. R2,…)

 *General Comments and suggestions:*

*The manuscript could represent a substantial contribution to scientific progress within the scope of Atmospheric Measurement Techniques and this is specifically highlighted by the derivation of Equations 25 and 27 for the limiting cases for the complete separation of multiply charged particles. In my opinion, the authors need to refer to these cases as something better than PP0 and make mention them in the abstract since this seems to be the most important takeaway. However, the clarity of the presented results is lacking, and it appears the presented transfer functions are not correct, which severely detracts from the manuscript. Rating: poor*

*The scientific approach and applied methods seem valid, but not necessarily the calculations, with limited discussion of the results but in an appropriate and balanced way. Rating: fair*

*The presentation of the scientific results and conclusions needs significant improvement. The use of English language is not fluent or precise in places and this distracts from the information the authors are trying to convey. Additionally, there are sections of the discussion that should be significantly expanded, and this expansion should aid in clarity. Rating: fair*

**Responses and Revisions:**

Thank you for the advice.   In summary, we are very sorry for the misunderstanding due to the poor expression. The revisions could be found in individual comment in details.

For our main contribution, Equations 26 and 28 demonstrate the relationship between $D_{fm}$ and the instrument resolutions when using DMA-CPMA to eliminate multiple charging effect. We have included a brief description of these equations in the abstract: "We propose an equation that constrains the resolutions of DMAs and CPMAs to eliminate the multiple charging effect when selecting particles with a certain mass–mobility relationship using the DMA-CPMA system. The equation for the DMA-AAC system is also derived"

For transfer functions of tandem system of DMA-CPMA and DAM-AAC, we think our calculation is credible, while the expression leads to the misunderstanding. We calculated the transfer functions of DMA-CPMA and DMA-AAC both in a static configuration. The transfer function of DMA-CPMA was derived by multiplying the transfer functions of DMA and CPMA. The transfer function of DMA-AAC was derived by multiplying the transfer functions of DMA and AAC. We have reorganized our language in the revised manuscript. The details could be found in Comments 4.

Our English writing has been promoted by native speaker.

**1. *Comments and suggestions:**

*I'm really struggling to understand the figures because of the rainbow color scale used and I strongly recommend using a different color scheme. Additionally, this color scheme is not visually accessible to all readers; e.g. (Nuñez, Anderton and Renslow 2018).*

**Responses and Revisions:**

We have revised Fig.3 to contour plot. It is more straightforward to compare the reported $D_{fm}$ and critical slope PP$_0$ values. The selected particles with multiple charging effect are resented as squares. The colors are used to distinguish literature data. Moreover, we have included the black and white version in the supplement (Fig. S4). According to the Color BLIndness Simulator, this figure should be readable for all readers.

[Figure]

**Figure 3: Variations of the slope of PP$_0$ as a function of classified $d_m$ and $m$.** The following parameter set was employed for the calculations: $\beta_{DMA} = 0.1$, $Q_{CPMA}=0.3$ L min$^{-1}$, $R_m = 8$. The contour lines denotes the slope of PP$_0$ with values labeled on them. The data points are soot particles measured in literatures (Park et al., 2003; Rissler et al., 2013; Tavakoli et al., 2014; Ait Ali Yahia et al., 2017; Dastanpour et al., 2017; Forestieri et al., 2018; Pei et al., 2018; Kazemimanesh et al., 2019) and generated in this study (See details in section 3.2).The $D_{fm}$ of these data points are listed in the legend. The data points become square when the $D_{fm}$ is smaller than the critical slope of PP0 in the background, i.e. the potential multiple charging effect may exist.

In the revised manuscript, the figures (Fig. 2, Fig. 4, Fig. 5a and Fig 6a) with rainbow color scale are kept to compare with the previous study representing the DMA-APM transfer function (Kuwata et al., 2015), we hence kept them likewise.

**2. *Comments and suggestions:***

*Uncertainties are missing on reported values throughout.*

**Responses and Revisions:**

We have added the uncertainties in Table 2 and Table 3.

**3. *Comments and suggestions:***

*The authors investigated the pairwise combinations of DMA-CPMA and DMA-AAC but some mention of their expectations when utilizing other orderings (AAC-CPMA, AAC-DMA, DMA-CPMA) is needed. This is especially true for an AAC-DMA since the ratio of the β's may depend upon this ordering. (i.e. does it matter whether the transfer function of the first instrument is narrower/similar/wider than the second?)*

**Responses and Revisions:**

This is a very good suggestion. The transfer function of AAC-DMA is calculated with the transfer functions of two classifiers (Tavakoli and Olfert, 2014):

$$\Phi_{AAC-DMA} = \Omega_{DMA}\Omega_{AAC} , \hspace{4cm} \text{R1}$$

we think the transfer function of DMA-AAC in a static configuration is independent on their ordering. An example can be found in comment 5.

As for DMA-CPMA, the resolution of CPMA is calculated assuming that all the particles have exactly the same mobility:

$$R_m = \frac{m_1}{m_{1,max}-m_1} = \frac{2\pi B_{1,max}L_{CPMA}r_c^2\omega^2 m_1}{Q_{CPMA}} \hspace{3cm} \text{R2}$$

where $m_1$ and $m_{1,max}$ are the nominal mass and the maximum mass that can be selected by CPMA, respectively. $B_{1,max}$ is the mobility of particles with mass of $m_{1,max}$. We assume that $B_{1,max}$ is equal to the mobility of nominal particles. This assumption is valid when the particles are mobility selected before fed into the CPMA, so the ordering of DMA-CPMA can't be changed.

We didn't include the transfer function of AAC-CPMA or CPMA-AAC in our study. We attempted to calculate the transfer function of AAC-CPMA, but we found it was difficult since particle mobility should be provided first to connect the calculations of AAC and CPMA. The relationship between $\tau$ selected by AAC and $m$ selected by CPMA is shown as follows (Yao et al., 2020):

$$\tau = m_{sp}B = \chi m B, \hspace{5cm} \text{R3}$$

in which $m_{sp}$ is mass of spherical particles. CPMA determines real mass of particles without any assumption of $\chi$. Mobility cannot be derived from Eq. R3 since $\chi$ is unknown.

**4. *Comments and suggestions:**

*It appears that the calculated transfer functions do not include the effect of the mass to charge ratio on the resolution of the CPMA in Figures 2, 5 and S2; effective masses, not absolute masses are the key quantity being measured.*

*For example: in Figure 5a, the authors state that $D_m = 80$ nm, $m = 0.16$ fg and $D_{fm} = 2.28$. Solving the $\underline{D}_{fm}$ relationship yields $\rho_f = 7.3 \times 10^{-6}$ fg nm$^{-2.28}$. At $D_m = 120$ nm, $m \approx 0.40$ fg which agrees with the data shown in the figure. Unfortunately, because $q = 2$, the effective mass would be a factor of 2 lower and should be $\approx 0.2$ fg. It appears that a similar error is present in Figures 2 and S2.*

**Responses and Revisions:**

Thank you for the comment. We thought that we did not clearly clarify our instruments configuration. Usually, the tandem setup of DMA-CPMA is that DMA firstly selects a fixed mobility, while CPMA is used in scanning mode to derive the corresponding mass . For example, in Figure 2 in Radney et al. (2013), they fixed the DMA to select particles at $d_m$ of 200 nm and particles bearing higher-order charges were also selected because of multiple charging effect. The mobility-selected particles included particles bearing 1, 2 and 3 charges with $d_m$ of 200 nm, 321 nm and 434 nm, respectively. The corresponding $m$ of 7.4 fg, 30.7 fg and 75.8 fg can be calculated using $d_m$ and effective density, respectively. The scanning mode of APM was used downstream to resolve multiple charging effect of these mobility-selected particles. The APM selects particles according to their mass-to-charge ratio. As a result, the effective masses measured by APM were 7.4 fg, 15.3 fg, 25.3 fg, respectively, which is shown as grey circles in Figure 2 (Radney et al., 2013)

In our study, we also use scanning mode of CPMA after DMA selection to determine the mode mass of the selected particles, then we use the tandem setup of DMA and CPMA both at fixed mode to select particle at fixed mobility and mode mass i.e. DMA and CPMA are used in a static configuration, no scanning for either instrument is used. In Figure 5a, DMA-CPMA is set to select singly charged particles with $d_m$ of 80 nm and $m$ of 0.16 fg, while the doubly charged particles with $d_m$ of 119.3 nm and $m$ of 0.32 fg will also be selected and the transfer function is presented as upper right region. Soot particles curve (red line) goes through the upper-right region which doubly charged particle can penetrate ($d_m$ of 113 nm~118 nm, $m$ of 0.35 fg~0.39 fg).

As a result, we conclude that multiple charging effect still exists when DMA-CPMA select soot particles with $d_m$ of 80 nm and $m$ of 0.16 fg.

**5. *Comments and suggestions:**

*Figure 4 appears to correspond to utilizing an AAC-DMA combination rather than a DMA-AAC as is discussed in the text. In the DMA-AAC, you'd get two populations of particles in $D_{ae}$-space since the distributions of particles have equivalent $Z_p$ exiting the DMA but are physically different sizes with different q. In the AAC-DMA, you'd only have one population in $D_{ae}$-space, the AAC selects one physical size, and then that distribution would have multiple charge states exiting the DMA.*

*AND*

***Line 226 to end of paragraph:*** *"In order to simulate the transfer function of DMA-AAC selecting the same particles as that used in calculations of DMA-CPMA. The corresponding $d_{ae}$…"*

*See Specific Comment 5.*

**Responses and Revisions:**

Yes, we agree that AAC selects only one population. ns of nonspherical particles as monodisperse as DMA or CPMA classification. The AAC selects relaxation time, not directly aerodynamic diameter. The relationship between the relaxation time and mobility diameter can be expressed as follows,

$$\tau = \frac{\rho_{eff} d_m^2 C_c(d_m)}{18\mu},$$
R4

which indicates the AAC selects monodispersed particles when particles have the same effective density. In our study, the effective density of soot particles decreases with increasing $d_m$. Particles with different $d_m$ but the same relaxation time will be selected.

We think the transfer functions of DMA-AAC or AAC-DMA are identical regardless of the order of DMA and AAC. For example, we use AAC-DMA to select particles with $d_{ae}$ of 68 nm and $d_m$ of 100 nm. In figure 4a in this study, the transfer function of AAC is the region between the horizontal lines of $d_{ae,max}$ (75 nm) and $d_{ae,min}$ (63 nm). The soot particles population (red line) goes through this region will be selected by AAC. The mobility diameter distribution of these relaxation time selected particles is around 80 nm to 120 nm. Then the DMA is fixed to select particles with $d_m$ of 100 nm, the particles with double charges and the same mobility ($d_m$ of 150 nm) have been excluded by AAC. As a result, AAC-DMA select monodispersed particles with $d_{ae}$ of 68.3 nm and $d_m$ of 100 nm. In Fig. 4b, the resolution of AAC is lower and transfer function of AAC is broader than that in Fig. 4a. The soot particles population (red line) goes through the transfer function region between the horizontal lines at $d_{ae}$ of $d_{ae,max}$ (50 nm) and $d_{ae,min}$ (86 nm). The mobility diameter distribution of these relaxation time selected particles is very wide from less than 80 nm to about 158 nm. Then these relaxation time selected particles were charged and selected by DMA at $d_m$ of 100 nm, singly charged particles with $d_m$ of 95 nm~106 nm and doubly charged particles with $d_m$ of 142 nm~158 nm will be selected.

If we use the DMA-AAC, the particles are selected by DMA first. For example, in Figure 4b, the transfer function of DMA is shown as two vertical regions which particles with single and double charges can penetrate. The soot particles (red line) goes through it and two populations of soot particles with mode $d_m$ of 100 nm and 150 nm will be selected. The corresponding $d_{ae}$ distributions of these singly and doubly charged particles are 66 nm~70 nm and 81 nm~87 nm. These mobility-selected particles are selected at $d_{ae}$ of 68.3 nm by AAC and the transfer function of AAC shows that particles with $d_{ae}$ of 50 nm~86 nm can penetrate. As a result, singly charged particles with $d_{ae}$ of 66 nm ~70 nm and doubly charged particles with $d_{ae}$ of 81 nm ~86 nm can be selected.

As a summary, the transfer functions of DMA-AAC and AAC-DMA in a static configuration are the same no matter the ordering of DMA and AAC.

**6. *Comments and suggestions:**

*The authors fit the distributions of aerodynamic diameter ($D_{ae}$) utilizing multiple log-normal distributions, but it doesn't seem to me that they have the resolution to constrain these values even though we know that multiple charges are present a priori. More discussion of the fitting routine is necessary. For example, in Fig. 5b, a) How does the fit compare to using just a single log-normal distribution? Or a single Gaussian or summation of multiple Gaussian distributions? b) Were the central values of $D_{ae}$ constrained prior to the fit? Or were they allowed to float? c) Were the widths of the distributions constrained in any way prior to the fit? d) What are the magnitude of the uncertainties in each of the fit coefficients? e) Were the uncertainties in particle number densities included in the fits? f) The peak of the distribution is significantly underfit. Is it possible that more than q = 1 and 2 are contained within the primary peak and what was identified as q = 3 is 4 or higher?*

**Responses and Revisions:**

Thank you for raising this question. First of all, the size distribution of aerosols is often found to be a log-normal distribution. Then, the $d_{ae}$ distribution is asymmetrical on a linear axis and we found that $d_{ae}$ distribution was fitted well with log-normal distribution.

a)  We used DMA-CPMA in a static configuration to select soot particles with specific $d_{m}$ and $m$. Particles with higher order charges can also be selected and the values of $d_{m}$ and $m$ can be calculated. According to Eq. (1) and Eq. (16), the $d_{ae}$ of particles with different charges can be calculated by:

$$\frac{\pi}{6}\rho_0 Cc(d_{ae})d_{ae}^2 = \frac{Cc(d_m)}{d_m}k_f[d_m(nm)]^{D_{fm}} \cdot 10^{-18} ,$$   R5

The corresponding $d_{ae}$ of the selected particles with different charges can be calculated with selected $d_{m}$ and $m$ according to Eq. R5. The $d_{ae}$ for different sizes soot particles with single, double and triple charges are shown in Table R1. The $d_{ae}$ distribution for different sizes particles was fit well with single log-normal distribution and the values of mode $d_{ae}$ were determined, which denoted as $d_{ae\_sd}$. The deviations between the calculated $d_{ae}$ and fitted $d_{ae}$ for particles with $d_{m}$ of 150 nm, 200 nm and 250 nm are within 0.31% while the $d_{ae\_sd}$ for particles with 80 nm and 100 nm are much larger than the calculated $d_{ae}$. The deviations for 80 nm and 100 nm particles are 7.38% and 6.85%, respectively. We think that the peaks shift right because of multiple charged particles, so we use the summation of multiple log-normal distributions. Although the deviation for 150 nm particles is 0.29%, the summation of multiple log-normal distributions is used because the points with $d_{ae}$ larger than 90 nm can't be fitted well with single log-normal distribution.

**Table R1. The calculated $d_{ae}$ for mobility and mass selected particles with single, double or triple charges and the single log-normal fitted $d_{ae\_sd}$. The deviations between the calculated $d_{ae\_1q}$ of particles with single charge and the fitted $d_{ae\_sd}$ for different size soot particles.**

| $d_{m}$(nm) | $m$ (fg) | Calculated $d_{ae}$ (nm) | | | | $d_{ae\_sd}$ (nm) | Deviation |
|---|---|---|---|---|---|---|---|
| | | $d_{ae\_1q}$ | $d_{ae\_2q}$ | $d_{ae\_3q}$ | $d_{ae\_4q}$ | | |
| 80 | 0.16 | 51.5 | 62 | 70.7 | 78.5 | 55.3 | 7.38% |
| 100 | 0.27 | 56.9 | 70.4 | 81.8 | 92.3 | 60.8 | 6.85% |
| 150 | 0.66 | 70.1 | 91.8 | 111.2 | 129.4 | 70.3 | 0.29% |
| 200 | 1.28 | 82.9 | 113.9 | 141.8 | 167.7 | 83.1 | 0.24% |
| 250 | 2.17 | 95.6 | 157.8 | 172.2 | 205.5 | 95.9 | 0.31% |

b)   The DMA-CPMA selected particles with specific $d_m$ and m, and the corresponding $d_{ae}$ of selected particles with different charges can be calculated with Eq. R5. The calculated values are set as the start points of central values of $d_{ae}$. For example, the central values of $d_{ae}$ used in 80 nm particles fitting are 51.5 nm, 62 nm and 70.7 nm.

c)   The constraint of peak width of particles with single, double and triple charges were the same. The start point of the coefficient $\sigma$ (two times of peak width) was e^0.5, and the lower and upper bounds were e^0 and e^1, respectively.

d)   The fitting coefficients with 95% confidence bounds are summarized in Table R2.

**Table R2. The fitting coefficients with 95% confidence bounds for different mobility and mass selected particles with single, double and triple charges.**

|      | N (95% confidence bounds) | $\mu$ (95% confidence bounds) | $\sigma$ (95% confidence bounds) |
|------|---------------------------|-------------------------------|----------------------------------|
| 80   | 1718.8 (1476.9, 1961.9)   | 53.9 (fixed at bound)         | 1.07 (1.05, 1.08)                |
|      | 637.1 (337.7, 936.5)      | 60.65 (fixed at bound)        | 1.06 (1.02, 1.10)                |
|      | 25.1 (906.7, 957.0)       | 70.9 (fixed at bound)         | 1.02 (0.07, 15.30)               |
| 100  | 5985.0 (5412.2, 6557.9)   | 59.5 (fixed at bound)         | 1.09 (1.08, 1.10)                |
|      | 1279.7 (746.7, 1812.6)    | 68.6 (fixed at bound)         | 1.08 (1.04, 1.12)                |
| 150  | 8361.9 (7997.1, 8726.7)   | 69.9 (69.5, 70.4)             | 1.11 (fixed at bound)            |
|      | 329.3 (-31.4, 690.0)      | 93.5 (78.6, 111.3)            | 1.11 (fixed at bound)            |
| 200  | 8086.6 (7864.7, 8308.4)   | 83.1 (82.9, 83.3)             | 1.099 (1.096, 1.102)             |
| 250  | 9835.5 (9603.9, 10067.1)  | 95.9 (95.7, 96.1)             | 1.095 (1.093, 1.098)             |

e)   We have included the uncertainties in particles number concentration. We scanned the $d_{ae}$ distributions at least three times for each size particles. The average of the number densities was used and error bar is shown in the figures.

f)   The $d_{ae}$ of particles with higher order of charges (q>3) can also be calculated using Eq. R5. The values are very large (e.g. 78.5 nm for 80 nm particle with 4 charges) and close to upper limit of scanning range (35 nm~80 nm for 80 nm particles). Nonetheless, we think that particles with higher order of charges do not exist since the number density already tends to zero at the end of the scanning.

**Technical corrections:**

**7. Comments and suggestions:**
**Line 18:** *"the potential multiple charging effect" Elaborate.*

**Responses and Revisions:**

Thank you for the comment. The potential multiple charging effects of Tandem System, such as DMA-CPMA and DMA-AAC et al., have been discussed in the following text from Line 21 to Line 28: "Our results show that the ability to remove multiply charged particles mainly depends on particles morphology and instruments resolutions of DMA-CPMA system. Using measurements from soot experiments and literature data, a general trend in the appearance of multiple charging effect with decreasing size when selecting aspherical particles was observed. Otherwise, our results indicated that the ability of DMA-AAC in a static configuration to eliminate particles with multiple charges is mainly related to the resolutions of classifiers. In most cases, DMA-AAC can eliminate multiple charging effect regardless of the particle morphology in a static configuration, but multiply charged particles will be selected when decreasing resolution of DMA or AAC".

**8. *Comments and suggestions:***

*Line 19:* *"remove"*

*Resolve?*

**Responses and Revisions:**

Thank you for your advice. We think "remove" should be used since we discuss DMA-CPMA in a static configuration.

**9. *Comments and suggestions:***

*Line 20:* *"instruments setups of DMA-CPMA system"What is meant by instrument setups? Elaborate.*

**Responses and Revisions:**

We have changed this in the revised manuscript:

"Our results show that the ability to remove multiply charged particles mainly depends on particles morphology and instruments resolutions of DMA and CPMA".

**10. *Comments and suggestions:***

*Line 23:* *"DMA-AAC can eliminate multiple charging effect"*

*This is not strictly correct as written since you can only "remove" the multiple charging artifact when usedin a static configuration. In a scanning mode, the contributions would be resolvable.*

**Responses and Revisions:**

Sorry we didn't make it clear that we used DMA-AAC in a static configuration. We have changed it in the revised manuscript:

"In most cases, DMA-AAC in a static configuration can eliminate multiple charging effect regardless of the particle morphology".

**11. *Comments and suggestions:***

*Line 24:* *"while particles with multiple charges can be selected when decreasing resolutions of DMA and AAC"*

*Confusing as written.*

**Responses and Revisions:**

We have changed this in the revised manuscript:

"In most cases, DMA-AAC in a static configuration can eliminate multiple charging effect regardless of the particle morphology, but multiply charged particles will be selected when decreasing resolutions of DMA or AAC".

**12. *Comments and suggestions:***

*Line 25:* *"We propose that the multiple charging effect should be reconsidered when using DMA-CPMA or DMA-AAC system in estimating size and mass resolved optical properties in the field and lab experiments."*

*This statement is not clear as written. How should the effects be "reconsidered"?*

**Responses and Revisions:**

We have changed this in the revised manuscript:

"We propose that the potential influence of the multiple charging effect should be considered when using DMA-CPMA or DMA-AAC systems in estimating size- and mass-resolved optical properties in field and lab experiments".

**13. *Comments and suggestions:***

***Line 35:*** *"is the most commonly used size classifier"*
*If the DMA is the most commonly used classifier, why is only the original Knutson and Whitby reference provided?*

**Responses and Revisions:**

Park et al. (2008) reviewed the tandem techniques of DMA and other measurements. We have included it and other references of application of DMA in laboratory and field studies therein.

**14. *Comments and suggestions:***

***Line 38:*** *"particles are required to be pre-charged"In what sense do they need to be pre-charged?*

**Responses and Revisions:**

We have changed this in the revised manuscript:

"However, particles must be precharged when classified by a DMA or PMA because DMA and PMA classify particles based on electrical mobility and mass-to-charge ratio".

**15. *Comments and suggestions:***

***Line 49:*** *"This conclusion implies that it can hardly to achieved that all the multiply charged particles are effectively excluded for aspherical particles, especially for soot particles."*
*Grammar makes this sentence confusing.*

**Responses and Revisions:**

We have changed this in the revised manuscript:

"This conclusion implies that multiply charged particles cannot be effectively excluded for aspherical particles, especially for soot particles"

**16. *Comments and suggestions:***

***Line 60:*** *"dynamic shape factor (χ), can be inferred…"*
*The measured χ may depend upon the combination of instruments used. See (Yao et al., 2020) and potentially Table 2 here.*

**Responses and Revisions:**

We have added the sentence: "The derived $\rho_{\text{eff}}$ and χ depend upon the combination of instruments used, while the nonphysical values of χ and $\rho_{\text{eff}}$ for aspherical particles can be determined by the AAC-APM(Yao et al., 2020) and AAC-CPMA (Kazemimanesh et al., 2022)".

**17. *Comments and suggestions:***

***Line 89:*** *"The transfer function is an isosceles triangle with value of 1 at Zp * and going to 0 at (1 ±*

*βDMA)·Zp\*."*

*I think it is important to mention that this translates to asymmetric distributions in Dm and mp since their relationship with Zp is nonlinear.*

**Responses and Revisions:**

We have added the sentence: "It translates to asymmetric in $d_m$ since the relationship between $d_m$ and $Z_p$ is nonlinear".

**18. *Comments and suggestions:***

*Line 106: "is much simpler and more robust"*
*Elaborate.*

**Responses and Revisions:**

We have changed this in the revised manuscript:

"They considered the Taylor series expansion about the center of the gap ($r_c=(r_{2\_CPMA}+r_{1\_CPMA})/2$) instead of the equilibrium radius to avoid problems with the scenario in which the equilibrium radius does not exist. This method is much simpler and more robust"

**19. *Comments and suggestions:***

*Line 146: "while the condensation particle counter"*
*Was only a single CPC used during the soot characterization experiments? In the previous sentences, the author make it seem like the measurements were made simultaneously and in parallel. Also, please include flow rates.*

**Responses and Revisions:**

Sorry for the misunderstanding. In this study, we only use one CPC. We have revised the manuscript and Figure 1. In the previous sentence, we have revised it to "For the soot characterization, the monodisperse aerosol flow was switched between two parallel lines and fed into the CPMA (Cambustion Ltd., UK) and AAC (Cambustion, Ltd., UK, $Q_{sh}/Q_a = 10$); meanwhile, the condensation particle counter (CPC, Model 3756, TSI, Inc., USA, 0.3 L min$^{-1}$) was switched between the CPMA and AAC. The particle mass ($m$) and aerodynamic diameter ($d_{ae}$) were determined by the scanning mode of the CPMA and AAC, while the CPC recorded their corresponding number concentrations at each setpoint. For each $d_m$, the $m$ and $d_{ae}$ distributions were measured three times. Between measurements of each $d_m$, the CPC was used behind the DMA, and the number size distribution of the generated soot particles was measured by SMPS to ensure that the generated soot particles did not change during the whole experiment".

**20. *Comments and suggestions:***

*Line 147: "concentration"*
*Number density of particles. Concentration is assumed to have units of mole per unit volume.*

**Responses and Revisions:**

We have revised it to "number concentration".

**21. *Comments and suggestions:***

*Line 148: "fitted to log-normal distribution"*

*Please include the equation that you used to fit your distribution since there are many ways to define the same relationship. Also, considering the shapes of the distributions shown in Fig. 5 and S2, why was a log-normal distribution utilized? The distributions appear symmetric, and they're plotted on a linear axis, so some justification is warranted.*

**Responses and Revisions:**

The equation $N(d_p) = \frac{N_0}{\sqrt{2\pi}\ln\sigma} \exp\left(\frac{-(\ln(d_p) - \ln(\mu))^2}{2(\ln\sigma)^2}\right)$ has been added. In Fig. 5 and S2, the distributions are plotted on a log axis, not a linear axis. This distribution is asymmetric on linear axis, hence we use log-normal distribution.

**22. *Comments and suggestions:**

*Line 151: "density of PSL" Please enumerate.*

**Responses and Revisions:**

We have added the density value of 1050 kg m$^{-3}$ for PSL.

**23. *Comments and suggestions:**

*Line 153: "effect of particles selected by DMA-CPMA system, the $d_{ae}$ distribution of twice classified particles".*

*Please provide additional information about this portion of the procedure.*

**Responses and Revisions:**

We have changed this in the revised manuscript:

"To quantify the multiple charging effect of particles selected by the DMA-CPMA system, the soot particles were initially selected by the DMA-CPMA at different $d_m$ and the corresponding $m$. Then, the $d_{ae}$ distribution of mobility and mass selected particles was obtained by stepping the AAC rotation speed of the cylinder with simultaneous measurement of the particle concentration at the AAC outlet using a CPC".

**24. *Comments and suggestions:**

*Line 167: "we explain the transfer functions of DMA-CPMA and DMA-AAC utilizing the literature data of soot particles"*

*When the transfer functions were calculated, what range of parameters were used? And how exactly were the transfer functions solved? Iteratively or something else? If iteratively, what was the $\Delta t$ for each step and the number of individual trajectories considered? These details need to be provided somewhere in the manuscript.*

**Responses and Revisions:**

This is a very useful suggestion. We have included the details in the revised manuscript: . "We explain the transfer functions of DMA-CPMA and DMA-AAC utilizing the literature data of soot particles (Pei et al., 2018). The $d_m$ and $m$ of the representative particles are 100 nm and 0.33 fg, respectively, and the corresponding $d_{ae}$ is 68.3 nm according to Eq. (19). In the calculation, the following parameter set was employed for the calculations: $d_m = 80$ nm, $Q_{CDMA} = 0.3$ L min$^{-1}$, $\beta_{DMA} = 0.1$, $m = 0.16$ fg, $Q_{CPMA}=0.3$ L min$^{-1}$, $R_m = 8$, $d_{ae} = 68.3$ nm, $Q_{AAC} = 0.3$ L min$^{-1}$, $\beta_{AAC} = 0.1$".

The maximum and minimum values of $d_m$ for particles with $n$ charges can be derived and denote as $d_{m\,n,max}$ and $d_{m\,n,min}$, respectively. The maximum and minimum m of particles bearing single and double charges which are derived from Eq. 15 denote as $m_{1,\,max}$, $m_{1,min}$, $m_{2,max}$ and $m_{2,min}$. The maximum and minimum values of $d_{ae}$

that can be selected by AAC denote as $d_{ae,max}$ and $d_{ae,min}$. The transfer functions of DMA-CPMA and DMA-AAC were solved iteratively using logarithmically spaced $d_m$, $m$ and $d_{ae}$, which included 600 points, respectively. the ranges of $d_m$, $m$ and $d_{ae}$ used in the calculations were from $< d_{m1,min}$ to $> d_{m2,max}$, from $< m_{1,min}$ to $> m_{2,max}$, from $< d_{ae,min}$ to $> d_{ae,max}$, respectively.

**25. *Comments and suggestions:**

*Line 178: "representing pre-exponential factor (ρf)"*
*Having the pre-exponential factor share a variable with effective density ($ρ_{eff}$) is confusing since I would expect them to share units when they do not; the units are (g nm$^{-Dfm}$) and (g nm$^{-3}$) for $ρ_f$ and $ρ_{eff}$, respectively. Additionally, I recommend including a normalization factor in the Dm term to avoid having fractional units, e.g., $D_{fm}$ = 2.28 will have $ρ_f$ with units of g nm$^{-2.28}$.*
**Responses and Revisions:**

Equation 22 has been changed to $m = k_f \frac{(d)_m^{D_{fm}}}{nm}$, then $k_f$ will have the unit of "g". Equation 23 has been changed to $\log(m) = D_{fm} \log\left(\frac{d_m}{nm}\right) + \log(k_f)$.

**26. *Comments and suggestions:**

*Line 184: "In the exemplary case, the derived Dfm of premixed flame generated soot particles was 2.28,"*
*What study does this refer to? Reference?*
**Responses and Revisions:**
The reference (Pei et al., 2018) has been added.

**27. *Comments and suggestions:**

*Line 189: "The DMA-CPMA system can eliminate the multiply charged particles only if the Dfm of particlesis larger than the slope of a line connecting line connecting (dm, m) = (dm2,min, m2,max)(dm1,,m1) (as PP0 shown in Fig. 2)."*
*The point (dm2, m2) appears to correspond to the actual mass and $d_m$ of a particle bearing 2 charges instead of the effective mass (m/2). This is unclear and has significant implications for the calculated transfer functions and resultant discussion since the effective mass is ultimately what affects instrument resolution.*
**Responses and Revisions:**
Thank you for your comment. We use the DMA-CPMA in a static configuration. When particles are selected by the DMA-CPMA with certain $d_m$ and m, the corresponding doubly charged particles of $d_{m2}$ and $m_2$ would also be selected.
The particles are shown in figure 2 in actual $d_m$ and $m$, but when we calculate the resolution of DMA and CPMA, the mobility and effective mass are used. The resolution of CPMA can be calculated by Eq. R2, where $m_1$ is the mass of singly charged particles which can be selected by the CPMA, i.e. effective mass.

**28. *Comments and suggestions:**

*Line 190: "line connecting (dm, m) = (dm2,min, m2,max)(dm1,,m1) (as PP0 shown in Fig. 2)."*
*I'm assuming that the location of (dm1, m1) is point P0 and is at the center of the q = 1 transfer function?This is not clear in the figure. There's an extra "," after dm1 in the text.*

**Responses and Revisions:**

We have removed the extra "," and the $PP_0$ has been marked out in Fig. 2.

**29. *Comments and suggestions:***

***Line 201:*** *"are necessary to reduce the potential of multiply charged particles"By increasing the resolution of the measurement?*

**Responses and Revisions:**

We have changed this in the revised manuscript:

"When selecting particles of certain $d_m$ and $m$, by decreasing $Q_{CPMA}$, or increasing $\omega$ and $\beta_{DMA}$ i.e. by increasing the resolution of the measurement, the potential of multiply charged particles is reduced".

**30. *Comments and suggestions:***

***Line 205:*** *"PP0 of 3.55 was derived when DMA-APM selects the same example soot particles"Compared to what?*

**Responses and Revisions:**

We have changed this in the revised manuscript:

"the slope of $PP_0$ of 3.55 was derived when the DMA-APM selects the same example soot particles from Pei et al. (2018) ($d_m$ of 100 nm and $m$ of 0.33 fg) with a $D_{fm}$ of 2.28".

**31. *Comments and suggestions:***

***Line 208:*** *"critical slope of PP0"*
*Z-axis in Figure is labelled as "The slope of PP0".*

**Responses and Revisions:**

We have revised it to "the critical slope of $PP_0$" in Fig. 3, consistent with the text.

**32. *Comments and suggestions:***

***Line 210:*** *"when the Dfm of particles is larger than the slope of PP0 which is represented as background color."*
*So, the color of the data point will be red shifted relative to the background when multiple charging is not affecting the output distribution? This is unclear and the rainbow color scheme isn't helping.*

**Responses and Revisions:**

The color scheme has been changed. The data points become square when $D_{fm}$ is smaller than the critical slope of PP0 in the background, i.e., the potential multiple charging effect may exist.

**33. *Comments and suggestions:***

***Line 213:*** *"Fig. 3, the dm, m and corresponding Dfm were taken from literature"*
*In the caption, the authors mention that the shapes correspond to the individual studies. Please elaboratewhich is which here?*

**Responses and Revisions:**

We have added more data to Fig. 3 and the references have been labeled in the figure.

**34.** *Comments and suggestions:*

*Line 214: "Generally, multiple charging effect can be avoided for DMA-CPMA to select soot particles with diameter larger than 200 nm.,"*

*I don't think the authors can conclude this without providing more data to support the claim. For fresh soot, my experience has been that multiple charging can be a problem at almost all Dm; e.g. see Figure S1of (Radney et al. 2014). Also, there's a "," after the "." at the end of the sentence.*

**Responses and Revisions:**

Thank you for your comment. More data has been added to Fig. 3. Generally, we think that the $D_{fm}$ of soot particles is around 2.2~2.4. For these soot particles, we conclude that multiple charging effect can be avoided for DMA-CPMA to select particles with mobility diameter larger than 200 nm. We have revised the conclusion to "Generally, for soot particles with $d_m$ of 2.2~2.4, multiple charging effect can be avoided for DMA-CPMA to select soot particles with mobility diameter larger than 200 nm".

We think the conclusion is consistent with the study of Radney et al (2014). The main difference between these two studies is the instruments configuration. In Figure S1 in Radney et al. (2014), DMA was used in static configuration and particles with $d_m$ of 350 nm were selected. The APM was used in scanning mode and the mass distribution of mobility selected particles was determined. However, we used the static configuration of both DMA and CPMA in this study. Particles were sequentially selected by their mobility and mass. Usually we select a certain $d_m$ and the corresponding mode mass. In the bottom of Figure S1 in Radney et al. (2014), three regions are determined and singly charged are selected at mass < 4 fg. In the top of Figure S1 in Radney et al. (2014), the mode mass of particles with $d_m$ of 350 nm is around 3.5 fg, as a result, no multiple charging exist when selecting these particles. Nonetheless, we demonstrate different result. The $d_m$ of lacy soot particles in Radney et al. is 1.79, which is smaller than common soot particles mentioned above. Applying particles with $d_m$ of 350 nm and $m$ of 3.5 fg to Figure 3 in this study, the $d_m$ of 1.79 is smaller than the critical PP0 of 2.02, i.e. multiple charging effect still exists after mobility and mass selection. We also apply other particles with smaller mass and the result shows that singly charged particles are selected at mass < 3 fg. In the bottom of Figure S1 (Radney et al., 2014), the slope of red line doesn't change obviously at mass of 3 fg maybe because the quantity of particles with higher charges is too small compared with particles with single charge since the peak of number concentration is at 3.5 fg. Moreover, we apply particles with $d_m$ of 350 nm and mass of 6 fg to Figure 3 and the result shows that doubly charged particles are selected by DMA-CPMA, which is consistent with the green line in the bottom of Figure S1 (Radney et al., 2014).

[Figure]

**RC-Figure 1:** The transfer functions of soot particles with $d_m$ of 350 nm and $m$ of (a) 3 fg, (b) 3.5 fg, (c) 4 fg and (d) 6 fg from Figure S1 in Radney et al. (2014).

**35. *Comments and suggestions:**

*Line 215: "diameter"*
*To which diameter metric are you referring?*
**Responses and Revisions:**
It has been changed to "mobility diameter".

**36. *Comments and suggestions:**

*Line 216: "eliminate"*
*Resolve?*
**Responses and Revisions:**
Thank you for your comment. We think it should be 'eliminate'. 'Resolve' is used when APM or CPMA is used in scanning mode and multiply charged particles can be identified from the mass distribution. We use the static configuration of DMA-CPMA and we are discussing if multiple charging can be avoided totally after mobility and mass selection. Hence, we think 'eliminate' is more proper.

**37. *Comments and suggestions:**

**Line 220:** *"Therefore, the multiple charge effect could be avoided theoretically."*
*Measurements by just an AAC will avoid multiple charging. Multiple charging only becomes a problem again when the tandem measurement is a DMA or PMA.*
**Responses and Revisions:**
We have changed this in the revised manuscript:
"Measuring solely with an AAC will avoid multiple charging. However, AAC cannot constrain the properties of aspherical particles as monodisperse as DMA or CPMA classification (Kazemimanesh et al., 2022). Multiple charging becomes a problem when the tandem measurement is made with a DMA or PMA".

**38. Comments and suggestions:**

**Line 242:** *"resulting in the minimum $D_{fm}$ of 1.41, which is the case for most atmospheric aerosol particles."*
*Is this the black dashed line drawn as PP0? If so, please label in figure. Also, please differentiate these dashes from the vertical and horizontal ones.*
**Responses and Revisions:**
The label of the black dashed line drawn as $PP_0$ has been added. The figure has been replotted.

**39. Comments and suggestions:**

**Line 245:** *"sheath flow rate of classifier is restricted by the instrument design" It's also important to note that sheath flow restricts the maximum size ranges.*
**Responses and Revisions:**
Agree. We have added the sentence: "In addition, the maximum size ranges are also restricted by the sheath flow, so in some cases, a lower sheath flow rate is required to select larger particles.".

**40. Comments and suggestions:**

**Line 255:** *"Dfm was 2.28" What was the value of $\rho_0$?*
**Responses and Revisions:**
We have added the value of $\rho_0$ and this sentence has been revised to "The fitted value of $D_{fm}$ and $k_f$ were 2.28 and $7.49 \times 10^{-6}$, respectively".

**41. Comments and suggestions:**

**Line 263:** *"which suggested that multiply charged particles are still classified in this circumstance."*
*This suggests that the contributions from the multiply charged particles can't be resolved.*
**Responses and Revisions:**
Thank you for your advice. We have changed this in the revised manuscript:
"which suggests that the contributions from the multiply charged particles can't be eliminated". We use "eliminate" here instead of "resolve" because we use the DMA-CPMA in a static configuration.

**42. Comments and suggestions:**

**Line 268:** *"the multiply charged particles can be resolved in aerodynamic size distribution"*
*I disagree that the q = 1 and 2 can be resolved. Please provide more evidence to support this claim.*
**Responses and Revisions:**

We have changed this in the revised manuscript:

"Since the classification of AAC is different from DMA and CPMA, the aerodynamic size distributions of mobility and mass selected particles were characterized."

**43. *Comments and suggestions:**

***Line 270:*** *"PNSDae was fitted using log-normal distributions and three peaks which correspond to singly, doubly and triply charged particles were identified."*
*What does the peak at Dae < 40 nm correspond to? Please mention.*

**Responses and Revisions:**

Some small particles remaining in the AAC induced the peak at $d_{ae}$ <40 nm. These residual particles were measured even if the sample flow is filtered. This reason has been added in the text.

**44. *Comments and suggestions:**

***Line 296:*** *"Mie theory was used to calculate the theoretical absorption coefficient at the wavelength of 550 nm."*
*Mie theory probably isn't the "best" method to use here since soot particles are aspherical agglomerates. Realistically though, the Mie comparison is only being used to prove a point about the impact of multiple charging. So, in this instance any errors in the calculated optical properties are somewhat inconsequential. Some mention of this nuance should be mentioned.*

**Responses and Revisions:**

Agree. The following text has been added in the text:

"Mie theory is probably not the "best" method to use here since soot particles are aspherical agglomerates. Realistically, however, the Mie comparison is only being used to prove a point about the impact of multiple charging. Therefore, in this instance, any errors in the calculated optical properties are somewhat inconsequential."

**45. *Comments and suggestions:**

***Line 301:*** *"integral concentration for particles"*
*Integrated number density of particles. Concentration is assumed to have units of mole per unit volume.*

**Responses and Revisions:**

It has been changed to "number concentration".

**46. *Comments and suggestions:**

***Table 2:*** *It'd be interesting (but not necessary) to include a comparison of the derived shape factors (χ) for each method.*

**Responses and Revisions:**

The derived shape factors have been added. The shape factors calculated from two methods are consistent. We think the values of $\chi$ are not necessary in this study, so we didn't include them in the text.

| $d_m$(nm) | $m$(fg) | $d_{ae}$(nm) | $\rho_{DMA-AAC}$ (kg m$^{-3}$) | $\rho_{DMA-CPMA}$ (kg m$^{-3}$) | Deviation | $\chi_{DMA-AAC}$ | $\chi_{DMA-CPMA}$ | Deviation |
|---|---|---|---|---|---|---|---|---|
| 80 | 0.16 | 48 | 551.2 | 596.8 | 7.65% | 2.05 | 1.96 | 5.05% |
| 100 | 0.27 | 55 | 488 | 515.7 | 5.38% | 2.17 | 2.10 | 3.42% |

| | | | | | | | | |
|---|---|---|---|---|---|---|---|---|
| 150 | 0.66 | 67 | 359. 1 | 373.5 | 3.86% | 2.49 | 2.43 | 2.34% |
| 200 | 1.28 | 82 | 303.2 | 305.6 | 0.77% | 2.62 | 2.60 | 0.44% |
| 250 | 2.17 | 96 | 262.8 | 265.2 | 0.90% | 2.71 | 2.70 | 0.50% |

**47.** *Comments and suggestions:*

*Figure 2: "Example of DMA-CPMA transfer function"*
*Transfer function for what? Soot?*

**Responses and Revisions:**

The reference has been added in the sentence. It has been changed to "Example of DMA-CPMA transfer function of flame-generated soot particles (Pei et al., 2018)".

**48.** *Comments and suggestions:*

*Figure 3: The coloration of this figure is very hard to understand.*

**Responses and Revisions:**

Agree. The figure has been changed.

**49.** *Comments and suggestions:*

*Figure 4: Z-axis is labelled "Transfer function of DMA-CPMA" and the caption says DMA-AAC.*

**Responses and Revisions:**

The Z-axis label has been changed to "Transfer function of DMA-AAC".

**50.** *Comments and suggestions:*

*Figure S1: Having the Z-axis go all the way to zero is confusing.*

**Responses and Revisions:**

Agree. The figure has been changed.

**Grammar corrections:**

We are very appreciated for all the spelling and grammar corrections. We have corrected all the mistakes which are listed in detail below.

**51.** *Comments and suggestions:*

*Line 15: "effect" effects*

**Responses and Revisions:**

It has been changed to "effects".

**52.** *Comments and suggestions:*

*Line 16: "technique" techniques*

**Responses and Revisions:**

It has been changed to "techniques".

**53.** *Comments and suggestions:*

*Line 39:* *"resulting in that particles" Delete "that"*
**Responses and Revisions:**
It has been deleted.

**54.** *Comments and suggestions:*

*Line 41:* *"subsequence" subsequent*
**Responses and Revisions:**
It has been changed to "subsequent".

**55.** *Comments and suggestions:*

*Line 51:* *"conducted" investigated?*
**Responses and Revisions:**
It has been changed to "investigated".

**56.** *Comments and suggestions:*

*Line 52:* *"ammonia" ammonium*
**Responses and Revisions:**
It has been changed to "ammonium".

**57.** *Comments and suggestions:*

*Line 80:* *"particle" elementary*
**Responses and Revisions:**
It has been changed to "elementary".

**58.** *Comments and suggestions:*

*Line 107:* *"radical" radial*
**Responses and Revisions:**
It has been changed to "radial".

**59.** *Comments and suggestions:*

*Line 113:* *"should" would*
**Responses and Revisions:**
It has been changed to "would".

**60.** *Comments and suggestions:*

*Line 134 and Line 328:* *"nominated" nominal*
**Responses and Revisions:**
They have been changed to "nominal".

**61.** *Comments and suggestions:*

*Line 142: "80, 100, 150, 200" missing units*
**Responses and Revisions:**
It has been changed to "80 nm, 100 nm, 150 nm, 200 nm".

**62.** *Comments and suggestions:*

*Line 149: "70, 150" missing units*
**Responses and Revisions:**
It has been changed to "70 nm and 150 nm".

**63.** *Comments and suggestions:*

*Line 182: "however" however is used twice in the same sentence*
**Responses and Revisions:**
The first "however" has been deleted.

**64.** *Comments and suggestions:*

*Line 258: "along" delete word*
**Responses and Revisions:**
It has been deleted.

**65.** *Comments and suggestions:*

*Line 261: "80, 100, 150, 200" units?*
**Responses and Revisions:**
It has been changed to "80 nm, 100 nm, 150 nm, 200 nm".

**66.** *Comments and suggestions:*

*Line 308: "70" units?*
**Responses and Revisions:**
It has been changed to "70 nm".

**References:**

Kazemimanesh, M., Rahman, M.M., Duca, D., Johnson, T.J., Addad, A., Giannopoulos, G., Focsa, C. and Boies, A.M.: A comparative study on effective density, shape factor, and volatile mixing of non-spherical particles using tandem aerodynamic diameter, mobility diameter, and mass measurements. J. Aerosol Sci., 161, 105930, https://doi.org/10.1016/j.jaerosci.2021.105930, 2022.

Tavakoli, F., and Olfert, J. S.: Determination of particle mass, effective density, mass–mobility exponent, and dynamic shape factor using an aerodynamic aerosol classifier and a differential mobility analyzer in

tandem, J. Aerosol Sci., 75, 35-42, https://doi.org/10.1016/j.jaerosci.2014.04.010, 2014.

Radney, J. G., R. You, X. Ma, J. M. Conny, M. R. Zachariah, J. T. Hodges, C. D. Zangmeister.: Dependence of soot optical properties on particle morphology: Measurements and model comparisons *Environ. Sci. Technol.* 48:3169-3176. https://doi: 10.1021/es4041804, 2014

Radney, J. G., Ma, X., Gillis, K. A., Zachariah, M. R., Hodges, J. T., and Zangmeister, C. D.: Direct Measurements of Mass-Specific Optical Cross Sections of Single-Component Aerosol Mixtures, Anal. Chem., 85, 8319-8325, https://doi.org/10.1021/ac401645y, 2013.

Yao, Q., Asa-Awuku, A., Zangmeister, C. D., and Radney, J. G.: Comparison of three essential sub-micrometer aerosol measurements: Mass, size and shape, Aerosol Sci. Technol., 1-18, https://doi.org/10.1080/02786826.2020.1763248, 2020.

**Response to Anonymous Referee #2**

We thank the reviewer for the constructive suggestions and comments concerning our manuscript entitled "Characterization of tandem aerosol classifiers for selecting particles: implication for eliminating multiple charging effect" (ID: amt-2021-436). Those comments are valuable and very helpful for improving our paper, as well as the important guiding significance to our studies. Below, we provide a point-by-point response to individual comment (Reviewer comments in italics, responses in plain font; page numbers refer to the AMTD version)

**1. *Comments and suggestions:**

*Line 88, Eq. (3): The equation for the transfer function of the DMA in non-diffusing caseis not given in the general form. Eq. (3) is only correct when aerosol inlet and aerosol sampling flow rates are equal (i.e. the DMA is operated in balanced flow mode), leadingto $\delta = 0$ in the general form of DMA transfer function [Eq. (7) in Stolzenburg and McMurry (2008)]. This information should be noted in the paper.*

**Responses and Revisions:**

Thank you for your comment. We have changed it in the revised manuscript:

"Assuming that aerosol inlet and aerosol sampling flow rates are equal, the transfer function of DMA can be expressed as follows when particle diffusion is negligible".

**2. *Comments and suggestions:**

*Line 139: The primary reference for the miniature inverted soot generator is as follows,which should be cited in the text: Kazemimanesh, M., Moallemi, A., Thomson, K., Smallwood, G., Lobo, P. and Olfert, J.S., 2019. A novel miniature inverted-flame burnerfor the generation of soot nanoparticles. Aerosol Science and Technology, 53(2), pp.184-195.*

**Responses and Revisions:**

Thank you for the reminder. We have added it in the text.

"Soot particles were generated by a miniature inverted soot generator (Argonaut Scientific Ltd., Canada) with a propane flow of 74.8 SCPM and an air flow rate of 12 SLPM. Although this operation setting is not in the open-tip flame regime, the flame is open-tip consistent with Fig. 2d in Moallemi et al. (2019). Detailed aerosol generation methods can be found in Kazemimanesh et al. (2019b) and Moallemi et al. (2019)"

**3. *Comments and suggestions:**

*Line 138-140: To reproduce the experiments in this study, it is necessary to givedetails of propane and air flow rates used in the soot generator.*

**Responses and Revisions:**

Thank you for your reminder. We have changed it in the revised manuscript:

"Soot particles were generated by a miniature inverted soot generator (Argonaut Scientific Ltd., Canada)

with the propane flow of 74.8 mL min$^{-1}$ and the air flow rate of 12 L min$^{-1}$. Although this operation setting is not in the open-tip flame regime, the flame is open-tip consistent with the Fig.2d in Moallemi et al. (2019)".

**4. *Comments and suggestions:***

***Lines 145-155:*** *For one or two mobility-selected particles, please add representativeplots for the measured spectral density of mass (dN/dlog m$_p$) and aerodynamic diameter (dN/dlog d$_{ae}$), either in the paper or in the supplementary material.*
**Responses and Revisions:**

The measured spectral density of mass (d$N$/dlog $m_p$) and aerodynamic diameter (d$N$/dlog $d_{ae}$) for particles with $d_m$ of 150 nm and 250 nm have been added in the supplementary material.

**5. *Comments and suggestions:***

***Lines 198-199:*** *It states that the ability of DMA-CPMA to eliminate multiply charged particles depends on the resolutions of both DMA and CPMA; however, dependence onthe resolution of CPMA is not obvious from Eq. (25). Can the authors clarify this?*
**Responses and Revisions:**

Thank you for your comment. The resolution can be calculated by

$$R_m = \frac{m_1}{m_{1,max}-m_1} = \frac{2\pi B_{1,max} L_{CPMA} r_c^2 \omega^2 m_1}{Q_{CPMA}},$$

We have changed the Eq. (25) to

$$D_{fm} > PP_0 = \frac{log(m_{2,max}/m_1)}{log(d_{m2,min}/d_{m1})} = \frac{log\left(2+\frac{1}{R_m(1+\beta_{DMA})}\right)}{log\left(\frac{2}{(1+\beta_{DMA})}\frac{Cc(d_{m2,min})}{Cc(d_{m1})}\right)}$$

in the text and the relationship between the slope of PP$_0$ and the resolution is more obvious.

**6. *Comments and suggestions:***

***Line 201:*** *The sentence states that a larger β is necessary to reduce the potential ofmultiply-charged particles, but it seems that Eq. (25) shows the opposite. Can the authors double-check this?*
**Responses and Revisions:**

Thank you for your comment. $(1 - \beta_{DMA})$ has been replaced by $(1 + \beta_{DMA})$. We have revised the Eq. (25) to

$$D_{fm} > PP_0 = \frac{log(m_{2,max}/m_1)}{log(d_{m2,min}/d_{m1})} = \frac{log\left(2+\frac{1}{R_m(1+\beta_{DMA})}\right)}{log\left(\frac{2}{(1+\beta_{DMA})}\frac{Cc(d_{m2,min})}{Cc(d_{m1})}\right)}$$

**7. *Comments and suggestions:***

***Line 303: In Fig. 5c*** *and all related figures, the unit used incorrectly for absorption cross section is Mm$^{-1}$, which is the unit for absorption coefficient (length$^{-1}$). The correctunit for absorption cross section should be m$^2$ particle$^{-1}$ (derived from absorption coefficient/particle number concentration).*

**Responses and Revisions:**

Thank you for the comment. The Y axis is the absorption coefficient $\alpha_{abs}$ ($Mm^{-1}$). We have revised it in the Fig. 5c and 6c. The text has also been changed accordingly.

**8.** *Comments and suggestions:*

*Lines 306-310: It is not very clear to me why multiple charging effects due to the use of DMA-CPMA would affect the MAC or DRF of soot particles. As far as I understand, global climate models consider a specific (mostly constant) MAC value for black carbon particles to estimate their DRF, without regard for multiple charging. Unless the authorsare claiming that the MAC values used in current climate models are grossly incorrect.*

**Responses and Revisions:**

Sorry we didn't make it clear. Previous studies used DMA-APM or DMA-CPMA to investigate the mass-specific MAC (Radney et al., 2013; Zangmeister et al., 2018). Our study tried to illustrate that multiple charging effect can affect classification of DMA-CPMA and the measured value of MAC. If using the measured MAC to evaluate the DRF of fresh soot particles, it can cause uncertainties.

**9.** *Comments and suggestions:*

*Introduction and discussion section: There are a few recent studies that have looked at tandem measurements of mobility diameter, mass, and aerodynamic diameter to studythe effective density and shape factor of spherical and non-spherical particles. These studies have used a combination of DMA, AAC, and APM or CPMA and, in my view, are relevant to this paper and should be mentioned in the introduction and their results discussed where necessary:*

*Yao, Q., Asa-Awuku, A., Zangmeister, C.D. and Radney, J.G., 2020. Comparison of three essential sub-micrometer aerosol measurements: Mass, size and shape. AerosolScience and Technology, 54(10), pp.1197-1209.*

*Kazemimanesh, M., Rahman, M.M., Duca, D., Johnson, T.J., Addad, A., Giannopoulos, G., Focsa, C. and Boies, A.M., 2022. A comparative study on effective density, shape factor, and volatile mixing of non-spherical particles using tandem aerodynamic diameter, mobility diameter, and mass measurements. Journal of Aerosol Science, 161,p.105930.*

**Responses and Revisions:**

Thank you for your suggestion. We have added these literatures in the introduction (Line 72 and Line 73) and discussion (Line 265)

**Editorial and technical corrections**

**10.** *Comments and suggestions:*

*Line 272: I cannot find Eq. (30) in the paper.*
**Responses and Revisions:**

Sorry for our mistake. We have revised it to Eq. (16).

**11.** *Comments and suggestions:*

*Line 301: Change the sentence to "Subsequently, absorption cross section, σabs, was derived using the absorption coefficient and total number concentration of particles with different charging states."*
**Responses and Revisions:**

We have changed it in the revised manuscript:

"Subsequently, absorption coefficient, $\alpha_{abs}$, was derived using the Mie theory and $PNSD_{ve}$ of particles with different charging states.".

**12.** *Comments and suggestions:*

*Lines 308-310: This sentence is written very poorly (huge amount? huge error?). Please rephrase this sentence and avoid ambiguous adjectives.*
**Responses and Revisions:**

We have changed it in the revised manuscript:

"A large amount of 70 nm -90 nm soot particles was emitted from a diesel engine (Wierzbicka et al., 2014), and neglecting the multiple charging effect in the measurement of mass-specific MAC on this size range will result in significant bias in the estimation of radiative forcing of automobile-emitted soot particles".

**13.** *Comments and suggestions:*

*Line 312: Reference to Table 3 should be given in the earlier paragraph (perhaps in line 302).*
**Responses and Revisions:**

Thank you for the comment. Table 3 now is given in line 374.

**Grammar corrections:**

We are very appreciated for all the spelling and grammar corrections. We have corrected all the mistakes which are listed in detail below.

**14.** *Comments and suggestions:*

*Line 41-44*
**Responses and Revisions:**

In this sentence, the word "subsequence" has been changed to "subsequent" and the word "while" has been deleted. It has been changed to "This may introduce uncertainty in the subsequence subsequent characterization. Radney et al. (2013) demonstrated that although the single-charged particles account for the highest number fraction (46.3%) of the DMA-classified particles (200 nm), while their contributions to the total mass concentration and extinction are insignificant (10.8% and 7.96%, respectively)".

**15.** *Comments and suggestions:*

*Line 49*
**Responses and Revisions:**

We have reorganized this sentence. It has been changed to "Theoretically, the ability of DMA-APM to eliminate multiply charged particles is governed by the particles morphology and setups of DMA-APM (Kuwata, 2015). This conclusion implies that it can hardly to achieved that all the multiply charged particles are cannot be effectively excluded for aspherical particles, especially for soot particles".

**16.** *Comments and suggestions:*

*Line 88 :  Eq. (3): Use the tilde over letter "Z" only, not over "Zp". Correct this in all subsequent occurrences.*
**Responses and Revisions:**

We have corrected all the "$\tilde{Z}_p$".

**17.** *Comments and suggestions:*

*Line 103-104*
**Responses and Revisions:**

In this sentence, the word "speed" has been changed to "speeds",   "electrode" has been changed to "electrodes" and "radius" has been changed to "radii".  This sentence has been changed to "$\omega_1$ and $\omega_2$ are the rotational speeds of the inner and outer electrodes, respectively. $\hat{r}$ is the ratio of the inner and outer radii".

**18.** *Comments and suggestions:*

*Line 105*
**Responses and Revisions:**

This sentence has been changed to "Sipkens et al. (2019) presented methods to calculate the transfer function of the CPMA".

**19.** *Comments and suggestions:*

*Line 107:* radical  àradial
**Responses and Revisions:**

It has been changed to "radial".

**20.** *Comments and suggestions:*

*Line 279-280: Fig. S2(a) and S2 (b) refer to 100-nm particles. The data for 150-nm particles are shown in Fig. S2(d) and S2(e).*

**Responses and Revisions:**

They have been changed.

**21.** *Comments and suggestions:*

*Line 300*: *Change to "The method to calculate PNSDve is described in section S1 of the Supplementary Material."*

**Responses and Revisions:**

It has been changed.

**22.** *Comments and suggestions:*

*Line 318*: *Severe à noticeable, significant. Leaded to à lead to*

**Responses and Revisions:**

They have been changed.

**23.** *Comments and suggestions:*

*Line 324*: *… but the method can suffer from multiple charging when decreasing the …*

**Responses and Revisions:**

It has been changed.

**24.** *Comments and suggestions:*

*Caption of Fig. 2*: *Subscripts and superscripts in this caption are not typed correctly.*

**Responses and Revisions:**

They have been changed.

**Response to James Radney**

We thank the reviewer for the constructive suggestions and comments concerning our manuscript entitled "Characterization of tandem aerosol classifiers for selecting particles: implication for eliminating multiple charging effect" (ID: amt-2021-436). Those comments are valuable and very helpful for improving our paper, as well as the important guiding significance to our studies. Below, we provide a point-by-point response to individual comment (Reviewer comments in italics, responses in plain font; page numbers refer to the revised version submitted last time)

**Specific comments:**

**1.** *The authors put quite a few equations inline in the text. It may improve readability to have them offset like the remainder, especially when the values are used multiple times throughout.*

**Responses and Revisions:**

Thank you for the advice. We have tried to move the equations to Appendix, but it seems this change makes difficulty to follow the text. Therefore, we didn't make changes in the current version, but we can modify if the reviewer insists.

**2.** *The authors need to double check that they are defining each variable to represent a single quantity. For example, m is used as mass in general or as the mass axis in a distribution and as the mode mass of the particle distribution.*

**Responses and Revisions:**

Thank you for the advice. We have rechecked the variables in the manuscript, and revised the ambiguous ones.

**3.** *It is still not clear to me how the charge (when > 1) is being included in the calculated transfer functions (e.g., paragraph starting on line 230 and Figure 2) since the authors are presenting everything as absolute mass instead of effective mass. Is the charge term somehow "baked into" the calculated transfer functions? Some clarification is warranted.*
*In the author response document, their response to technical comment 27 was "The particles are shown in figure 2 in actual dm and m, but when we calculate the resolution of DMA and CPMA, the mobility and effective mass are used. The resolution of CPMA can be calculated by Eq. R2, where m1 is the mass of singly charged particles which can be selected by the CPMA, i.e. effective mass."*
*Including this would be sufficient.*

**Responses and Revisions:**

Thank you for the advice. We have added it after the description of Fig. 2:

"The DMA-CPMA transfer function ($\Phi_{\text{DMA-CPMA}}$) for particles mentioned above, i.e. particles with $d_m$ of 100 nm and $m$ of 0.33 fg, is calculated in $\log(d_m)$-$\log(m)$ space, as shown in Fig. 2. The particles are shown in figure 2 in actual $d_m$ and $m$, but when we calculate the resolution of DMA and CPMA, the mobility and effective mass are used. The resolution of CPMA can be calculated by Eq. (15), where $m_1$ is the mass of singly charged particles which can be selected by the CPMA, i.e., effective mass".

**4.** *In the reviewer response, the authors include the following paragraph: "In our study, we also use scanning mode of CPMA after DMA selection to determine the mode mass of the selected particles, then we use the tandem setup of DMA and CPMA both at fixed mode to select particle at fixed mobility and mode mass i.e.*

*DMA and CPMA are used in a static configuration, no scanning for either instrument is used. In Figure 5a, DMA-CPMA is set to select singly charged particles with dm of 80 nm and m of 0.16 fg, while the doubly charged particles with dm of 119.3 nm and m of 0.32 fg will also be selected and the transfer function is presented as upper right region. Soot particles curve (red line) goes through the upper- right region which doubly charged particle can penetrate (dm of 113 nm~118 nm, m of 0.35 fg~0.39 fg). As a result, we conclude that multiple charging effect still exists when DMA-CPMA select soot particles with dm of 80 nm and m of 0.16 fg."*

*This is one of the best and most concise paragraphs describing the DMA-CPMA data in the manuscript. I highly recommend integrating it into the body of the manuscript.*

**Responses and Revisions:**

Thank you for the advice. We have included this description in the revised manuscript:

"When selecting particles with $d_m$ of 80 nm and $m$ of 0.16 fg, the corresponding DMA-CPMA transfer function is shown in Fig. 5a. DMA-CPMA is set to select singly charged particles with $d_m$ of 80 nm and $m$ of 0.16 fg, while the doubly charged particles with $d_m$ of 119.3 nm and $m$ of 0.32 fg will also be selected and the transfer function is presented as upper right region. Soot particles curve (red line) goes through the upper- right region which doubly charged particle can penetrate ($d_m$ of 113 nm~118 nm, $m$ of 0.35 fg~0.39 fg). As a result, we conclude that multiple charging effect still exists when DMA-CPMA select soot particles with $d_m$ of 80 nm and $m$ of 0.16 fg".

**5.** *In the reviewer response, the authors provide a discussion on the ordering of the DMA-AAC and the impact on static measurements in response to my comment 5. (Starts at "This population has one physical size (dae) but the dm"). This is an excellent discussion and I strongly recommend integrating it into the manuscript.*

**Responses and Revisions:**

The relevant discussion has been added after the discussion of multiple charging effect of DMA-AAC.

**Technical corrections:**

**6.** *Comments and suggestions:*

*Line 15: "specific size or mass."*
*Size is a nebulous term with respect to mobility (Dm) or aerodynamic (Dae) diameter. Maybe rewrite to "specific mobility diameter, aerodynamic diameter or mass, respectively."?*

**Responses and Revisions:**

Thank you for the comment. We have revised it as "Differential mobility analyzer (DMA), centrifugal particle mass analyzer (CPMA) and aerodynamic aerosol classifier (AAC) are commonly used to select particles with a specific size mobility diameter, aerodynamic diameter or mass, respectively".

**7.** *Comments and suggestions:*

*Line 17: "demonstrate"*
*Calculate?*

**Responses and Revisions:**

Revised.

**8.** *Comments and suggestions:*

**Line 18:** *"in static configurations." For flame generated soot particles.*
**Responses and Revisions:**
Revised.

**9. Comments and suggestions:**

**Line 22:** *"resolutions"*
*How is resolution defined in this sense?*
**Responses and Revisions:**
Resolutions refer to the electrical mobility resolution and mass resolution settings of DMA and CPMA, respectively. We have revised it to "our results show that the ability to remove multiply charged particles mainly depends on the particle morphology and resolution settings of the DMA and CPMA".

**10. Comments and suggestions:**

**Line 24:** *"Otherwise, our results indicate…" Repeated sentence from line 21.*
**Responses and Revisions:**
Sorry we didn't make it clear. The line 21 is about the ability to eliminate multiple charging effect for DMA-CPMA in a static configuration, while the Line 24 is about the DMA-AAC in a static configuration. We have revised the Line 21 and the Line 24 to "For DMA-CPMA in a static configuration, our results show that the ability to remove multiply charged particles mainly depends on the particle morphology and resolutions of the DMA and CPMA." and "As for DMA-AAC in a static configuration, the ability to eliminate particles with multiple charges is mainly related to the resolutions of classifiers", respectively.

**11. Comments and suggestions:**

**Line 36:** *"size dependence" size-dependent*
**Responses and Revisions:**
Revised.

**12. Comments and suggestions:**

**Line 44:** *"particles must be precharged"*
*Unclear what this means since, depending upon particle size, most aerosols possess a net charge. Instead, I think the authors are referring to bringing the particles to a known charge distribution by passing through a charge neutralizer or similar?*
**Responses and Revisions:**
Thank you for the comment. We have revised it to "The charge distribution of particles must be known by passing through a neutralizer or similar when classified by DMA or PMA. However, particles with higher-order charges and identical apparent mobility or mass-to-charge ratio can be selected simultaneously, which are referred to as the multiple charging effect.".

**13. Comments and suggestions:**

**Line 45:** *"mass-to-charge ratio," mass-to-charge ratio, respectively,*
**Responses and Revisions:**

Revised.

**14.** *Comments and suggestions:*

*Line 55:* *"of DMA-APM"*
*of the DMA and APM*
**Responses and Revisions:**
Revised.

**15.** *Comments and suggestions:*

*Line 64:* *"is that no charging process is needed"*
*Should this be "the charge state of the particles does not need to be known"?*
**Responses and Revisions:**
Thank you for the comment. We have revised it to "The advantage of utilizing an AAC is that the charge state of the particles does not need to be known in particle classification compared with the aforementioned classifiers".

**16.** *Comments and suggestions:*

*Line 71:* *"(Johnson et al., 2018)."*
*This is not a peer-reviewed manuscript but rather a conference presentation.*
**Responses and Revisions:**
Thank you for the comment. We have deleted it and cited another reference.

**17.** *Comments and suggestions:*

*Line 73:* *"APM(Yao"*
*Missing a space between APM and (Yao*
**Responses and Revisions:**
Revised.

**18.** *Comments and suggestions:*

*Line 97:* *"elemental"*
*elementary*
**Responses and Revisions:**
Revised.

**19.** *Comments and suggestions:*

*Line 98:* *"Zp*"*
*$Z_p^*$ needs to be defined. Also, the symbol used for Zp* on line 98 and in the table of symbols is different than that used in Eq. 2.*
**Responses and Revisions:**
Thank you for the comment. It has been revised to "the centroid mobility, $Z_p^*$, selected by the DMA is defined

as". All the symbols have been changed to "$Z_p^*$".

**20. *Comments and suggestions:***

*Line 102: "DMA, respectively." DMA electrodes, respectively.*
**Responses and Revisions:**
Revised.

**21. *Comments and suggestions:***

*"$d_{m\,n,max}$ and $d_{m\,n,min}$"*
*Is there supposed to be a space or a , between m and n?*
**Responses and Revisions:**
Thank you for the comment. We have revised it to "$d_{mn,max}$ and $d_{mn,min}$".

**22. *Comments and suggestions:***

*Line 111: "Zp"*
*Z should be italic.*
**Responses and Revisions:**
Revised.

**23. *Comments and suggestions:***

*Line 113: "The construction of the CPMA is similar to the APM,"*
*How so? No discussion on the construction of the APM has been provided.*
**Responses and Revisions:**
Thank you for the comment. The construction of APM, "The APM consists of two coaxial electrodes which are rotating at an equal angular velocity and a voltage is applied between these electrodes to create an electrostatic field", has been added.

**24. *Comments and suggestions:***

*Line 123: $\alpha$ and $\beta$, in this usage, should be explicitly defined to avoid confusion with other quantities (e.g., $\alpha_{abs}$, $\beta_{DMA}$, etc.)*
**Responses and Revisions:**
Thank you for the comment. We have added "$\alpha$ and $\beta$ are the azimuthal flow velocity distribution parameters" and the symbols of $\alpha$ and $\beta$ have also been added into the table A1. The ratio of flow rates of sample flow and sheath flow of DMA and AAC have been specifically denoted as $\beta_{DMA}$ and $\beta_{AAC}$, respectively.

**25. *Comments and suggestions:***

*Line 136: "Assuming a plug flow," Delete "a".*
**Responses and Revisions:**
Revised.

**26. *Comments and suggestions:***

***Line 139 and 140:*** *Formatting on functions should be consistent throughout. Min, max, exp should all be formatted similarly. Also, min (and max, preferably) should be explicitly defined after use as the minimum (and maximum) of the quantities in the brackets since min is also the abbreviation for minutes.*

**Responses and Revisions:**

Thank you for the comment. We have made the format consistent throughout and added the descriptions of min and max:

"$r_a = \min\left\{r_{2\_CPMA}, \max\{r_{1\_CPMA}, G_0(r_{1\_CPMA})\}\right\}$, $\qquad\qquad$ (12)

$r_b = \min\left\{r_{2\_CPMA}, \max\{r_{1\_CPMA}, G_0(r_{2\_CPMA})\}\right\}$, $\qquad\qquad$ (13)

$G_0(r_L) = r_c + \left(r_L - r_c + \frac{C_3}{C_4}\right)\exp(-C_4 L\bar{v}) - \frac{C_3}{C_4}$, $\qquad$ (14)

where $G_0(r)$ is the operator used to map the final radial position of the particle to its position at the inlet and $\bar{v}$ is the average flow velocity. min{} and max{} are the minimum and maximum values of the quantities in the brackets.".

**27. *Comments and suggestions:***

***Line 161:*** *"τ"*
*τ should be italic.*

**Responses and Revisions:**

Revised.

**28. *Comments and suggestions:***

***Line 162:*** *"denote" denoted*

**Responses and Revisions:**

Revised.

**29. *Comments and suggestions:***

***Line 165 and 166:*** *"SCPM" and "SLPM" are non-standard units and should be explicitly defined; e.g. SLPM (standard L per min, flow in L min$^{-1}$ converted from ambient to T = and P = ).*

**Responses and Revisions:**

Thank you for the comment. We have revised it to "with a propane flow of 74.8 SCPM (standard mL per minute, flow in mL min$^{-1}$ converted from ambient to T =298.15 K and P = 101.325 kPa) and an air flow rate of 12 SLPM (Standard L per minute, flow in L min$^{-1}$ converted from ambient to T =298.15 K and P = 101.325 kPa)".

**30. *Comments and suggestions:***

***Line 171 and 173:*** *"Qsh/Qa = 10"*
*the $\beta_{DMA}$ and $\beta_{AAC}$ formulation should be used for consistency throughout.*

**Responses and Revisions:**

Thank you for the comment. We have revised them to "DMA (Model 3081, TSI Inc., USA, $\beta_{DMA} = 10$)" and "AAC (Cambustion, Ltd., UK, $\beta_{AAC} = 10$)", respectively.

**31. *Comments and suggestions:***

*Line 171: "monodisperse"*
*This is not technically correct as written and the authors should remind the reader that the stream is monodisperse in $Z_p$.*
**Responses and Revisions:**
Thank you for the comment. It has been revised to "the mobility-selected aerosol flow was switched between two parallel lines and fed into the CPMA (Cambustion Ltd., UK) and AAC (Cambustion, Ltd., UK, $\beta_{AAC}$ = 10)".

**32. *Comments and suggestions:***

*Line 174: "The particle mass (m) and aerodynamic diameter ($d_{ae}$) were determined by the scanning mode"*
*This is not exactly true as written. The distributions of particle number density as a function of particle mass and aerodynamic were measured and mode particle mass was then determined from a fit of that distribution.*
**Responses and Revisions:**
Thank you for the comment. We have revised it to "The distributions of particle number concentration as a function of particle mass ($m$) and aerodynamic diameter ($d_{ae}$) were measured by the scanning mode of the CPMA and AAC".

**33. *Comments and suggestions:***

*Line 175: "CPMA and AAC, while" CPMA and AAC, respectively, while*
**Responses and Revisions:**
Revised.

**34. *Comments and suggestions:***

*Line 178: "SMPS"*
*Needs to be defined.*
**Responses and Revisions:**
Thank you for the comment. We have revised it to "the number size distribution of the generated soot particles was measured by a scanning mobility particle sizer (SMPS)".

**35. *Comments and suggestions:***

*Line 179: "soot particles did not change during the whole experiment." It is unclear to what change the authors are referring.*
**Responses and Revisions:**
Thank you for the comment. We have revised it to "to ensure the number size distribution of generated soot particles did not change during the whole experiment".

**36. *Comments and suggestions:***

*Line 180: "The m and $d_{ae}$ distributions were fitted to log- normal distributions; thus, the modes m and $d_{ae}$ for the mobility-selected particles were determined"*

*According to this sentence, m and $d_{ae}$ represent multiple quantities, both the axis and the modal value. Separate variables should be used for each quantity to avoid confusion.*

**Responses and Revisions:**

Thank you for the comment. We have revised it to "The $m$ and $d_{ae}$ distributions were fitted to log-normal distributions; thus, the modal values denoted as $m_c$ and $d_{ae,c}$ for the mobility-selected particles were determined".

**37. *Comments and suggestions:**

**Line 181:** *"equation of log-normal distribution used in this study is expressed as"*
*This equation only applies to $N(d_p)$ and not to the m or $d_{ae}$ fits that the authors are referring.*

**Responses and Revisions:**

Thank you for the comment. We have revised it to

"$N(m) = \frac{N_0}{\sqrt{2\pi}\ln\sigma_{\mathrm{m}}}\exp(\frac{-(\ln(m)-\ln(m_{\mathrm{c}}))^2}{2(\ln\sigma_{\mathrm{m}})^2})$

$N(d_{\mathrm{ae}}) = \frac{N_0}{\sqrt{2\pi}\ln\sigma_{\mathrm{dae}}}\exp(\frac{-(\ln(d_{\mathrm{ae}})-\ln(d_{\mathrm{ae,c}}))^2}{2(\ln\sigma_{dae})^2})$ ,          (20)

where $\sigma_{\mathrm{m}}$ and $\sigma_{\mathrm{dae}}$ are the geometric standard deviation of $m$ and $d_{\mathrm{ae}}$ distributions, respectively. $m_c$ and $d_{\mathrm{ae,c}}$ are the geometric mean of $m$ and $d_{\mathrm{ae}}$, respectively".

**38. *Comments and suggestions:**

**Line 183:** *"where σ is the geometric standard deviation and μ is the geometric mean"*
*Did the distributions of m and Dae have the same σ and μ? That is what is implied.*

**Responses and Revisions:**

No, different distributions have different $\sigma$ and $\mu$. Sorry for the confusing expression. We have revised it to "$\sigma$ is the geometric standard deviation and $\mu$ is the geometric mean which are fitted from $m$ or $d_{\mathrm{ae}}$ distributions, respectively.".

**39. *Comments and suggestions:**

**Line 194:** *"electrical diameter" electrical mobility diameter*
**Responses and Revisions:**

Revised.

**40. *Comments and suggestions:**

**Line 196:** *"ρeff"*
*The mathematical relationship for $\rho_{eff}$ has not been defined anywhere.*

**Responses and Revisions:**

Thank you for the comment. The mathematical relationship "$\rho_{\mathrm{eff}} = \frac{6m}{\pi d_m^3}$" has been added for $\rho_{\mathrm{eff}}$.

**41. *Comments and suggestions:**

**Line 202:** *"where Φ and Ω are the transfer functions of each classification system expressed by subscripts."*
*Should be "where Φ and Ω are the transfer functions of the combined and individual classification systems expressed by subscripts, respectively."*

**Responses and Revisions:**

Revised.

**42. Comments and suggestions:**

*Line 206: "dm = 80 nm"*

*Why is the dm, m and dae different in this sentence than in the previous?*

**Responses and Revisions:**

Sorry for the mistake. We have revised it to "$d_m$ = 100 nm, $Q_{DMA}$ = 0.3 L min$^{-1}$, $\beta_{DMA}$ = 0.1, $m$ = 0.33 fg, $Q_{CPMA}$=0.3 L min$^{-1}$, $R_m$ = 8, $d_{ae}$ = 68.3 nm, $Q_{AAC}$ = 0.3 L min$^{-1}$, $\beta_{AAC}$ = 0.1".

**43. Comments and suggestions:**

*Line 208: "included 600 points, respectively"*

*600 points each?*

**Responses and Revisions:**

We have revised it to "which included 600 points each".

**44. Comments and suggestions:**

*Line 210: ">m2,max, from" How far < or > the respective values were investigated?*

*missing "and"*

**Responses and Revisions:**

We have revised it to "The ranges of $d_m$, $m$ and $d_{ae}$ used in the calculations were from 0.8 times $d_{m1,min}$ to 1.2 times $d_{m2,max}$, and from 0.8 times $m_{1,min}$ to 1.2 times $m_{2,max}$, from 0.8 times $d_{ae,min}$ to 1.2 times $d_{ae,max}$, respectively.".

**45. Comments and suggestions:**

*Line 213: "The DMA-CPMA transfer function is calculated in log(dm)-log(m) space, as shown in Fig. 2."*

*"DMA-CPMA transfer function ($\Phi_{DMA-CPMA}$)". What is the transfer function calculated for?*

**Responses and Revisions:**

Thank you for the comment. It is calculated for particles mentioned in Line 206 and 207, i.e. particles with dm of 100 nm and m of 0.33 fg. It has been revised to "The DMA-CPMA transfer function ($\Phi_{DMA-CPMA}$) for particles mentioned above, i.e. particles with $d_m$ of 100 nm and $m$ of 0.33 fg, is calculated in log($d_m$)-log($m$) space, as shown in Fig. 2".

**46. Comments and suggestions:**

*Line 215 and 216: "nm"*

*nm should not be italicized when used as a unit.*

**Responses and Revisions:**

Thank you for the comment. It has been revised.

**47. Comments and suggestions:**

**Line 217:** *"and smaller than 3 for aspherical particles"*
*Dfm can be larger than 3 for particles that are non-spherical at small $D_m$ and approach spherical as $D_m$ increases.*
**Responses and Revisions:**
Thank you for the comment. We have revised it to "In general, $D_{fm}$ equals 3 for spherical particles and smaller than 3 for aspherical particles, although $D_{fm}$ can be larger than 3 for particles that are non-spherical at small $d_m$ and approach spherical as $d_m$ increases".

**48. *Comments and suggestions:**

**Line 219:** *"Under this specific operation condition" What specific operating condition? Please explain.*
**Responses and Revisions:**
Sorry we didn't make it clear. This specific operating condition refers to "$d_m$ = 100 nm, $Q_{DMA}$ = 0.3 L min$^{-1}$, $\beta_{DMA}$ = 0.1, $m$ = 0.33 fg, $Q_{CPMA}$=0.3 L min$^{-1}$, $R_m$ = 8" which mentioned in Line 206 and Line 207.

**49. *Comments and suggestions:**

**Line 220:** *"spherical particle population (black line)"*
*Is this a "theoretical" spherical particle population? What would be the $\rho_{eff}$ of these particles?*
**Responses and Revisions:**
Yes, this is a "theoretical" spherical particle population. The effective densities of this particles are the same as nominal particles with dm of 100 nm and m of 0.33 fg, i.e. 630.3 kg m$^{-3}$.

**50. *Comments and suggestions:**

**Line 220:** *"classification region"*
*What is the classification region? Please elaborate.*
**Responses and Revisions:**
Thank you for the comment. The classification region is presented as the rainbow color blocks. We have revised it to "no overlap was observed between the spherical particle population (black line) and the classification region (the colored blocks) for doubly charged particles".

**51. *Comments and suggestions:**

**Line 231:** *"than the slope of a line connecting (dm, m) = (dm2,min, m2,max)(dm1,m1) (as PP0 shown in Fig. 2)"*
*PP0 is not clearly shown in Fig. 2 and this was **the** source of my confusion. From the figure, it appears that PP0 is drawn as the Dfm = 2.28 line. So, while the Dfm = 3 discussion seems reasonable, the Dfm = 2.28 does not. In contrast, for Figure 4 the PP$_0$ line is clearly visible making the discussion much easier to understand. My recommendation is to either switch the ordering of the DMA-CPMA and DMA-AAC sections or to add an additional plot to Figure 2 at a larger dm1 where PP0 is clearly visible.*
**Responses and Revisions:**
Sorry we didn't make it clear. We have added a dashed line to indicate PP$_0$. It is easier to compare the $D_{fm}$ = 2.28 and PP$_0$.

**52. Comments and suggestions:**

*Line 236: ". Accordingly, the ideal condition…" Under static operation at this set point*
**Responses and Revisions:**
Thank you for the comment. We have revised it to "Accordingly, the ideal condition under static operation to completely eliminate the multiply charged particles is".

**53. Comments and suggestions:**

*Line 240: "Eq. (26) gives instructions in actual operation" It is unclear how Eq. 26 gives instructions.*
**Responses and Revisions:**
Thank you for the comment. We have added an equation of Rm:
"The mass resolution ($R_m$) of CPMA is related to particles mobility. When selecting the particles with mass of $m_1$ and mobility of $B_1$, the $R_m$ can be calculated by

$$R_m = \frac{2\pi B_1 L_{CPMA} r_c^2 \omega^2 m_1}{Q_{CPMA}},$$ (15)

where $\omega$ is the equivalent rotational speed calculated by $\omega = \alpha + \frac{\beta}{r_c^2}$, $m_1$ is the nominal mass that the CPMA can select. The limiting mass can be calculated by

$$m_{n,min}^{n,max} = n \cdot m_1 \pm \frac{Q_{CPMA}}{2\pi B_{n,min}^{n,max} L_{CPMA} r_c^2 \omega^2} = n \cdot m_1 \pm \frac{m_1}{R_m} \cdot \frac{B_1}{B_{n,min}^{n,max}},$$ (16)

where $m_{n,min}^{n,max}$ and $B_{n,min}^{n,max}$ are the maximum and minimum mass and corresponding mobility of particles bearing number of elementary charges of $n$ that the CPMA can select, respectively."
and this sentence has been changed to "Combining Eq. (16), equation (26) gives instructions in actual operation to eliminate multiply charged particles. When selecting particles of certain $d_m$ and $m$, by decreasing $Q_{CPMA}$, or increasing $\omega$ and $\beta_{DMA}$, i.e., by increasing the resolution of the measurement, the potential of multiply charged particles is reduced".

**54. Comments and suggestions:**

*Line 244: "and the slope of PP0 derived from the actual condition"*
*This is unclear as written. How is the slope of $PP_0$ derived from actual conditions? Weren't the transfer functions from which $PP_0$ is determined theoretically calculated?*
**Responses and Revisions:**
We have revised it to "the key to evaluating whether there is a multiple charging effect lies in the particle morphology ($D_{fm}$) and the slope of $PP_0$ calculated from Eq. (26) theoretically".

**55. Comments and suggestions:**

*Line 246: "According to the theoretical calculation described in Kuwata (2015), the slope of PP0 of 3.55 was derived when the DMA-APM selects the same…"*
*Kuwata did not calculate a $PP_0$, so it is unclear where this value is coming from. What was the slope of $PP_0$ for the DMA-CPMA for reference? Should the value of 3.55 be 2.55? If not, are the authors claiming that the DMA-APM would be unable to separate spherical particles (Dfm = 3) under these theoretical conditions? I completely agree that the DMA-APM is more susceptible to multiple charging, but this comparison to the APM needs to be clarified or removed.*
**Responses and Revisions:**
Thank you for the advice. We have removed it.

**56. *Comments and suggestions:***

***Line 251:*** *"Rm = 8"*
*What is Rm? This is the first instance of it in the manuscript.*
**Responses and Revisions:**
Thank you for the comment. The description of $R_m$ has been added which can be found in the reply to comment 53.

**57. *Comments and suggestions:***

***Line 254:*** *"the slope" the critical slope*
**Responses and Revisions:**
Revised.

**58. *Comments and suggestions:***

***Line 251:*** *"contour lines in Fig. 3"*
*How were these contours calculated? From Eq. 26? If so, how was dm2 determined?*
**Responses and Revisions:**
Yes, they were calculated from Eq. 26. We have revised this sentence to "Here, we simulate the critical slope of $PP_0$ when selecting different $d_m$ and $m$ under the common selecting conditions ($\beta_{DMA}$ = 0.1, $Q_{CPMA}$=0.3 L min$^{-1}$, $R_m$ = 8) using Eq. 26, which is represented as contour lines in Fig. 3". The values of $d_{m2,min}$ were calculated from Eq. 1 and $(1 \pm \beta_{DMA}) \cdot Z_p^*$, and we have added the description in Line 108 and Line 109 : "The limiting electrical mobilities that DMA can select are $(1 \pm \beta_{DMA}) \cdot Z_p^*$. The maximum and minimum values of $d_m$ for particles with $n$ charges can be derived combining $(1 \pm \beta_{DMA}) \cdot Z_p^*$ and Eq. 1, and denote as $d_{mn,max}$ and $d_{mn,min}$, respectively".

**59. *Comments and suggestions:***

***Line 259:*** *"mobility diameters larger than 200 nm, while it fails to eliminate …"*
*As shown by the circles and squares in Figure 3.*
**Responses and Revisions:**
Revised.

**60. *Comments and suggestions:***

***Line 263:*** *"to charge aerosol particles"*
*To Boltzmann distribution or a known charge state?*
**Responses and Revisions:**
Thank you for the comment. We have revised it to "The advantage of the AAC versus the CPMA is that there is no need for a neutralizer to charge aerosol particles to a known charge state".

**61. *Comments and suggestions:***

***Line 264:*** *"AAC cannot constrain the properties of aspherical particles as monodisperse as DMA or CPMA classification"*

*Unclear as written.*

**Responses and Revisions:**

It has been revised to "However, aspherical particles with different mass can be selected by the AAC as having identical aerodynamic diameter".

**62. *Comments and suggestions:**

*Line 267: "selecting the same representative particles" The same as what? Please give values.*

**Responses and Revisions:**

The same particles as particles used in DMA-CPMA calculation, which is mentioned in Line 204 and Line 205. This sentence and the Line 273 are repeated, so we deleted this sentence and revised the Line 273 to "To simulate the transfer function of the DMA-AAC, the same particles ($d_m$ = 100 nm, $m$ = 0.33 fg, $D_{fm}$ = 2.28) as those used in the calculations of the DMA-CPMA were selected".

**63. *Comments and suggestions:**

*Line 268: "aspherical particles can be expressed as follows"*
*The log(Dae) is expressed on the next line. Not aspherical particles.*

**Responses and Revisions:**

Thank you for the comment. We have revised it to "the relationship of $d_{ae}$ and $d_m$ of aspherical particles can be expressed as follows".

**64. *Comments and suggestions:**

*Line 273: "the same particles" Please give values as a reminder.*

**Responses and Revisions:**

Thank you for the comment. It has been revised to "To simulate the transfer function of the DMA-AAC, the same particles ($d_m$ = 100 nm, $m$ = 0.33 fg, $D_{fm}$ = 2.28) as those used in the calculations of the DMA-CPMA were selected. The corresponding $d_{ae}$ was numerically solved using the known mass–mobility relationship. The transfer function of the DMA-AAC is shown in $\log(d_{ae})$-$\log(d_m)$ (Fig. 4a).".

**65. *Comments and suggestions:**

*Line 276: "are in parallel for the DMA-AAC"*
*Unclear as written. I think the authors are referring to the fact that transfer function will have the same $D_{ae}$ and different $D_m$?*

**Responses and Revisions:**

Thank you for the comment. We have revised it to "In the transfer function of DMA-CPMA, the classification regions of singly charged particles and doubly charged particles are on the diagonal. The oblique line of particles population is more likely to go through the region of doubly charged particles in the transfer function of DMA-CPMA. The transfer functions of singly charged and doubly charged particles are in parallel for the DMA-AAC, suggesting that the particles population is less likely to overlap with the region of multiply charged particles".

**66. *Comments and suggestions:**

***Line 277:*** *"the example setups" What example setups?*

**Responses and Revisions:**

It was mentioned in Line 206 and Line 207. We have revised it to "Using the example setups ($d_m$ = 100 nm, $Q_{DMA}$ = 0.3 L min$^{-1}$, $\beta_{DMA}$ = 0.1, $d_{ae}$ = 68.3 nm, $Q_{AAC}$ = 0.3 L min$^{-1}$, $\beta_{AAC}$ = 0.1.) of the DMA-AAC, truly monodispersed particles are selected for spherical particles and typical soot particles.".

**67. *Comments and suggestions:***

***Line 290:*** *"which is the case for most atmospheric aerosol particles." What is the case? This $D_{fm}$ is smaller than for most aerosols.*

**Responses and Revisions:**

Thank you for the comment. We have revised it to "This $D_{fm}$ is smaller than that for most aerosols".

**68. *Comments and suggestions:***

***Line 292:*** *"is required" "may be required"*

**Responses and Revisions:**

Revised.

**69. *Comments and suggestions:***

***Line 295:*** *"When increasing $\beta_{AAC}$ to 0.3"*

*Increasing is a misnomer here since an increase in $\beta_{AAC}$ is a decrease in resolution. Please remind the reader of this distinction.*

**Responses and Revisions:**

Thank you for the comment. We have revised it to "When increasing $\beta_{AAC}$ to 0.3 (decreasing the resolution of AAC) and leaving $\beta_{DMA}$ unchanged, the transfer function becomes broader".

**70. *Comments and suggestions:***

***Line 302:*** *"the corresponding dae and m were determined using the AAC and CPMA scan modes"*

*This isn't exactly true as written. The distributions of number density as a function of $D_{ae}$ and m were determined by the scans. These distributions were then fit to a log-normal to determine the modal values and from these values the $\rho_{eff}$ were determined.*

**Responses and Revisions:**

Thank you for the comment. It has been revised to "For each mobility-selected particles, the distributions of number density as a function of $d_{ae}$ and $m$ were determined by the scans. These distributions were then fit to a log-normal to determine the modal values and from these values the $\rho_{eff}$ were determined".

**71. *Comments and suggestions:***

***Line 304:*** *"measured spectral density" Measured distributions?*

**Responses and Revisions:**

Thank you for the comment. It has been changed to "Representative plots for the measured distributions of $m$ and $d_{ae}$ of particles with $d_m$ of 150 nm and 250 nm are shown in Fig. S2".

**72. *Comments and suggestions:***

*Line 305: "The results are summarized in Table 2."*
*How were the uncertainties in Table 2 determined? 1σ standard deviation of multiple measurements? Or something else? Please describe.*
**Responses and Revisions:**
Thank you for the comment. We have added the description of the uncertainties: "The uncertainties of $d_{ae,c}$ and $m_c$ were standard deviation of multiple measurements".

**73. *Comments and suggestions:***

*Line 306: kf has units of mass.*
**Responses and Revisions:**
Thank you for the comment. The unit of "fg" has been added.

**74. *Comments and suggestions:***

*Line 308: "two methods" Which two methods?*
**Responses and Revisions:**
Thank you for the comment. We have revised it to "The effective densities of generated soot particles vary from >500 kg m$^{-3}$ at $d_m$ = 80 nm to <300 kg m$^{-3}$ at $d_m$ of 250 nm determined by DMA-CPMA and DMA-AAC".

**75. *Comments and suggestions:***

*Line 308: "the deviation" What deviation?*
**Responses and Revisions:**
Thank you for the comment. We have revised it to "the deviation of values of $\rho_{eff}$ measured by DMA-CPMA and DMA-AAC monotonically decreases with increasing particle size".

**76. *Comments and suggestions:***

*Line 316: "the corresponding transfer function" DMA-CPMA transfer function?*
**Responses and Revisions:**
Thank you for the comment. It has been revised to "the corresponding DMA-CPMA transfer function is shown in Fig. 5a".

**77. *Comments and suggestions:***

*Line 317: "The particle population" Shown by the red $D_{fm}$ = 2.28 line?*
**Responses and Revisions:**
Thank you for the comment. It has been revised to "The particle population (red line) overlaps the transfer function region of doubly charged particles, suggesting the potential interferences of doubly charged particles in DMA-CPMA selection".

**78. *Comments and suggestions:***

*Line 320: "particles number aerodynamic size distribution"*
*Should be "particles number density aerodynamic size distribution"*
**Responses and Revisions:**
Revised.

**79. *Comments and suggestions:***

*Line 324: "The mean dae values" For particles with dm = 80 nm?*
**Responses and Revisions:**
Yes, it is. We have revised it to "For particles with $d_m$ = 80 nm, the mean $d_{ae}$ values were 53.9 nm, 60.6 nm and 70.9 nm, and the corresponding $d_{ae}$ values were calculated as 51.5 nm, 62.0 nm and 70.7 nm using Eq. (1) and Eq. (16)".

**80. *Comments and suggestions:***

*Line 327: "In contrast, ..."*
*In contrast to what?*
**Responses and Revisions:**
Thank you for the comment. It has been deleted.

**81. *Comments and suggestions:***

*Line 332: "PP0"*
*Subscript 0.*
**Responses and Revisions:**
Revised.

**82. *Comments and suggestions:***

*Line 363: "26.7±3.0%"*
*What is the unit on 26.7? Is it %? If so, should be written as (26.7±3.0) % or 26.7 % ±3.0 % to avoid the confusion of 3.0 % being a relative value and the absolute being 26.7 ±0.8. Other values in this paragraph need to be similarly corrected.*
**Responses and Revisions:**
Thank you for the comment. We have revised all the units.

**83. *Comments and suggestions:***

*Line 387: "Under the same setups" Same as what?*
**Responses and Revisions:**
In this sentence, we tried to discuss the influence of particles $D_{fm}$ and selected size on the multiple charging effect when the setups of DMA-CPMA is unchanged. We are sorry for the confusing expression. We have deleted it and this sentence has been revised to "This tandem system is more sensitive to multiple charging effect with decreasing $D_{fm}$ and decreasing nominal size of particles.".

**84. *Comments and suggestions:***

***Table A1:*** *Thank you for including this table. But, can you please sort values alphabetically to assist the reader in locating values and include the corresponding units where appropriate.*

**Responses and Revisions:**

Thank you for the comment. It has been changed.

**85. *Comments and suggestions:***

***Table 2:*** *"M (fg)"*

*Is this the modal mass from the log-normal fit? This is the wrong symbol.*

**Responses and Revisions:**

It has been revised to "$m_c$".

**86. *Comments and suggestions:***

***Table 3:*** *$f_N$ and $f_{abs}$ are not mathematically defined in the body of the manuscript. Can the authors provide the calculated MAC for each size and the overall? And should "MAC overestimation" have a units associated with it?*

**Responses and Revisions:**

Thank you for the comment. We have added the description of $f_N$, $f_{abs}$ and overestimation of MAC:

"The fractional number concentration of particles with different charging state is expressed as follows,

$$f_{N,n} = \frac{\int_{d_{ae,min}}^{d_{ae,max}} \frac{dN_n}{dlog(d_{ae})} dlog(d_{ae})}{\sum_{i=1}^{3} \int_{d_{ae,min}}^{d_{ae,max}} \frac{dN_n}{dlog(d_{ae})} dlog(d_{ae})}, \tag{29}$$

where $f_{N,n}$ and $N_n$ are the fractional number concentration and number concentration of particles bearing n charges. $d_{ae,min}$ and $d_{ae,max}$ denote the minimum and maximum value of $d_{ae}$ scanned by AAC, respectively. The uncertainties are standard deviations of multiple measurements".

"The fractional absorption coefficient for particles with different charging state was calculated as follows,

$$f_{abs,n} = \frac{\int_{d_{ve,low,n}}^{d_{ve,high,n}} \frac{d\alpha_{abs,n}}{dlog(d_{ve})} dlog(d_{ve})}{\sum_{i=1}^{3} \int_{d_{ve,low,n}}^{d_{ve,high,n}} \frac{dN_n}{dlog(d_{ve})} dlog(d_{ve})}, \tag{30}$$

where $f_{abs,n}$ and $\alpha_{abs,n}$ are the fractional absorption coefficient and absorption coefficient of particles bearing n charges, respectively. $d_{ve,low,n}$ and $d_{ve,high,n}$ denote the minimum and maximum value of $d_{ve}$ of particles with n charges, which are converted from $d_{ae,low}$ and $d_{ae,high}$ scanned by AAC, respectively. The overestimation of mass absorption cross-section (MAC) was calculated by

$$\frac{\Delta MAC}{MAC} = \frac{\frac{\alpha_{abs,tot}}{m_p N_{tot}} - \frac{f_{abs,1} \cdot \alpha_{abs,tot}}{m_p \cdot f_{N,1} \cdot N_{tot}}}{\frac{f_{abs,1} \cdot \alpha_{abs,tot}}{m_p \cdot f_{N,1} \cdot N_{tot}}} = \frac{f_{N,1}}{f_{abs,1}} - 1, \tag{31}$$

where $\alpha_{abs,tot}$ and $N_{tot}$ is the total absorption coefficient and number concentration of particles selected by DMA-CPMA, respectively. $m_p$ is the actual mass of singly charged particles selected by DMA-CPMA. The uncertainties were calculated from propagation of errors"

**87. *Comments and suggestions:***

***Figure 2 caption:*** *Please note that this plot is in log-log space as a reminder for the reader. It is not clear from just looking at the axes.*

**Responses and Revisions:**

Thank you for the comment. We have revised it to "Example of the DMA-CPMA transfer function of flame-generated soot particles (Pei et al., 2018) in $log(m)$-$log(d_m)$".

**88.** *Comments and suggestions:*

*Line 533:* *"DMA and CPMA."*
DMA and CPMA, respectively.
**Responses and Revisions:**
Revised.

**89.** *Comments and suggestions:*

*Figure 3:* *What are the minimum values on the dm and m axis?*
**Responses and Revisions:**
Thank you for the comment. The minimum values on the $d_m$ and $m$ axis has been added.

[Figure]

**90.** *Comments and suggestions:*

*Line 537:* *"The contour lines denote the slope" Critical slope?*
**Responses and Revisions:**
Revised.